# ViEEG: Hierarchical Visual Neural Representation for EEG Brain Decoding

**Minxu Liu** [1]  **Donghai Guan** [1]  **Chuhang Zheng** [2]  **Chunwei Tian** [3]  **Jie Wen** [4]  **Qi Zhu** [2]

## Abstract

Understanding and decoding brain activity into visual representations is a fundamental challenge at the intersection of neuroscience and artificial intelligence. While electroencephalogram (EEG) visual decoding has shown promise due to its non-invasive and low-cost nature, existing methods suffer from Hierarchical Neural Encoding Neglect (HNEN), a critical limitation in which flat neural representations fail to model the brain's hierarchical visual processing. Inspired by the hierarchical organization of visual cortex, we propose ViEEG, a neuro-inspired framework that addresses HNEN. ViEEG decomposes each visual stimulus into three biologically aligned components, namely contour, foreground object, and contextual scene, which serve as anchors for a three-stream EEG encoder. These EEG features are progressively integrated via cross-attention routing, simulating cortical information flow from low-level to high-level vision. We further adopt hierarchical contrastive learning for EEG-CLIP representation alignment, enabling zero-shot object recognition. Extensive experiments on THINGS-EEG dataset demonstrate that ViEEG significantly outperforms previous methods by a large margin in both subject-dependent and subject-independent settings. Results on THINGS-MEG dataset further confirm ViEEG's generalization to different neural modalities. ViEEG not only advances the performance frontier but also sets a new paradigm for EEG brain visual decoding. Our code is available at https://github.com/LauMason/ViEEG.

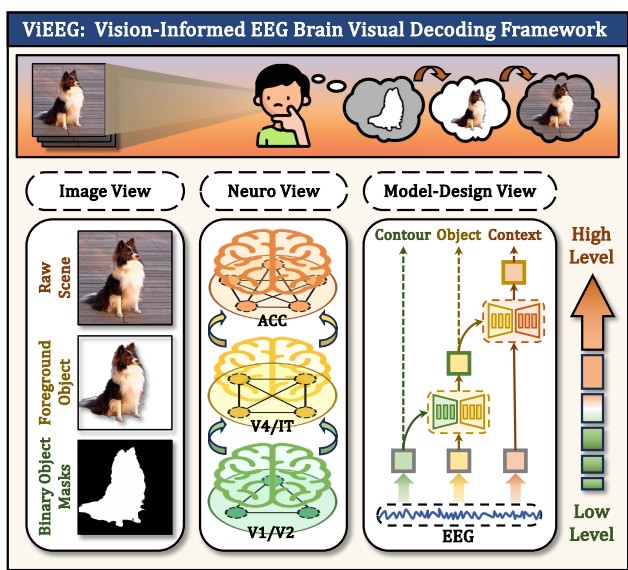

*Figure 1.* The diagram of our proposed triple-view hierarchical framework for EEG visual decoding.

[1]College of Computer Science and Technology, Nanjing University of Aeronautics and Astronautics, Nanjing, China [2]College of Artificial Intelligence, Nanjing University of Aeronautics and Astronautics, Nanjing, China [3]Harbin Institute of Technology, Harbin, China [4]Harbin Institute of Technology (Shenzhen), Shenzhen, China. Correspondence to: Donghai Guan <dhguan@nuaa.edu.cn>, Qi Zhu <zhuqi@nuaa.edu.cn>.

*Proceedings of the $43^{rd}$ International Conference on Machine Learning*, Seoul, South Korea. PMLR 306, 2026. Copyright 2026 by the author(s).

## 1. Introduction

Electroencephalogram (EEG) visual decoding aims to bridge the gap between cortical dynamics and machine perception by reconstructing visual experiences from brain activity (Du et al., 2023). As a pivotal modality in brain-computer interfaces, EEG (Craik et al., 2019) has gained prominence due to its cost-effectiveness, portability, and millisecond-level temporal resolution. Recent years have witnessed substantial progress in EEG decoding methodologies (Liu et al., 2024; Bai et al., 2023). However, unlike functional magnetic resonance imaging (fMRI) signals (Worsley et al., 2002; Allen et al., 2022) that have established sophisticated decoding frameworks (Horikawa & Kamitani, 2017; Sun et al., 2023; Zhou et al., 2024), EEG visual decoding remains in infancy, with significant challenges unaddressed, particularly in modeling the brain hierarchical visual processing mechanisms.

Contemporary EEG brain decoding approaches predominantly establish direct visual-semantic mappings between EEG and stimulus images. For instance, Mb2C (Wei et al., 2024) employed contrastive learning for direct EEG-image embedding alignment with contrastive language-image pre-

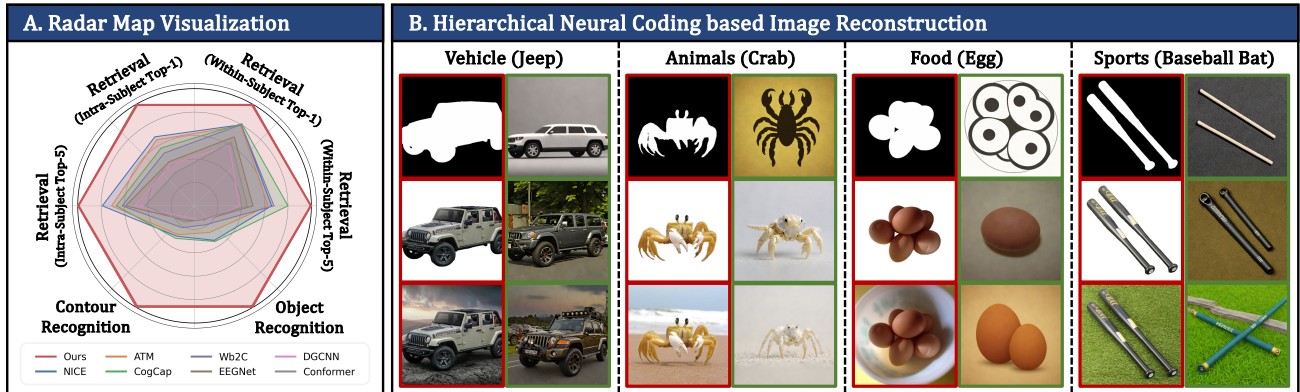

*Figure 2.* EEG visual retrieval, classification, and reconstruction tasks of ViEEG. (A) Our ViEEG achieves the highest performance compared to other methods in EEG visual decoding tasks. (B) Hierarchical image reconstruction results of ViEEG (red: ground-trues images; green: EEG reconstructed images).

training (CLIP) (Radford et al., 2021; Ramesh et al., 2022), while NICE (Song et al., 2024) developed spatiotemporal neural architectures tailored for EEG visual decoding. Recent innovations include incorporating brain topological connectivity (Li et al., 2024) and integrating multimodal semantic cues (Zhang et al., 2024). Despite these advancements, most existing methods overlook a fundamental property of the human visual system: its hierarchical nature. We term this issue as Hierarchical Neural Encoding Neglect (HNEN). Rooted in the canonical hierarchical cortical processing pathway, the human visual system processes stimuli through dissociable stages: from low-level edge detection to high-level semantic integration (Felleman & Van Essen, 1991; Riesenhuber & Poggio, 1999). In contrast, existing methods collapse biological hierarchy into flat representation learning, resulting in incomplete visual information capturing.

Substantial neuroscientific evidence (Ben-Yishai et al., 1995; Li et al., 2022) supports a hierarchically organized cortical visual pathway for visual perception: The early visual cortex (V1[1]/V2[2]) (Bridge et al., 2005) specializes in elementary low-level feature extraction (e.g., edge contours), the intermediate ventral stream (V4[3]/IT[4]) (Hansen et al., 2007; Arcaro & Livingstone, 2021) processes object-level semantics, while higher-order association cortex correlates[5] (Grill-Spector & Malach, 2004) integrate high-level contextual scene understanding. This hierarchical visual

decoding enables humans to progressively distill visual information, evolving from coarse contour representations toward fine-grained semantic understanding, through layered cortical interactions. However, existing EEG brain decoding methods adopt a monolithic and flat representation processing paradigm that collapses biological hierarchy. By attempting direct mappings between EEG and complex image embeddings, these approaches suffer from HNEN, failing to distinguish:

- Contour edge saliency (V1/V2 correlates)
- Foreground-object semantics (V4/IT correlates)
- Contextual scene attributes (AC correlates)

The absence of cortically-aligned feature disentanglement fundamentally limits the capacity to localize visual saliency, particularly in preserving contour fidelity.

Our proposed vision-informed EEG (ViEEG) brain decoding framework addresses HNEN through hierarchical visual decoding, as shown in Figure 1. We design three complementary layers: (1) Contour Priming Layer where binary object masks guide EEG feature extraction for edge saliency, (2) Object Purification Layer isolating foreground semantics and removing background, and (3) Contextual Integration Layer, capturing scene understanding from raw images. Three parallel spatiotemporal encoders decode EEG signals following the hierarchical visual comprehension (Contour → Object → Context), progressively integrating features through cross-attention mechanisms. The mask-constrained modules extract edge gradients, the object-centric attention refines core semantics, and the cross-stream fusion integrate contextual cues. Hierarchical CLIP anchoring reinforces biological plausibility: mask embeddings align with contour EEG patterns to preserve shape fidelity, object embeddings couple with foreground semantic features to suppress background interference, and raw image embeddings bind with

---

[1]Primary visual cortex (V1): early processing for basic edge detection and orientation

[2]Secondary visual cortex (V2): integration of local contours and simple shapes

[3]Visual area V4 (V4): processing of shape and color information

[4]Inferotemporal cortex (IT): object recognition and semantic categorization representation

[5]Association cortex (AC): integration of contextual and high-level scene information

contextual dynamics to maintain scene coherence. This structured mimicry of visual processing stages, from low-level edge detection to high-level semantic comprehension, resolves the HNEN and flat representation bottleneck in conventional methods through biologically-grounded feature disentanglement.

Our main contributions can be summarized as follows:

- **A biologically-inspired hierarchical framework** for EEG visual decoding, pioneering tri-stream modeling (Contour → Object → Context). To our knowledge, this is the first work to explicitly enforce feature disentanglement along the visual cortical hierarchy.

- **A novel cross-attention routing mechanism** that integrates multilevel EEG features via bottom-up hierarchical attention, enabling effective contour-object-contextual information fusion.

- **Bidirectional neuro-AI validation**, demonstrating how neuroscientific principles guide model design, while our decoding results support biologically plausible brain representations.

- **New state-of-the-art performance** on THINGS-EEG, achieving 40.9% Top-1 (↑49.82%) in subject-dependent and 22.9% Top-1 (↑45.86%) in subject-independent settings, substantially outperforming previous benchmarks.

## 2. Related Works

Decoding the brain's response to visual stimuli has been a central pursuit in neuroscience and brain-computer interface (BCI) research (Lee et al., 2020; Shi et al., 2024; Liu et al., 2025). Early efforts predominantly focused on fMRI visual decoding (Zafar et al., 2015; Huo et al., 2024), where high spatial resolution inherently captures hierarchical visual processing. For instance, MindEye (Scotti et al., 2023) mapped fMRI to CLIP features via diffusion models, while MindBridge (Wang et al., 2024) developed to align fMRI with images and textual captions. These methods leverage fMRI's spatial precision to distinguish visual features implicitly, even when relying on simplistic alignment.

In contrast, the spatial ambiguity of the EEG centimeter scale forces existing methods (Rakhimberdina et al., 2021; Song et al., 2024; Fu et al., 2025) into flat representation paradigms that confuse hierarchical visual processing. As the dominant brain decoding approach of fMRI and EEG, direct EEG-image alignment fails to model the brain's progressive refinement from edges to semantics, a limitation we term HNEN. While some attempts incorporate attention-based electrode topology (Li et al., 2024) or semantic depth cues (Zhang et al., 2024), these merely address others (e.g.,

*Table 1.* Key symbol definitions. $C$: EEG channels, $T$: timepoints, $H \times W$: image resolution, $D \times D$: reshaped image resolution (512), $d$: embedding dimension (1024).

| Symbol | Dim. | Definition |
|---|---|---|
| $E$ | $R^{C \times T}$ | Raw EEG signals |
| $I$ | $R^{H \times W}$ | Raw image |
| $I_b$ | $R^{D \times D}$ | Binary object mask image |
| $I_f$ | $R^{D \times D}$ | Foreground object image |
| $I_r$ | $R^{D \times D}$ | Raw scene image |
| $F_b, F_f, F_r$ | $R^d$ | Contour/object/context EEG embedding |
| $C_b, C_f, C_r$ | $R^d$ | Contour/object/context CLIP embedding |

spatial noise) rather than HNEN's root cause: EEG's inability to functionally isolate cortical processing stages. This reveals a key gap, fMRI passively benefits from anatomical hierarchy, whereas EEG requires explicit hierarchy modeling. Additionally, UBP (Wu et al., 2025) improves recognition accuracy by blurring target images to simplify CLIP embeddings. While effective for classification, this strategy suppresses fine-grained visual information in EEG representations, fundamentally limiting their support for image reconstruction, a defining objective of brain visual decoding. In contrast, ViEEG enforces hierarchical EEG feature disentanglement via artificial cortical anchors, progressing from contour edges to foreground objects and contextual scenes, enabling object recognition gains while preserving multi-level visual information for faithful reconstruction.

## 3. Method

### 3.1. Preliminaries

The core components of ViEEG are summarized in Table 1, where each symbol is defined based on both computational roles and neuroscientific triple-view alignment.

We formulate EEG visual decoding as a hierarchical zero-shot object recognition, where test categories are disjoint from the training set. Given EEG recordings $E$ with $N$ samples elicited by visual stimuli $\{I_b, I_f, I_r\}$, our objective is to learn hierarchical EEG embeddings $\{F_b, F_f, F_r\}$ that align with the corresponding CLIP embeddings $\{C_b, C_f, C_r\}$ through hierarchical visual decoding. The learning objective for zero-shot object recognition is formulated as:

$$\min_{\theta} \frac{1}{N} \sum_{i=1}^{N} D(F^i, C^i), \tag{1}$$

where $D(\cdot, \cdot)$ measures feature similarity. On the THINGS-EEG dataset, we train with class-disjoint stimuli and test on unseen categories. At inference time, the concatenated EEG embeddings $F_b, F_f, F_r$ are compared to CLIP embeddings using cosine similarity within an InfoNCE-style contrastive learning framework (Radford et al., 2021).

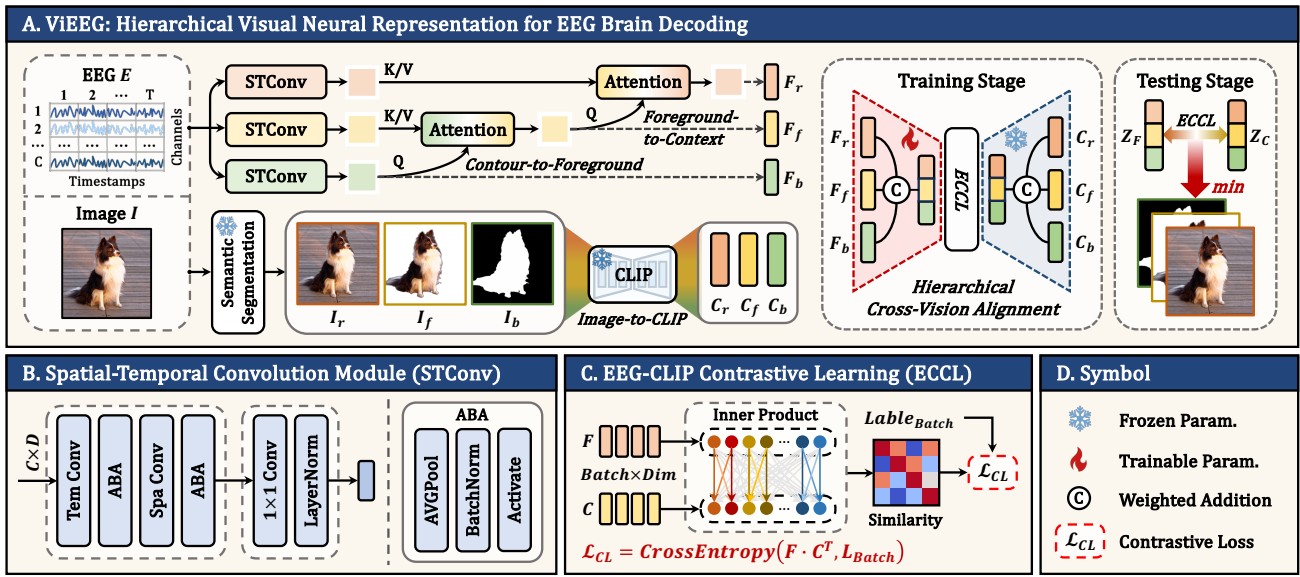

*Figure 3.* Overview of the ViEEG framework. The input image is decomposed into three biologically inspired views: binary mask, foreground object, and raw scene. Corresponding EEG responses are encoded by parallel spatiotemporal encoders, integrated via hierarchical cross-attention, and aligned with CLIP embeddings through hierarchical contrastive learning.

## 3.2. Overall Architecture

ViEEG introduces a biologically inspired hierarchical architecture for EEG visual decoding, simulating layered cortical processing. As illustrated in Figure 3, ViEEG comprises three synergistic components: (1) hierarchical image decomposition, (2) hierarchical EEG encoding, and (3) hierarchical contrastive learning. During initialization, we extract CLIP embeddings from the three decomposed visual representations. In the training phase, EEG features extracted from EEG encoder are aligned with the image CLIP embeddings using contrastive learning. During testing, EEG embeddings extracted from the test set are matched to target samples based on similarity measurements for evaluation.

## 3.3. Hierarchical Image Decomposition

Inspired by the hierarchical structure of the human visual system, we decompose each stimulus image $I$ into three biologically aligned representations via representation segmentation.

**Step 1: Image Processing.** We employ the pre-trained BiRefNet[6] (Zheng et al., 2024), a state-of-the-art high-resolution segmentation model, to generate hierarchical visual representations. The raw image $I$ is first normalized to yield $I_r$, ensuring consistent dimensions and intensity scaling. Saliency detection is then applied, and a binary contour representation is obtained via thresholding:

$$I_b = I\left(\mathcal{F}_{\text{BiRefNet}}(I_r) > \tau\right) \qquad (2)$$

---

[6]https://huggingface.co/ZhengPeng7/BiRefNet

where $\mathcal{F}_{\text{BiRefNet}}$ denotes the pre-trained BiRefNet model, $I(\cdot)$ is the indicator function, and $\tau$ is threshold value. The resulting $I_b$ serves as a binary object mask that highlights salient contours. Next, to extract the foreground object, we perform element-wise multiplication:

$$I_f = I_r \odot I_b \qquad (3)$$

where $\odot$ denotes element-wise multiplication, and $I_f$ is the foreground object image with the background suppressed.

**Step 2: Image CLIP Embedding Processing.** To capture high-quality information from each image representation, we utilize a frozen CLIP-ViT-H/14 to extract embeddings:

$$C_b = \text{CLIP}(I_b), \quad C_f = \text{CLIP}(I_f), \quad C_r = \text{CLIP}(I_r) \qquad (4)$$

where $C_b$, $C_f$, and $C_r$ represent the embeddings corresponding to the binary mask, foreground object, and raw scene image, respectively. These embeddings serve as the foundation for subsequent hierarchical contrastive learning with EEG representations.

## 3.4. Hierarchical EEG Encoding

Our neuro-inspired brain decoding architecture extracts hierarchical (contour & object & context) EEG representations. Given raw EEG $E$, three parallel streams extract cortical hierarchy-aligned EEG representations through spatial-temporal convolution (STConv) and cross-attention hierarchical integration (CAHI).

## 3.5. Spatial-Temporal Convolution

To extract meaningful spatial-temporal representations from EEG signals, we employ a multi-stage convolutional processing pipeline. The first stage applies temporal convolution using a kernel of size $(1, K_t)$ and a stride of $S_t$, capturing temporal dependencies across EEG signals. This is followed by an average pooling operation with a kernel size of $(1, K_p)$ and a stride of $S_p$, which reduces temporal resolution while preserving crucial information. The second stage applies spatial convolution across electrodes using a kernel size of $(C, 1)$, performing spatial filtering. The processed features are then normalized and activated through an ELU function before being projected into the final hierarchical feature representation:

$$
\begin{aligned}
F^{(1)} &= \text{ELU}(\text{BatchNorm}(\text{AvgPool}(\text{Conv2D}_t(E)))) \\
F^{(2)} &= \text{ELU}(\text{BatchNorm}(\text{Conv2D}_s(F^{(1)}))) \\
F^0 &= \text{Conv2D}_{proj}(F^{(2)})
\end{aligned} \tag{5}
$$

where $\text{Conv2D}_t(\cdot)$ is temporal convolution, $\text{Conv2D}_s(\cdot)$ represents spatial convolution, and $\text{Conv2D}_{proj}(\cdot)$ is a $1 \times 1$ convolution. These hierarchical spatial-temporal features facilitate robust representation learning, which is further refined through the cross-attention mechanism. After three parallel STConv operations, EEG signals are decoupled into three features: $F_b^0$, $F_f^0$, and $F_r^0$.

## 3.6. Cross-Attention Hierarchical Integration

The CAHI integrates biologically inspired features by sequentially combining contour-to-object and object-to-context information, where contour cues refine object representations and object cues enrich contextual understanding. Assuming that low-level representation is denoted by $a$ and higher-level representation by $b$, we perform bottom-up integration, and transfer information from $a$ to $b$. We use $a$ and apply several linear transformations to generate query embeddings in attention, while generate key and value embeddings from $b$ as:

$$
K^i = b^i W^K, \quad Q^i = a^i W^Q, \quad V^i = b^i W^V \tag{6}
$$

where $\{a, b\}^i$ represents the $i$-th attention head, and $W^Q$, $W^K$, and $W^V$ are the parameter matrices for $Q^i$, $K^i$, and $V^i$, respectively. We determine the attention coefficients by computing the scaled dot-product between $Q$ and $K$, followed by a softmax to obtain the attention weights. The final feature for each head is computed as:

$$
F^i = \text{Attention}(Q^i, K^i, V^i) = \text{softmax}\left(\frac{Q^i K^{iT}}{\sqrt{d}}\right) V^i \tag{7}
$$

where $d$ is the normalization hyperparameter. The output of the multi-head self-attention is then given by:

$$
F = \text{MHA}(Q = a, K = V = b) = (F^1 \| F^2 \| \dots \| F^h) W^O + b \tag{8}
$$

where $\text{MHA}(\cdot)$ denote multi-head attention function, $h$ is the number of attention heads, $W^O$ is a linear transformation, and $\|$ denotes concatenation.

**Contour-to-Object Integration:** The low-level and high-level representation is contour feature $F_b^0$ and object feature $F_f^0$, respectively, and refined object feature is computed as:

$$
F_f = \text{LN}\left(F_f^0 + \text{Dropout}\left(\text{MHA}(Q = F_b, K = V = F_f^0)\right)\right) \tag{9}
$$

where $\text{LN}(\cdot)$ is the LayerNorm function, $F_b$ denotes $F_b^0$, and $F_f$ is refined object feature.

**Object-to-Context Integration:** The low-level and high-level representation is object feature $F_f$ and context feature $F_r^0$, and refined context feature is computed as:

$$
F_r = \text{LN}\left(F_r^0 + \text{Dropout}\left(\text{MHA}(Q = F_f, K = V = F_r^0)\right)\right) \tag{10}
$$

where $F_r$ denotes refined context feature.

## 3.7. Hierarchical Contrastive Learning

To further enhance representation learning, we adopt a contrastive learning framework that aligns the concatenated EEG features with their corresponding image embeddings. Specifically, we first concatenate the hierarchical EEG features, i.e., $Z_F = (F_b \| F_f \| F_r)$, and similarly, the image features are concatenated as $Z_C = (C_b \| C_f \| C_r)$. We then compute the cosine similarity between these concatenated features, scaled by a learnable temperature parameter $\alpha$. The final contrastive loss $\mathcal{L}_{CL}$ is computed as the cross entropy loss on the resulting similarity logits:

$$
\mathcal{L}_{CL} = \text{CrossEntropy}\left(\alpha \cdot \cos(Z_F, Z_C), Y\right) \tag{11}
$$

where $cos(\cdot, \cdot)$ is cosine similarity function, and $Y$ denotes the ground-truth labels. This formulation encourages the EEG features to closely align with the corresponding image features in the shared embedding space.

# 4. Experiment

## 4.1. Dataset

The THINGS-EEG dataset (Gifford et al., 2022) is a large-scale benchmark and the most widely used in zero-shot EEG object recognition, comprising brain responses from ten participants exposed to images from the THINGS database under an RSVP paradigm. The dataset comprises 1654 training concepts and 200 zero-shot test concepts, yielding a total of 82,160 trials per subject. EEG signals were acquired using a 64-channel cap, band-pass filtered between 0.1–100 Hz, sample rate of 1000 Hz, and downsampled to 250 Hz. Preprocessing involved segmenting epochs from 200 ms before to 800 ms after stimulus onset, followed by baseline

*Table 2.* Overall accuracy (%) comparison: Top-1 and Top-5 in 200-way zero-shot object recognition.

| Methods | Subject 1 | | Subject 2 | | Subject 3 | | Subject 4 | | Subject 5 | | Subject 6 | | Subject 7 | | Subject 8 | | Subject 9 | | Subject 10 | | Ave | |
|---|---|---|---|---|---|---|---|---|---|---|---|---|---|---|---|---|---|---|---|---|---|---|
| | top-1 | top-5 | top-1 | top-5 | top-1 | top-5 | top-1 | top-5 | top-1 | top-5 | top-1 | top-5 | top-1 | top-5 | top-1 | top-5 | top-1 | top-5 | top-1 | top-5 | top-1 | top-5 |
| Subject dependent - train and test on one subject | | | | | | | | | | | | | | | | | | | | | | |
| ConvNet (Schirrmeister et al., 2017) | 14.9 | 39.3 | 18.9 | 42.1 | 17.8 | 49.0 | 23.9 | 55.8 | 12.2 | 32.9 | 20.1 | 46.5 | 15.5 | 42.6 | 20.7 | 48.9 | 20.7 | 49.8 | 19.3 | 47.5 | 18.4 | 45.5 |
| EEGNet (Lawhern et al., 2018) | 16.0 | 42.9 | 17.9 | 48.6 | 18.2 | 51.5 | 23.9 | 59.0 | 14.4 | 37.7 | 19.5 | 52.0 | 18.5 | 50.2 | 30.2 | 61.2 | 23.3 | 51.2 | 22.5 | 58.3 | 20.4 | 51.2 |
| DGCNN [TAFFC'18] (Song et al., 2018) | 12.3 | 36.6 | 11.5 | 39.5 | 15.7 | 43.8 | 19.7 | 50.6 | 10.6 | 32.6 | 15.4 | 46.6 | 14.0 | 43.2 | 25.1 | 54.5 | 16.0 | 43.7 | 17.9 | 53.0 | 15.8 | 44.4 |
| EEG-Conformer [TNSRE'22] (Song et al., 2022) | 11.4 | 32.4 | 15.2 | 41.9 | 19.8 | 50.9 | 23.0 | 56.6 | 13.6 | 33.4 | 18.1 | 49.0 | 18.5 | 48.2 | 27.1 | 56.9 | 15.2 | 40.0 | 22.6 | 57.8 | 18.5 | 46.7 |
| EEG-ChannelNet [TPAMI'20] (Palazzo et al., 2020) | 18.0 | 45.9 | 19.1 | 47.9 | 22.7 | 53.8 | 24.9 | 57.0 | 15.6 | 39.5 | 22.3 | 52.4 | 20.5 | 52.2 | 31.2 | 60.5 | 21.5 | 51.3 | 24.3 | 57.7 | 22.0 | 51.8 |
| Mb2C [ACM MM'24] (Wei et al., 2024) | 23.6 | 56.3 | 22.6 | 50.5 | 26.3 | 60.1 | 34.8 | 67.0 | 21.3 | 53.0 | 31.0 | 62.3 | 25.0 | 54.8 | 39.0 | 69.3 | 27.5 | 59.3 | 33.1 | 70.8 | 28.4 | 60.3 |
| NICE [ICLR'24] (Song et al., 2024) | 21.7 | 51.2 | 23.3 | 55.0 | 29.1 | 60.5 | 32.3 | 69.6 | 18.2 | 45.6 | 29.3 | 62.1 | 24.3 | 59.2 | 41.3 | 72.4 | 24.3 | 59.0 | 28.9 | 62.6 | 27.3 | 59.7 |
| ATM [NeurIPS'24] (Li et al., 2024) | 25.6 | 60.5 | 22.0 | 54.5 | 25.0 | 62.4 | 31.4 | 60.9 | 12.9 | 43.0 | 21.3 | 51.1 | 30.5 | 61.5 | 38.8 | 72.0 | 24.4 | 51.5 | 29.1 | 63.5 | 26.1 | 58.1 |
| CognitionCapturer [AAAI'25] (Zhang et al., 2024) | 27.2 | 59.5 | 28.7 | 56.9 | 37.1 | 66.1 | 37.6 | 63.2 | 21.8 | 47.7 | 31.5 | 58.0 | 32.8 | 59.5 | 47.6 | 73.5 | 33.3 | 57.6 | 35.0 | 63.5 | 33.3 | 60.5 |
| BrainFLORAr [ACM MM'25] (Li et al., 2025) | 23.6 | 53.1 | 24.7 | 58.4 | 31.5 | 64.3 | 33.4 | 70.1 | 20.3 | 48.9 | 29.0 | 64.2 | 25.9 | 61.6 | 41.5 | 73.2 | 29.5 | 61.3 | 31.2 | 65.0 | 29.1 | 62.0 |
| SRT [ICCV'25] (Kim et al., 2025) | 25.9 | 57.7 | 25.2 | 60.8 | 30.4 | 62.8 | 33.5 | 66.8 | 22.8 | 52.0 | 24.1 | 61.6 | 29.8 | 63.9 | 40.1 | 70.2 | 31.5 | 64.4 | 28.7 | 65.7 | 29.2 | 62.6 |
| **ViEEG [Ours]** | **34.1** | **71.3** | **38.4** | **67.9** | **40.6** | **74.7** | **50.1** | **80.8** | **28.9** | **61.5** | **44.3** | **76.5** | **38.6** | **75.2** | **54.0** | **82.5** | **37.3** | **74.9** | **42.8** | **79.8** | **40.9** | **74.5** |
| Subject independent - leave one subject out for test | | | | | | | | | | | | | | | | | | | | | | |
| ConvNet (Schirrmeister et al., 2017) | 10.4 | 31.5 | 14.2 | 35.0 | 8.7 | 26.9 | 12.1 | 31.4 | 5.7 | 22.5 | 10.2 | 28.0 | 8.3 | 22.5 | 10.0 | 29.9 | 6.6 | 19.8 | 12.2 | 34.0 | 9.8 | 28.2 |
| EEGNet (Lawhern et al., 2018) | 11.0 | 31.1 | 11.4 | 34.0 | 6.5 | 24.7 | 13.4 | 34.4 | 6.7 | 26.7 | 8.5 | 29.4 | 7.9 | 22.0 | 10.7 | 32.9 | 8.9 | 27.7 | 15.1 | 41.6 | 10.0 | 30.4 |
| DGCNN [TAFFC'18] (Song et al., 2018) | 10.8 | 30.6 | 11.9 | 31.3 | 6.3 | 21.3 | 8.2 | 24.6 | 6.1 | 18.5 | 11.2 | 30.4 | 6.8 | 20.5 | 11.0 | 28.6 | 9.4 | 25.4 | 12.2 | 30.6 | 9.4 | 26.2 |
| EEG-Conformer [TNSRE'22] (Song et al., 2022) | 6.3 | 22.3 | 5.7 | 20.4 | 5.8 | 15.7 | 7.8 | 21.8 | 6.7 | 18.4 | 10.4 | 32.9 | 7.0 | 24.1 | 9.1 | 25.2 | 5.0 | 17.2 | 11.7 | 33.2 | 7.5 | 23.1 |
| EEG-ChannelNet [TPAMI'20] (Palazzo et al., 2020) | 11.2 | 33.6 | 12.8 | 34.4 | 8.4 | 26.5 | 13.5 | 33.8 | 9.7 | 29.0 | 10.8 | 33.9 | 9.7 | 27.7 | 11.6 | 32.3 | 10.4 | 30.8 | 15.7 | 39.5 | 11.4 | 32.2 |
| Mb2C [ACM MM'24] (Wei et al., 2024) | 10.5 | 28.1 | 11.3 | 32.8 | 8.8 | 27.6 | 13.6 | 33.5 | 10.6 | 27.5 | 12.1 | 33.1 | 11.5 | 31.8 | 12.0 | 32.1 | 12.1 | 31.3 | 16.1 | 42.1 | 11.9 | 32.0 |
| NICE [ICLR'24] (Song et al., 2024) | 13.5 | 39.3 | 16.7 | 42.7 | 12.4 | 35.5 | 17.9 | 41.9 | 14.7 | 38.1 | 15.6 | 44.6 | 13.6 | 39.0 | 13.5 | 37.2 | 17.0 | 42.0 | 22.8 | 50.7 | 15.7 | 41.1 |
| ATM [NeurIPS'24] (Li et al., 2024) | 17.1 | 41.8 | 20.2 | 44.2 | 13.2 | 36.7 | 17.0 | 40.7 | 15.1 | 41.0 | 13.5 | 38.3 | 10.1 | 29.0 | 15.2 | 41.9 | 13.5 | 38.4 | 20.0 | 45.4 | 15.5 | 39.6 |
| CognitionCapturer [AAAI'25] (Zhang et al., 2024) | 16.3 | 42.3 | 16.2 | 37.9 | 8.8 | 26.8 | 15.4 | 37.6 | 10.1 | 31.7 | 14.0 | 35.4 | 10.7 | 26.9 | 13.9 | 34.2 | 9.0 | 32.4 | 15.3 | 38.6 | 13.0 | 34.4 |
| BrainFLORAr [ACM MM'25] (Li et al., 2025) | 13.5 | 42.0 | 18.5 | 44.3 | 13.4 | 38.6 | 18.1 | 41.0 | 15.5 | 39.4 | 15.4 | 42.8 | 12.4 | 35.2 | 14.5 | 38.9 | 13.5 | 40.1 | 20.2 | 49.2 | 15.5 | 41.2 |
| SRT [ICCV'25] (Kim et al., 2025) | 17.3 | 39.8 | 19.4 | 42.4 | 14.5 | 38.8 | 15.8 | 40.2 | 15.9 | 38.1 | 17.1 | 41.4 | 11.7 | 31.1 | 14.9 | 37.0 | 13.4 | 38.3 | 23.5 | 54.6 | 16.4 | 40.2 |
| **ViEEG [Ours]** | **22.7** | **53.5** | **24.7** | **52.5** | **19.0** | **48.4** | **25.5** | **54.1** | **19.8** | **47.3** | **20.7** | **49.3** | **20.9** | **49.4** | **20.8** | **46.8** | **23.8** | **52.7** | **31.2** | **60.3** | **22.9** | **51.4** |

*Table 3.* Summary of recognition performance and relative improvement over NICE across Top-1 and Top-5 metrics.

| Metric | ViEEG | NICE | Improve | Best Case (Sub.10) |
|---|---|---|---|---|
| Avg. Top-1 (Dep.) | 40.9% | 27.3% | 49.82% | 42.8% vs 28.9% (↑ 48.1%) |
| Avg. Top-5 (Dep.) | 74.5% | 59.7% | 24.79% | 79.8% vs 62.6% (↑ 27.5%) |
| Avg. Top-1 (Ind.) | 22.9% | 15.7% | 45.86% | 31.2% vs 22.8% (↑ 36.8%) |
| Avg. Top-5 (Ind.) | 51.4% | 41.1% | 25.06% | 60.3% vs 50.7% (↑ 19.0%) |

correction using the pre-stimulus interval. Additionally, we evaluate ViEEG on THINGS-MEG dataset (Hebart et al., 2023) in Appendix C, to validate the universality of ViEEG across various neural signals. More details of datasets are in Appendix A.

### 4.2. Experimental Details

All experiments were implemented in PyTorch and conducted on an NVIDIA RTX 3090 (4-GPU) environment. The key hyperparameters include a mask threshold $\tau = 0.5$, temporal convolution kernel size $1 \times 25$, spatial convolution kernel $63 \times 1$, average pooling kernels $1 \times 51$ and $1 \times 5$, a dropout rate of 0.5, attention layer depth of 1, and 3 attention heads. The model is optimized using Adam with a batch size of 1000 and a learning rate of $2 \times 10^{-3}$.

For fair comparison, experiments were conducted follow the protocol of NICE. All methods used the same CLIP backbone (CLIP-ViT-H/14[7]) to ensure alignment consistency.

[7] https://huggingface.co/laion/CLIP-ViT-H-14-laion2B-s32B-b79K

For subject-dependent training, 740 trials were randomly selected as the validation set, and the model with the lowest validation loss was retained. For subject-independent training, we employed a leave-one-subject-out (LOSO) protocol, using 6660 trials for validation in each fold. All experiments were repeated five times, and the average test accuracy was reported to mitigate variance and ensure result robustness. The computational requirements against recognition performance is provided in Appendix F.

### 4.3. Overall Performance

The experimental comparison against state-of-the-art methods demonstrates ViEEG's superior decoding capability across both evaluation paradigms. We conduct comprehensive comparisons against several EEG decoding methods, including classical EEG feature extraction methods such as ConvNet (Schirrmeister et al., 2017), EEGNet (Lawhern et al., 2018), DGCNN (Song et al., 2018), EEG-Conformer (Song et al., 2022), and EEG-ChannelNet (Palazzo et al., 2020), as well as recent brain decoding approaches such as MB2C (Wei et al., 2024), NICE (Song et al., 2024), ATM (Li et al., 2024), CognitionCapturer (Zhang et al., 2024), BrainFLORA (Li et al., 2025), and SRT (Kim et al., 2025), with more details in Appendix B. Table 2 summarizes the 200-way zero-shot image retrieval performance across both subject-dependent and subject-independent experimental paradigms, where ViEEG consistently achieves state-of-the-art performance through hierarchical visual modeling.

*Table 4.* Ablation study: Top-1 and Top-5 accuracy (%) in 200-way zero-shot object recognition.

| Methods | Subject 1 | | Subject 2 | | Subject 3 | | Subject 4 | | Subject 5 | | Subject 6 | | Subject 7 | | Subject 8 | | Subject 9 | | Subject 10 | | Ave | |
|---|---|---|---|---|---|---|---|---|---|---|---|---|---|---|---|---|---|---|---|---|---|---|
| | top-1 | top-5 | top-1 | top-5 | top-1 | top-5 | top-1 | top-5 | top-1 | top-5 | top-1 | top-5 | top-1 | top-5 | top-1 | top-5 | top-1 | top-5 | top-1 | top-5 | top-1 | top-5 |
| *Subject dependent - train and test on one subject* | | | | | | | | | | | | | | | | | | | | | | |
| $F_b$ Feature Only | 14.2 | 41.8 | 14.0 | 39.8 | 17.0 | 39.1 | 20.9 | 53.2 | 8.6 | 30.3 | 13.6 | 40.9 | 14.7 | 40.7 | 17.6 | 46.4 | 12.8 | 37.6 | 14.8 | 44.0 | 14.8 | 41.4 |
| $F_f$ Feature Only | 22.2 | 52.1 | 24.5 | 56.1 | 29.9 | 66.0 | 34.0 | 67.1 | 19.2 | 50.0 | 32.5 | 63.5 | 26.5 | 58.7 | 38.7 | 73.9 | 29.3 | 61.8 | 31.8 | 68.5 | 28.9 | 61.8 |
| $F_r$ Feature Only | 22.1 | 52.6 | 24.8 | 58.0 | 32.4 | 62.7 | 33.1 | 67.4 | 18.5 | 49.1 | 29.7 | 66.2 | 26.8 | 60.1 | 40.6 | 74.1 | 27.4 | 62.5 | 31.8 | 68.6 | 28.6 | 62.1 |
| w/o Cross-Attention | 32.4 | 66.8 | 34.6 | 65.9 | 39.1 | 71.3 | 46.5 | 80.2 | 27.2 | 59.8 | 40.5 | 74.3 | 35.3 | 71.3 | 52.2 | 80.8 | 34.6 | 71.9 | 43.9 | 75.5 | 38.6 | 71.8 |
| **ViEEG** | **34.1** | **71.3** | **38.4** | **67.9** | **40.6** | **74.7** | **50.1** | **80.8** | **28.9** | **61.5** | **44.3** | **76.5** | **38.6** | **75.2** | **54.0** | **82.5** | **37.3** | **74.9** | **42.8** | **79.8** | **40.9** | **74.5** |
| *Subject independent - leave one subject out for test* | | | | | | | | | | | | | | | | | | | | | | |
| $F_b$ Feature Only | 5.9 | 18.6 | 7.9 | 22.7 | 5.7 | 19.1 | 7.4 | 26.3 | 7.6 | 20.5 | 4.2 | 18.6 | 9.0 | 23.1 | 7.9 | 21.8 | 7.0 | 20.8 | 10.2 | 30.9 | 7.3 | 22.3 |
| $F_f$ Feature Only | 15.8 | 43.0 | 19.8 | 42.5 | 13.3 | 40.3 | 20.5 | 45.1 | 13.7 | 37.1 | 17.1 | 41.7 | 13.2 | 39.7 | 15.5 | 41.9 | 15.3 | 42.9 | 23.4 | 51.4 | 16.8 | 42.6 |
| $F_r$ Feature Only | 14.9 | 41.5 | 18.4 | 45.9 | 14.1 | 41.3 | 18.1 | 42.2 | 15.9 | 38.3 | 18.5 | 42.8 | 15.4 | 41.5 | 16.0 | 41.6 | 18.3 | 45.3 | 20.5 | 50.2 | 17.0 | 43.1 |
| w/o Cross-Attention | 19.1 | 52.1 | 22.0 | 48.9 | 15.1 | 45.1 | 24.2 | **55.2** | 16.7 | 41.6 | **21.0** | **50.4** | 17.2 | 44.7 | 19.5 | 45.6 | 20.9 | 48.8 | 31.1 | 59.8 | 20.7 | 49.2 |
| **ViEEG** | **22.7** | **53.5** | **24.7** | **52.5** | **19.0** | **48.4** | **25.5** | 54.1 | **19.8** | **47.3** | 20.7 | 49.3 | **20.9** | **49.4** | **20.8** | **46.8** | **23.8** | **52.7** | **31.2** | **60.3** | **22.9** | **51.4** |

**Subject-Dependent Experiment.** ViEEG achieves new performance benchmarks, reaching 40.9% Top-1 and 74.5% Top-5 average accuracy across 10 subjects, outperforming the previous SOTA method by substantial margins. Specifically, compared to the baseline NICE, ViEEG demonstrates an average improvement of 49.82% in Top-1 accuracy and 24.79% in Top-5 accuracy, reflecting its capacity to effectively exploit subject-specific neural patterns. This performance highlights ViEEG's capacity to handle visual scenes through hierarchical feature disentanglement.

**Subject-Independent Experiment.** When tested on unseen subjects under LOSO protocol, ViEEG maintains robust performance with 22.9% Top-1 and 51.4% Top-5 accuracy. The biological plausibility of hierarchical design enables better generalization compared to flat representation. For average object recognition accuracy, ViEEG demonstrates relative improvement of 45.86% in Top-1 accuracy and 25.06% in Top-5 accuracy. This performance gap stems from ViEEG's hierarchical feature disentanglement captures invariant representations across individuals.

Consistent performance gains across all evaluation metrics (Table 3) statistically validate modeling of visual processing hierarchy substantially improves EEG decoding accuracy.

Additionally, we evaluate ViEEG on the THINGS-MEG dataset (Hebart et al., 2023) in Appendix C to validate its applicability beyond EEG signals. Results show consistent performance gains over prior methods, further supporting the generalizability of ViEEG across neural modalities.

### 4.4. Ablation Study

We conducted ablation study to assess the contributions of each core module in ViEEG, as summarized in Table 4.

#### 4.4.1. ABLATION ON HIERARCHICAL EMBEDDING

This experiment analyzes the effectiveness of different levels of hierarchical EEG embeddings. Specifically, we evaluated

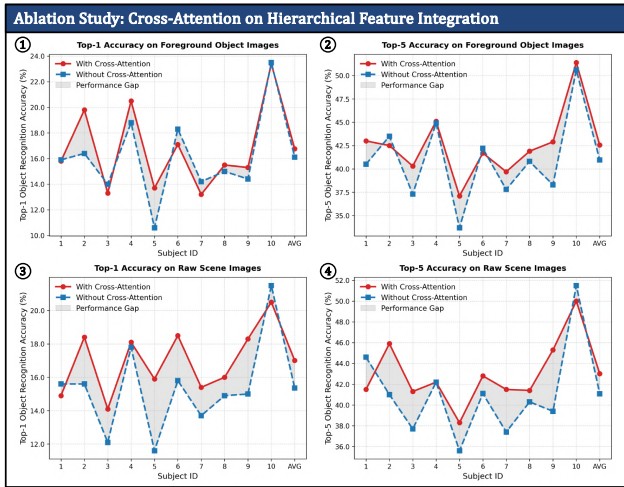

*Figure 4.* Effect of cross-attention on feature integration. Sub-figures 1-2: Top-1 and Top-5 accuracy of object embedding ($F_f$) with/without cross-attention module; Subfigures 3-4: same metrics for context embedding ($F_r$).

configurations that utilized only one type of embedding at a time: $F_b$ (contour-level), $F_f$ (object-level), and $F_r$ (scene-level). In the subject-dependent setting, $F_b$ achieved 14.2% Top-1 and 41.8% Top-5 accuracy. In contrast, $F_f$ and $F_r$ both improved performance to approximately 22% Top-1 accuracy. These results suggest that while each representation captures distinct neural elements, none is sufficient alone to support optimal decoding performance. Notably, $F_f$ and $F_r$ embeddings, which encode mid-level and high-level visual semantics, outperformed $F_b$, highlighting the importance of semantic and contextual elements in EEG brain decoding.

#### 4.4.2. ABLATION ON CROSS ATTENTION

We further examined the role of cross-attention mechanism in integrating multi-level EEG features across the visual hierarchy. Ablating the cross-attention led to noticeable performance degradation. For instance, in the subject-dependent setting, the Top-1 and Top-5 accuracies dropped from 40.9%

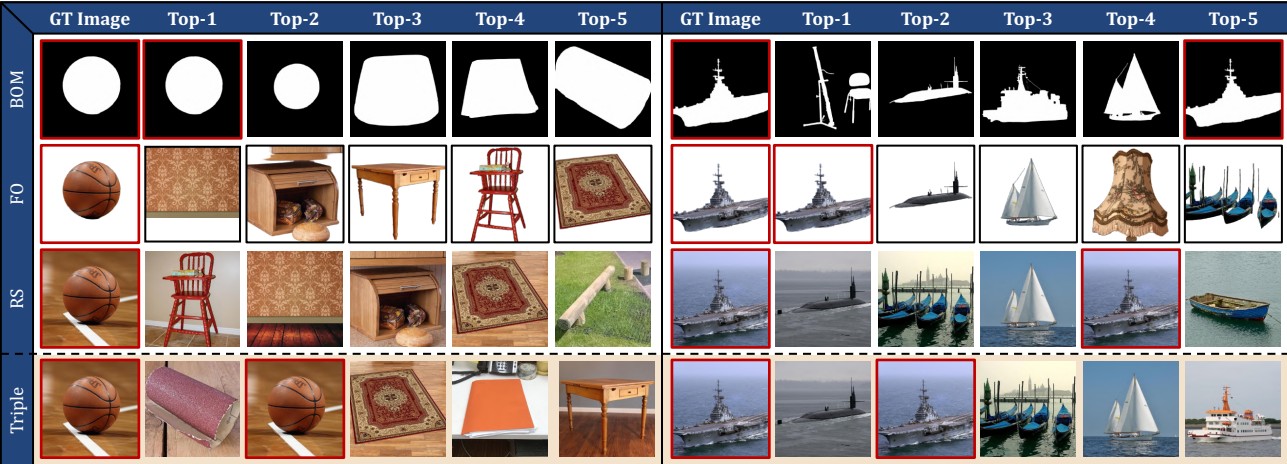

*Figure 5.* Zero-shot image retrieval with different embeddings (Subject 8). Left: basketball example shows BOM excels due to clean contours. Right: aircraft carrier case favors FO due to complex shape and background blending.

and 74.5% to 38.6% and 71.8%, respectively, when cross-attention was removed. In the subject-independent setting, similar trends were observed, with the full ViEEG outperforming the variant without cross-attention by notable margins. These findings underscore the importance of cross-attention for aligning and integrating hierarchical EEG embeddings, thereby enhancing overall model robustness and generalization across subjects. The performance gain is particularly evident in the integration of $F_f$ and $F_r$, as visualized in Figure 4, demonstrating that adding cross-attention for hierarchical visual information integration effectively improves EEG decoding of the integrated features.

More details of ablation study are provided in Appendix D.

## 5. Discussion

### 5.1. Representational Analysis

To assess the representational consistency of ViEEG, we visualize the representational similarity matrices (RSMs) as heatmaps using Subject 8 from the THINGS-EEG dataset. RSMs reflect pairwise similarities among neural embeddings across visual categories, providing insights into how well each model captures the intrinsic EEG structure. For comparison, RSMs from NICE and ATM are also shown in Figure 6, with additional subjects in Appendix Section E.

All three models exhibit block-diagonal patterns when test samples are ordered by category, indicating basic class-level separation. However, ViEEG shows noticeably sharper diagonals within each block, reflecting stronger intra-class consistency and more accurate instance-level alignment. For example, in the "Food" category, ViEEG maintains focused diagonal peaks, whereas NICE and ATM produce more diffuse patterns. This suggests ViEEG achieves finer discrimination within categories. In less structured or ambiguous

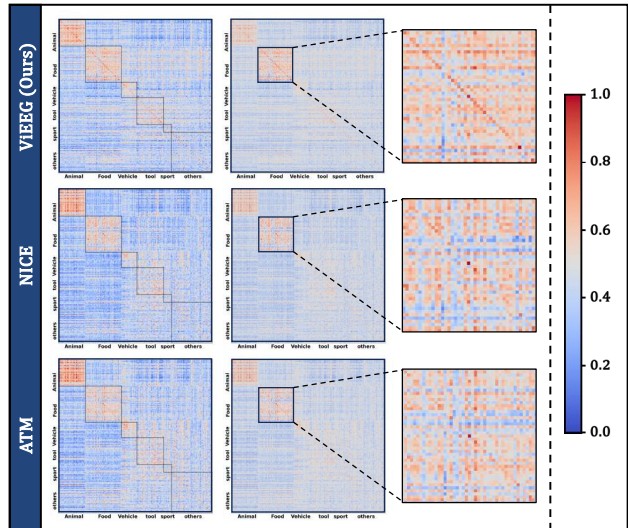

*Figure 6.* Representational similarity matrices (RSM) of ViEEG and baselines across categories (Animal, Food, Vehicle, Tool, Sports, and Others), and zoomed-in view of Food category.

classes such as "Tool", "Sports", and "Other", all models show reduced separability, though ViEEG retains relatively clearer patterns. Overall, ViEEG better preserves both category structure and exemplar specificity, which is crucial for downstream EEG retrieval and classification tasks.

### 5.2. Zero-shot Image Retrieval

We further evaluate ViEEG's decoding ability through zero-shot image retrieval using different visual embeddings: binary object mask (BOM), foreground object (FO), raw scene (RS), and their concatenation (Triple). As shown in Figure 5, each embedding performs differently based on the visual characteristics of the object. For instance, BOM performs

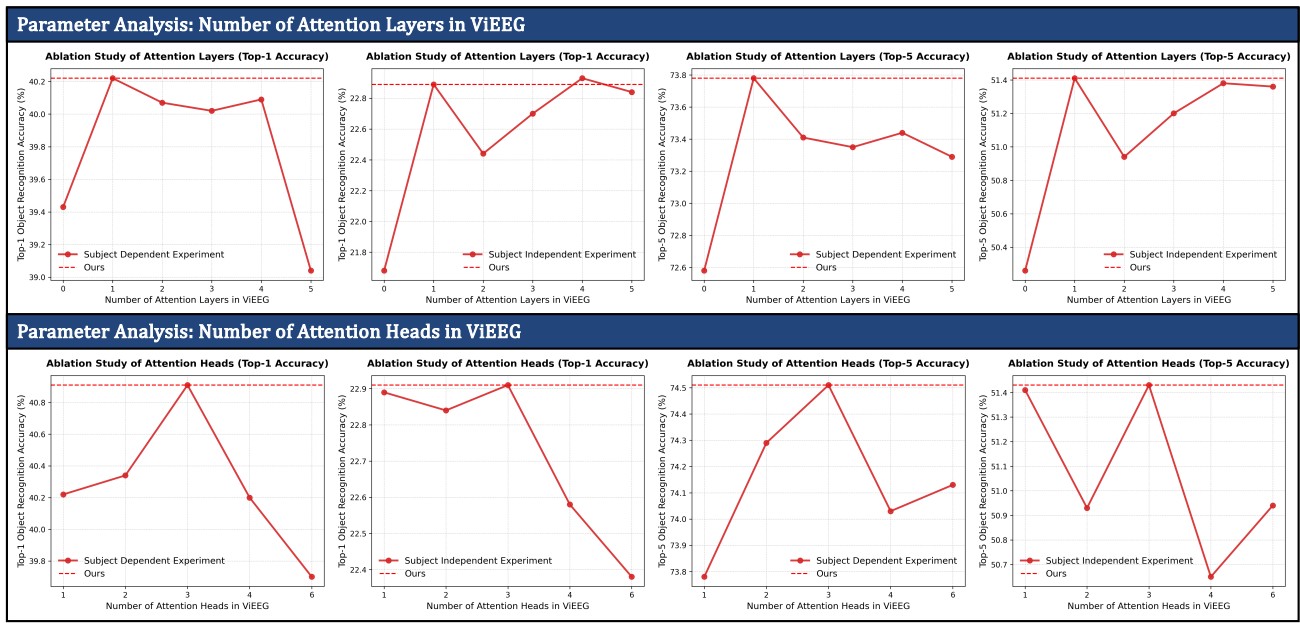

*Figure 7.* Parameter analysis for the attention module. Top row: Accuracy trends for varying numbers of attention layers; Bottom row: Accuracy trends for different numbers of attention heads. Both Top-1 and Top-5 metrics under subject-dependent and subject-independent settings are reported.

best on simple silhouettes like basketballs, while FO excels for complex scenes like aircraft carriers where background blending challenges other embeddings. Notably, the triple fusion consistently achieves the highest accuracy, leveraging complementary information from all views. This fusion mitigates the weaknesses of individual embeddings and enhances retrieval robustness. More examples of image retrieval task are provided in Appendix G.

### 5.3. Image Reconstruction

To further assess the visual fidelity of ViEEG's representations, we perform image reconstruction using visual embeddings derived from EEG. As shown in Figure 2, the reconstructed images preserve key semantic information of the original stimuli, including object category, shape, and spatial structure. Notably, ViEEG is capable of reconstructing the foreground object and contour, demonstrating its capacity to capture fine-grained visual features from brain signals. More examples of image reconstruction task are provided in Appendix H.

### 5.4. Parameter Analysis

ViEEG integrates a cross-attention to hierarchically fuse contour, object, and context features. We perform grid search experiments in Figure 7 to explore the impact of attention layers and heads. Varying the number of attention layers from 0 to 5, we find that a one-layer configuration yields the best Top-1 accuracy in both settings. Deeper attention stacks

lead to overfitting or marginal gains, making one layer the optimal balance between performance and complexity. For attention heads, we test 1 to 6 heads. Three heads achieve the highest accuracy in the subject-dependent case, while subject-independent performance remains relatively stable across configurations. Overall, ViEEG performs best with one attention layer and three heads. More details of parameter analysis are provided in Appendix D.

## 6. Conclusion

In this work, we propose ViEEG, a novel EEG visual decoding framework that emulates the hierarchical structure of human visual perception. By integrating biologically motivated image decomposition with hierarchical EEG encoding and cross-attention fusion, ViEEG captures the progressive flow of visual information from edge detection to semantic and contextual understanding. This design directly addresses the critical issue of hierarchical neural encoded neglect in conventional EEG decoding approaches. Through extensive experiments on THINGS-EEG and THINGS-MEG datasets, ViEEG significantly outperformed state-of-the-art baselines in both subject-dependent and subject-independent zero-shot image retrieval, and image reconstruction tasks. Beyond performance gains, ViEEG offers a cognitively plausible model that strengthens the connection between neuroscience and artificial intelligence. We believe our work advances neuro-inspired learning toward more robust, interpretable, and generalizable brain-computer interface systems.

## Acknowledgements

This work was supported by National Natural Science Foundation of China (Nos. 62472220, 62371234, 62076129), Jiangsu Province 100 Foreign Experts Introduction Plan (BX2022012), Natural Science Foundation of Jiangsu Province (No. BK20231438), Key Research and Development Plan of Jiangsu Province (No. BE2022842), and the Basic Research Program of the Bureau of Science and Technology (ILF24001).

## Impact Statement

This work aims to advance machine learning methods for non-invasive brain signal decoding by introducing a biologically inspired hierarchical framework for EEG-based visual reconstruction. By improving the interpretability and fidelity of brain–vision alignment, our approach has the potential to benefit fundamental neuroscience research, brain–computer interfaces, and assistive technologies for individuals with sensory or communication impairments.

The proposed method is evaluated exclusively on publicly available datasets and does not involve real-world deployment or personal data collection. We do not foresee immediate negative societal impacts arising from this work. Overall, this study contributes to the understanding of brain-inspired representation learning and is intended to support positive, responsible applications of machine learning in neuroscience and healthcare.

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

# A. Details of the Datasets

## A.1. THINGS-EEG Dataset

We conducted experiments on the large-scale THINGS-EEG dataset (Gifford et al., 2022), designed to support neural decoding of visual object recognition. The dataset contains EEG recordings from ten healthy adult participants (mean age: 28.5 years; 8 female, 2 male), all with normal or corrected-to-normal vision. EEG signals were acquired using a 64-channel (see Figure 8) cap arranged according to the 10–10 international system and sampled at 1000 Hz, with online band-pass filtering between 0.1–100 Hz and referencing to Fz.

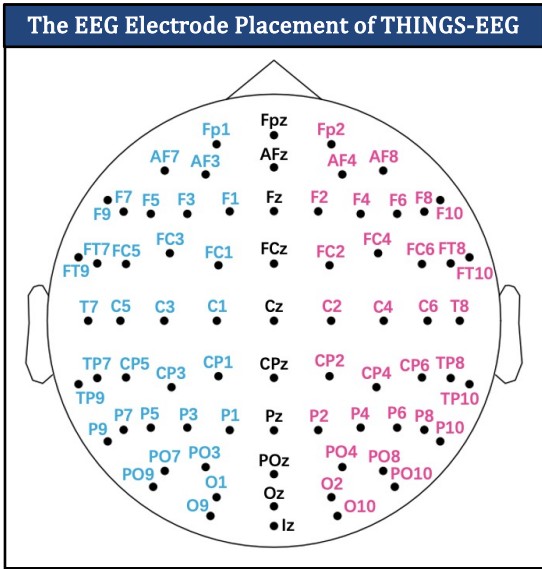

*Figure 8.* The corresponding EEG electrode placement of THINGS-EEG dataset.

Visual stimuli were selected from the THINGS image database (Hebart et al., 2019), which includes naturalistic object images spanning 1854 distinct concepts grouped into 27 higher-level categories. The THINGS image dataset comprises six major categories, including animals (e.g., cats, dogs), vehicles (e.g., airplanes, ships), food items (e.g., cake, corn), tools (e.g., cameras, phones), sports equipment (e.g., balls, golf clubs), and other miscellaneous categories. For model training and evaluation, these concepts were split into 1654 training and 200 testing categories. Each training concept was associated with ten images, while each test concept had one image. The experiment followed a rapid serial visual presentation (RSVP) paradigm (Thorpe et al., 1996), where participants viewed 20 images per trial, each presented for 100 ms followed by a 100 ms blank screen (i.e., 200 ms SOA). An orthogonal target detection task was used to maintain attention.

Each participant completed four sessions, resulting in a total of 82,160 trials per subject, including 16,540 training image trials (each repeated four times) and 200 test images (each repeated 80 times). Trials containing target stimuli were excluded. EEG data were segmented from 200 ms before to 800 ms after stimulus onset, followed by baseline correction and downsampling to 250 Hz. All 64 channels were retained, and signals were averaged across repeated presentations of the same image to enhance signal-to-noise ratio. Following prior work (Song et al., 2024), multivariate noise normalization was applied to the training set. Input images were resized to 224×224 pixels and normalized before being fed into the visual encoder.

## A.2. THINGS-MEG dataset

We additionally evaluated our model on the THINGS-MEG dataset (Hebart et al., 2023), which provides magnetoencephalography (MEG) recordings from four participants across 12 sessions. Visual stimuli were drawn from the THINGS database (Hebart et al., 2019), encompassing 1854 object concepts and over 26,000 curated naturalistic images. During each session, participants viewed images for 500 ms followed by a blank interval of 1000 ± 200 ms while maintaining central fixation. An orthogonal oddball detection task was employed to ensure sustained attention.

For zero-shot evaluation, 200 object concepts were held out from the training set. The training set consisted of 1854

concepts, each paired with 12 unique images ($1854 \times 12 \times 1$), and the test set included 200 concepts with one image repeated 12 times ($200 \times 1 \times 12$). MEG data were acquired with a 271-channel whole-head system and epoched from 0 to 1000 ms after stimulus onset. Preprocessing involved band-pass filtering in the 0.1–100 Hz range, baseline correction, and downsampling to 200 Hz. To improve signal stability, we averaged repeated responses for each image. Due to the limited number of participants, statistical analysis was not conducted on this dataset, and it serves as a supplementary validation of the EEG-based findings.

## B. Comparison Methods

To validate the efficacy of our proposed approach, we compare ViEEG with some classical and recent methods for EEG cognitive decoding and visual decoding. Below, we provide detailed descriptions of the baseline approaches:

- **ConvNet (Schirrmeister et al., 2017):** A convolutional neural network designed for end-to-end EEG decoding, focusing on extracting spatial and spectral features directly from raw EEG signals.

- **EEGNet (Lawhern et al., 2018):** A lightweight convolutional network that efficiently extracts temporal and spatial features from EEG signals, optimized for BCI applications.

- **DGCNN [IEEE TAFFC'2018] (Song et al., 2018):** The Dynamical Graph CNN employs a learnable adjacency matrix along with Chebyshev filters to capture dynamic relationships for EEG emotion classification.

- **EEG-Conformer [IEEE TNSRE'2022] (Song et al., 2022):** This architecture combines convolutional neural networks with Transformer modules to extract both local and global dependencies in sequential EEG data.

- **EEG-ChannelNet [IEEE TPAMI'2020] (Palazzo et al., 2020):** EEG-ChannelNet models spatial–temporal relationships among EEG channels and learns a latent brain manifold aligned with visual representations through a siamese framework, enabling effective visual decoding and improved performance in image recognition and saliency prediction.

- **MB2C [ACM MM'2024] (Wei et al., 2024):** The Multimodal Bidirectional Cycle Consistency framework utilizes dual-GAN architectures to generate and reconcile modality-specific features, effectively bridging the modality gap in zero-shot tasks, EEG classification, and image reconstruction.

- **NICE [ICLR'2024] (Song et al., 2024):** A self-supervised framework that learns image representations from EEG by incorporating attention modules to capture spatial correlations within brain activity.

- **ATM [NeurIPS'2024] (Li et al., 2024):** The Adaptive Thinking Mapper projects neural signals into a shared subspace with CLIP embeddings via a two-stage EEG-to-image generation strategy, enabling zero-shot visual decoding and reconstruction.

- **CognitionCapturer [AAAI'2025] (Zhang et al., 2024):** This unified framework enhances EEG signal representations by leveraging multimodal data and modality-specific expert encoders. It employs a diffusion prior to map EEG embeddings to the CLIP space, achieving high-fidelity reconstruction of visual stimuli without requiring fine-tuning.

- **BrainFLORA [ACM MM'2025] (Li et al., 2025):** A multimodal framework that integrates EEG, MEG, and fMRI using large language models with modality-specific adapters. This approach creates a shared neural representation, enabling cross-subject visual retrieval and revealing consistent brain concept alignments.

- **SRT [ICCV'2025] (Kim et al., 2025):** A framework to decode visual perception from EEG. By aligning EEG with image-text embeddings, it retrieves semantically related samples that guide a diffusion model for generating high-quality visual reconstructions.

These methods represent a diverse set of strategies for EEG decoding and visual reconstruction, against which ViEEG is rigorously benchmarked.

*Table 5.* Overall Top-1/5 accuracy (%) on THINGS-MEG for 200-way zero-shot recognition.

| Methods | Subject 1 | | Subject 2 | | Subject 3 | | Subject 4 | | Ave | |
|---|---|---|---|---|---|---|---|---|---|---|
| | top-1 | top-5 | top-1 | top-5 | top-1 | top-5 | top-1 | top-5 | top-1 | top-5 |
| Subject dependent - train and test on one subject | | | | | | | | | | |
| EEGNet (Lawhern et al., 2018) | 9.8 | 29.2 | 17.9 | 48.9 | 14.8 | 41.3 | 9.1 | 28.7 | 12.9 | 37.0 |
| DGCNN [TAFFC'18] (Song et al., 2018) | 6.7 | 22.0 | 14.1 | 36.2 | 11.9 | 35.8 | 7.7 | 22.8 | 10.1 | 29.2 |
| EEG-Conformer [TNSRE'22] (Song et al., 2022) | 9.1 | 27.1 | 20.8 | 47.4 | 15.2 | 40.7 | 9.8 | 27.9 | 13.7 | 35.8 |
| EEG-ChannelNet [TPAMI'20] (Palazzo et al., 2020) | 10.8 | 33.0 | 22.6 | 52.2 | 18.5 | 45.2 | 10.4 | 32.6 | 15.6 | 40.8 |
| Mb2C [ACM MM'24] (Wei et al., 2024) | 9.3 | 33.6 | 20.6 | 49.2 | 18.2 | 44.3 | 10.2 | 33.6 | 14.6 | 39.9 |
| NICE [ICLR'24] (Song et al., 2024) | 11.5 | 35.6 | 25.7 | 54.4 | 21.0 | 47.8 | 11.2 | 35.2 | 17.4 | 43.3 |
| ATM [NeurIPS'24] (Li et al., 2024) | 8.0 | 29.3 | 30.2 | 61.5 | 20.3 | 50.5 | 11.8 | 33.3 | 17.6 | 43.7 |
| CognitionCapturer [AAAI'25] (Zhang et al., 2024) | 10.1 | 30.6 | 26.9 | 55.1 | 20.4 | 49.3 | 11.5 | 34.2 | 17.2 | 42.3 |
| **ViEEG** [Ours] | **16.6** | **50.3** | **37.4** | **79.1** | **30.6** | **73.9** | **17.2** | **49.5** | **25.5** | **63.2** |

# C. Experiment in THINGS-MEG Dataset

Due to the limited number of subjects in the THINGS-MEG dataset, we conducted subject-dependent experiments, training and testing separately on each subject. Table 5 reports the 200-way zero-shot recognition results. As shown in Table 5, ViEEG also achieves superior performance on the THINGS-MEG dataset, outperforming all baselines in both top-1 and top-5 accuracy across subjects. These results demonstrate that our neuroscience-inspired framework is not only effective for EEG-based decoding, but also generalizes well to other neural signals such as MEG, further highlighting the robustness and cross-modality potential of our proposed ViEEG.

# D. Whole Results of Ablation Study and Parameter Analysis

Due to space limitations, only partial information is presented in the main text. The complete results of the ablation experiments on Hierarchical Embedding and cross-attention are provided here. Table 6 shows the object recognition accuracy of Hierarchical Embedding in the full ViEEG, while Table 7 presents the object recognition accuracy of Hierarchical Embedding in ViEEG without cross-attention.

### D.1. Ablation on Hierarchical Embedding

The results presented in Table 6 showcase the performance of different hierarchical EEG feature combinations on object recognition. These results were obtained from ViEEG with the full cross-attention mechanism.

We investigated the impact of three distinct visual embeddings: $F_b$ (contour-based), $F_f$ (foreground-object-focused), and $F_r$ (contextual). Each of these embeddings contributes valuable information, but their individual contributions are limited when used in isolation.

$\mathbf{F_b}$**-Only (contour-based embedding)** showed moderate performance, with Top-1 accuracy ranging from 14.2% to 17.6% in the subject-dependent setting and 5.9% to 10.2 in the subject-independent setting. The contour-based features alone provide basic structure but lack the granularity needed for high accuracy.

$\mathbf{F_f}$**-Only (object-focused embedding)** consistently outperformed $F_b$-Only, with Top-1 accuracy ranging from 22.2% to 38.7% in the subject-dependent setting and 15.8% to 23.4% in the subject-independent setting. The object-focused embedding captures detailed object features, enhancing the model's ability to distinguish between classes, particularly in controlled settings.

$\mathbf{F_r}$**-Only (contextual embedding)** also improved performance over $F_b$-Only, with Top-1 accuracy ranging from 22.1% to 40.6% in the subject-dependent setting and 14.9% to 31.8% in the subject-independent setting. Contextual information helps provide broader scene understanding, aiding the model in handling more complex visual stimuli.

When combining all three embeddings ($F_b$, $F_f$, and $F_r$) as $\mathbf{F_{EEG}}$, the model achieved the best results, with Top-1 accuracy reaching 34.1% to 54.0% in the subject-dependent setting and 22.7% to 31.2% in the subject-independent setting. This combination benefits from the strengths of all three feature types, with $F_f$ contributing detailed object-level features, $F_r$ enhancing contextual understanding, and $F_b$ offering contour-based structural information. The complementary nature of

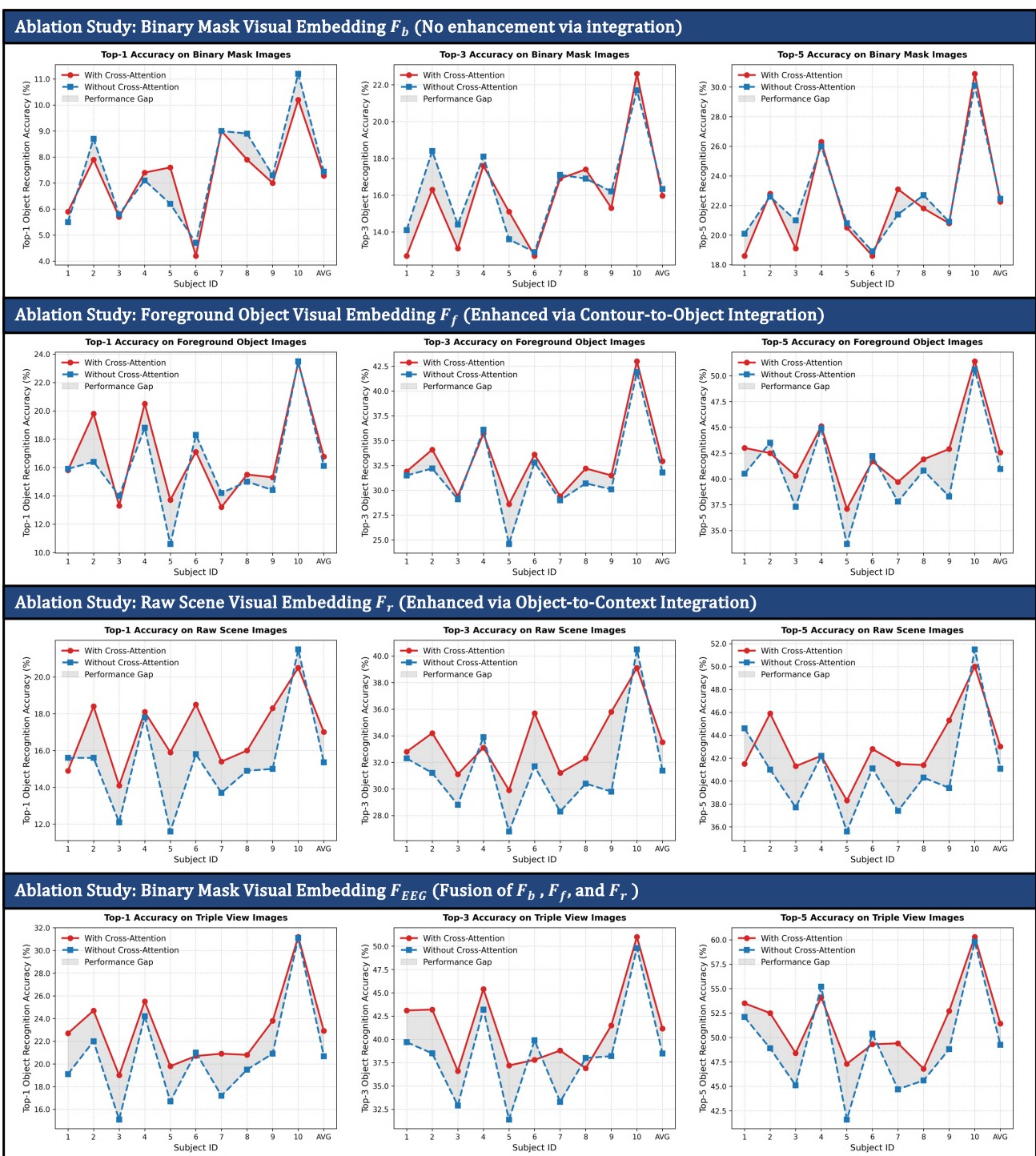

*Figure 9.* Detailed ablation study of attention on hierarchical visual embeddings in ViEEG.

*Table 6.* Object recognition accuracy for different hierarchical representation integration in ViEEG with cross-attention module.

| $F_b$ $F_f$ $F_r$ | Subject-Dependent Setting | | | | | | | | | | | Subject-Independent Setting | | | | | | | | | | |
|---|---|---|---|---|---|---|---|---|---|---|---|---|---|---|---|---|---|---|---|---|---|---|
| | Sub-1 | Sub-2 | Sub-3 | Sub-4 | Sub-5 | Sub-6 | Sub-7 | Sub-8 | Sub-9 | Sub-10 | AVG | Sub-1 | Sub-2 | Sub-3 | Sub-4 | Sub-5 | Sub-6 | Sub-7 | Sub-8 | Sub-9 | Sub-10 | AVG |
| Top-1 Object Recognition Accuracy | | | | | | | | | | | | | | | | | | | | | | |
| ✓ ✗ ✗ | 14.2 | 14.0 | 17.0 | 20.9 | 8.6 | 13.6 | 14.7 | 17.6 | 12.8 | 14.8 | 14.8 | 5.9 | 7.9 | 5.7 | 7.4 | 7.6 | 4.2 | 9.0 | 7.9 | 7.0 | 10.2 | 7.28 |
| ✗ ✓ ✗ | 22.2 | 24.5 | 29.9 | 34.0 | 19.2 | 32.5 | 26.5 | 38.7 | 29.3 | 31.8 | 28.9 | 15.8 | 19.8 | 13.3 | 20.5 | 13.7 | 17.1 | 13.2 | 15.5 | 15.3 | 23.4 | 16.76 |
| ✗ ✗ ✓ | 22.1 | 24.8 | 31.4 | 33.1 | 18.5 | 29.7 | 26.8 | 40.6 | 27.4 | 31.8 | 28.6 | 14.9 | 18.4 | 14.1 | 18.1 | 15.9 | 18.5 | 15.4 | 16.0 | 18.3 | 20.5 | 17.01 |
| ✓ ✓ ✗ | 31.4 | 32.8 | 35.0 | 46.4 | 25.6 | 37.5 | 33.3 | 46.8 | 33.8 | 40.3 | 36.3 | 16.8 | 20.4 | 12.8 | 20.1 | 16.1 | 16.4 | 17.8 | 17.6 | 16.9 | 28.4 | 18.33 |
| ✓ ✗ ✓ | 32.2 | 30.8 | 35.6 | 47.5 | 24.1 | 38.1 | 34.9 | 45.1 | 32.0 | 39.6 | 36.0 | 16.5 | 20.5 | 13.1 | 20.6 | 16.0 | 18.0 | 17.6 | 17.6 | 19.9 | 27.5 | 18.73 |
| ✗ ✓ ✓ | 25.1 | 28.8 | 33.8 | 39.6 | 22.0 | 34.7 | 28.4 | 43.0 | 31.5 | 33.9 | 32.1 | 18.5 | 21.8 | 16.7 | 22.0 | 19.3 | **21.3** | 16.1 | 18.7 | 19.8 | 27.3 | 20.15 |
| ✓ ✓ ✓ | **34.1** | **38.4** | **40.6** | **50.1** | **28.9** | **44.3** | **38.6** | **54.0** | **37.3** | **42.8** | **40.9** | **22.7** | **24.7** | **19.0** | **25.5** | **19.8** | 20.7 | **20.9** | **20.8** | **23.8** | **31.2** | **22.9** |
| Top-3 Object Recognition Accuracy | | | | | | | | | | | | | | | | | | | | | | |
| ✓ ✗ ✗ | 29.8 | 30.0 | 29.8 | 40.1 | 20.2 | 28.6 | 30.6 | 35.3 | 26.7 | 32.9 | 30.4 | 12.7 | 16.3 | 13.1 | 17.6 | 15.1 | 12.7 | 16.9 | 17.4 | 15.3 | 22.6 | 15.97 |
| ✗ ✓ ✗ | 41.2 | 46.0 | 51.4 | 58.3 | 36.6 | 53.6 | 45.4 | 63.3 | 50.7 | 58.1 | 50.5 | 31.9 | 34.1 | 29.4 | 35.7 | 28.6 | 33.6 | 29.4 | 32.2 | 31.5 | 43.0 | 32.94 |
| ✗ ✗ ✓ | 40.6 | 46.2 | 51.5 | 56.8 | 36.6 | 55.9 | 48.9 | 65.1 | 51.3 | 55.9 | 50.9 | 32.8 | 34.2 | 31.1 | 33.1 | 29.9 | 35.7 | 31.2 | 32.3 | 35.8 | 39.1 | 33.52 |
| ✓ ✓ ✗ | 54.3 | 53.5 | 57.6 | 69.9 | 45.6 | 60.8 | 56.7 | 69.4 | 56.7 | 62.0 | 58.7 | 33.2 | 35.0 | 30.4 | 39.4 | 28.5 | 32.4 | 33.3 | 31.1 | 32.5 | 47.6 | 34.34 |
| ✓ ✗ ✓ | 56.2 | 53.4 | 57.7 | 68.8 | 43.5 | 60.1 | 57.9 | 69.7 | 55.5 | 63.4 | 58.6 | 34.0 | 38.9 | 32.7 | 39.0 | 31.5 | 34.1 | 33.0 | 32.4 | 34.2 | 45.4 | 35.52 |
| ✗ ✓ ✓ | 46.5 | 50.4 | 56.3 | 61.2 | 40.9 | 57.8 | 52.3 | 68.7 | 56.0 | 61.1 | 55.1 | 35.2 | 38.5 | 34.5 | 38.3 | 34.4 | 36.4 | 33.8 | 36.6 | 38.1 | 46.3 | 37.21 |
| ✓ ✓ ✓ | **58.3** | **58.1** | **62.6** | **72.2** | **52.3** | **66.8** | **63.2** | **74.1** | **62.0** | **68.9** | **63.9** | **43.1** | **43.2** | **36.6** | **45.4** | **37.2** | **37.8** | **38.8** | **36.9** | **41.5** | **51.0** | **41.2** |
| Top-5 Object Recognition Accuracy | | | | | | | | | | | | | | | | | | | | | | |
| ✓ ✗ ✗ | 41.8 | 39.8 | 39.1 | 53.2 | 30.3 | 40.9 | 40.7 | 46.4 | 37.6 | 44.0 | 41.4 | 18.6 | 22.8 | 19.1 | 26.3 | 20.5 | 18.6 | 23.1 | 21.8 | 20.8 | 30.9 | 22.25 |
| ✗ ✓ ✗ | 52.1 | 56.1 | 66.0 | 67.1 | 50.0 | 63.5 | 58.7 | 73.9 | 61.8 | 68.5 | 61.8 | 43.0 | 42.5 | 40.3 | 45.1 | 37.1 | 41.7 | 39.7 | 41.9 | 42.9 | 51.4 | 42.56 |
| ✗ ✗ ✓ | 52.6 | 58.0 | 62.7 | 67.4 | 49.1 | 66.2 | 60.1 | 74.1 | 62.5 | 68.6 | 62.1 | 41.5 | 45.9 | 41.3 | 42.2 | 38.3 | 42.8 | 41.5 | 41.4 | 45.3 | 50.0 | 43.02 |
| ✓ ✓ ✗ | 64.5 | 62.7 | 69.2 | 78.8 | 55.5 | 73.0 | 69.3 | 78.1 | 66.8 | 72.5 | 69.0 | 43.7 | 45.6 | 40.2 | 47.9 | 38.8 | 41.3 | 43.1 | 39.3 | 42.8 | 56.3 | 43.90 |
| ✓ ✗ ✓ | 66.4 | 63.2 | 69.7 | 78.3 | 54.8 | 72.7 | 69.5 | 79.0 | 65.2 | 72.6 | 69.1 | 43.9 | 46.8 | 42.8 | 49.1 | 41.0 | 41.9 | 42.0 | 39.6 | 46.6 | 54.5 | 44.82 |
| ✗ ✓ ✓ | 57.8 | 62.2 | 68.3 | 70.8 | 52.5 | 66.9 | 64.7 | 77.7 | 66.9 | 72.0 | 66.0 | 45.1 | 48.1 | 46.5 | 46.7 | 43.7 | 45.8 | 45.0 | 45.5 | 49.3 | 56.7 | 47.24 |
| ✓ ✓ ✓ | **71.3** | **67.9** | **74.7** | **80.8** | **61.5** | **76.5** | **75.2** | **82.5** | **74.9** | **79.8** | **74.5** | **53.5** | **52.5** | **48.4** | **54.1** | **47.3** | **49.3** | **49.4** | **46.8** | **52.7** | **60.3** | **51.4** |

these embeddings highlights the importance of integrating multiple types of visual information for effective EEG-based object recognition.

These findings underscore the need for a comprehensive, multi-embedding approach, where each embedding type contributes specific and essential information. Combining these embeddings leads to the most effective hierarchical feature representation, improving both subject-dependent and subject-independent recognition accuracy.

### D.2. Ablation on Cross Attention

We further analyzed the effectiveness of the cross-attention module, which is essential for integrating the hierarchical features in ViEEG. Removing the cross-attention component (w/o C-Att) resulted in a noticeable performance drop. In the subject-dependent experiments, the configuration without cross-attention achieved a Top-1 accuracy of 32.4% and Top-5 accuracy of 66.8%, compared to 34.1% and 71.3% for the full ViEEG. In the subject-independent setting, similar trends were observed, with the full model outperforming the variant without cross-attention by significant margins. This highlights the crucial role of cross-attention in aligning and integrating hierarchical EEG embeddings, thereby improving the robustness and generalization of the model across different subjects.

The impact of cross-attention is particularly evident when integrating the different feature embeddings. As illustrated in Figure 9, the addition of cross-attention leads to significant improvements in the EEG decoding of the integrated features, especially for the following:

**Binary Mask Visual Embedding** $F_b$: Since this feature was not integrated or enhanced via cross-attention, its performance remains similar to the ablation results. As shown in Table 6, the Top-1 accuracy for $F_b$-Only is quite low, with values around 14.2% to 17.6% in the subject-dependent setting and 5.9% to 10.2% in the subject-independent setting. This shows that, without cross-attention, the feature alone is insufficient for effective object recognition.

*Table 7.* Object recognition accuracy for different hierarchical representation integration in ViEEG without cross-attention module.

| $F_b$ | $F_f$ | $F_r$ | Subject-Dependent Setting | | | | | | | | | | | Subject-Independent Setting | | | | | | | | | | |
|---|---|---|---|---|---|---|---|---|---|---|---|---|---|---|---|---|---|---|---|---|---|---|---|---|
| | | | Sub-1 | Sub-2 | Sub-3 | Sub-4 | Sub-5 | Sub-6 | Sub-7 | Sub-8 | Sub-9 | Sub-10 | AVG | Sub-1 | Sub-2 | Sub-3 | Sub-4 | Sub-5 | Sub-6 | Sub-7 | Sub-8 | Sub-9 | Sub-10 | AVG |
| | | | Top-1 Object Recognition Accuracy | | | | | | | | | | | | | | | | | | | | | |
| ✓ | ✗ | ✗ | 15.9 | 14.5 | 16.0 | 20.1 | 9.7 | 11.3 | 15.3 | 17.9 | 12.0 | 15.1 | 14.8 | 5.5 | 8.7 | 5.8 | 7.1 | 6.2 | 4.7 | 9.0 | 8.9 | 7.3 | 11.2 | 7.44 |
| ✗ | ✓ | ✗ | 21.4 | 22.8 | 28.6 | 34.6 | 19.4 | 26.0 | 26.8 | 38.6 | 21.8 | 30.8 | 27.1 | 15.9 | 16.4 | 14.0 | 18.8 | 10.6 | 18.3 | 14.2 | 15.0 | 14.4 | 23.5 | 16.11 |
| ✗ | ✗ | ✓ | 21.6 | 23.2 | 27.8 | 33.4 | 18.9 | 28.1 | 25.8 | 35.6 | 23.4 | 32.8 | 27.1 | 15.6 | 15.6 | 12.1 | 17.8 | 11.6 | 15.8 | 13.7 | 14.9 | 15.0 | 21.5 | 15.36 |
| ✓ | ✓ | ✗ | 30.5 | 30.0 | 32.9 | 42.4 | 24.1 | 33.7 | 33.8 | 46.3 | 31.6 | 36.9 | 34.2 | 17.9 | 18.5 | 12.9 | 22.7 | 13.3 | 17.8 | 16.5 | 18.0 | 17.8 | 26.7 | 18.21 |
| ✓ | ✗ | ✓ | 30.0 | 31.9 | 32.5 | 44.2 | 24.5 | 34.4 | 35.4 | 43.3 | 31.2 | 38.1 | 34.6 | 17.0 | 19.6 | 11.9 | 19.6 | 14.4 | 16.2 | 15.9 | 18.0 | 18.4 | 26.5 | 17.75 |
| ✗ | ✓ | ✓ | 24.0 | 26.2 | 31.6 | 38.9 | 21.9 | 31.4 | 27.8 | 42.7 | 26.8 | 35.6 | 30.7 | 17.9 | 18.7 | 14.8 | 20.4 | 13.4 | 20.3 | 14.2 | 17.4 | 18.1 | 24.6 | 17.98 |
| ✓ | ✓ | ✓ | **32.4** | **34.6** | **39.1** | **46.5** | **27.2** | **40.5** | **35.3** | **52.2** | **34.6** | **43.9** | **38.6** | **19.1** | **22.0** | **15.1** | **24.2** | **16.7** | **21.0** | **17.2** | **19.5** | **20.9** | **31.1** | **20.7** |
| | | | Top-3 Object Recognition Accuracy | | | | | | | | | | | | | | | | | | | | | |
| ✓ | ✗ | ✗ | 30.6 | 29.8 | 28.6 | 37.9 | 23.1 | 26.6 | 31.3 | 34.9 | 29.8 | 33.8 | 30.6 | 13.6 | 17.9 | 13.9 | 17.6 | 13.1 | 12.4 | 16.6 | 16.4 | 15.7 | 21.2 | 15.84 |
| ✗ | ✓ | ✗ | 38.8 | 40.1 | 48.7 | 55.3 | 35.7 | 49.5 | 46.5 | 62.6 | 41.9 | 55.5 | 47.5 | 31.5 | 32.2 | 29.1 | 36.1 | 24.6 | 32.8 | 29.0 | 30.7 | 30.1 | 41.9 | 31.80 |
| ✗ | ✗ | ✓ | 40.0 | 44.2 | 48.3 | 55.4 | 36.6 | 50.7 | 48.7 | 60.5 | 46.5 | 55.4 | 48.6 | 32.3 | 31.2 | 28.8 | 33.9 | 26.8 | 31.7 | 28.3 | 30.4 | 29.8 | 40.5 | 31.37 |
| ✓ | ✓ | ✗ | 53.4 | 51.7 | 56.1 | 66.8 | 43.7 | 56.5 | 55.1 | 66.7 | 53.9 | 62.0 | 56.6 | 34.5 | 34.3 | 29.7 | 38.6 | 28.1 | 34.1 | 30.3 | 31.9 | 33.4 | 45.5 | 34.04 |
| ✓ | ✗ | ✓ | **54.8** | 51.8 | 54.9 | 67.5 | 43.0 | 58.5 | 57.4 | 67.8 | 53.9 | 61.6 | 57.1 | 34.4 | 32.6 | 28.8 | 37.8 | 27.6 | 34.7 | 28.9 | 31.4 | 34.2 | 44.8 | 33.52 |
| ✗ | ✓ | ✓ | 43.4 | 47.7 | 52.4 | 61.2 | 40.3 | 55.5 | 52.2 | 67.0 | 51.2 | 60.8 | 53.2 | 35.8 | 35.8 | 31.2 | 37.8 | 29.1 | 36.5 | 29.9 | 35.0 | 34.3 | 45.9 | 35.13 |
| ✓ | ✓ | ✓ | 54.3 | **56.1** | **60.7** | **71.4** | **48.6** | **62.2** | **57.6** | **72.4** | **59.9** | **65.1** | **60.8** | **39.7** | **38.5** | **32.9** | **43.2** | **31.4** | **39.9** | **33.3** | **38.0** | **38.2** | **49.8** | **38.5** |
| | | | Top-5 Object Recognition Accuracy | | | | | | | | | | | | | | | | | | | | | |
| ✓ | ✗ | ✗ | 41.6 | 39.8 | 37.2 | 51.1 | 31.6 | 37.9 | 42.4 | 46.1 | 40.7 | 46.3 | 41.5 | 20.1 | 22.6 | 21.0 | 26.0 | 20.8 | 18.9 | 21.4 | 22.7 | 20.9 | 30.1 | 22.45 |
| ✗ | ✓ | ✗ | 47.6 | 52.9 | 60.7 | 66.4 | 47.3 | 60.0 | 59.1 | 71.4 | 54.8 | 66.8 | 58.7 | 40.5 | 43.5 | 37.3 | 44.9 | 33.7 | 42.2 | 37.8 | 40.8 | 38.3 | 50.6 | 40.96 |
| ✗ | ✗ | ✓ | 50.4 | 55.2 | 59.4 | 67.4 | 47.6 | 61.6 | 60.4 | 71.0 | 60.4 | 66.0 | 59.9 | 44.6 | 41.0 | 37.7 | 42.2 | 35.6 | 41.1 | 37.4 | 40.3 | 39.4 | 51.5 | 41.08 |
| ✓ | ✓ | ✗ | 65.1 | 61.9 | 69.2 | 78.8 | 54.6 | 69.1 | 66.6 | 76.3 | 64.7 | 72.3 | 67.9 | 43.7 | 43.1 | 41.9 | 48.5 | 36.5 | 42.6 | 41.7 | 40.3 | 40.8 | 56.0 | 43.51 |
| ✓ | ✗ | ✓ | 64.6 | 59.9 | 66.4 | 78.4 | 55.4 | 69.5 | 68.9 | 75.6 | 66.5 | 72.4 | 67.8 | 44.7 | 43.2 | 39.9 | 48.6 | 38.5 | 43.5 | 39.1 | 40.1 | 43.3 | 54.5 | 43.54 |
| ✗ | ✓ | ✓ | 55.9 | 60.1 | 64.3 | 69.4 | 52.1 | 65.9 | 64.6 | 75.5 | 63.1 | 70.2 | 64.1 | 47.2 | 47.8 | 40.8 | 47.6 | 38.5 | 45.1 | 41.4 | 44.4 | 44.0 | 53.9 | 45.07 |
| ✓ | ✓ | ✓ | **66.8** | **65.9** | **71.3** | **80.2** | **59.8** | **74.3** | **71.3** | **80.8** | **71.9** | **75.5** | **71.8** | **52.1** | **48.9** | **45.1** | **55.2** | **41.6** | **50.4** | **44.7** | **45.6** | **48.8** | **59.8** | **49.2** |

**Foreground Object Visual Embedding** $F_f$: This feature benefits from contour-to-object integration, which improves performance but to a smaller extent compared to other more comprehensive features. The Top-1 accuracy for $F_f$-Only ranged from 22.2% to 38.7% in the subject-dependent setting and 15.8% to 23.4% in the subject-independent setting. Adding cross-attention slightly enhances its performance, demonstrating the importance of cross-attention in boosting object-specific feature extraction.

**Raw Scene Visual Embedding** $F_r$: This feature also shows an improvement through object-to-context integration, leading to higher accuracy compared to $F_b$ and $F_f$. The Top-1 accuracy for $F_r$-Only ranged from 22.1% to 40.6% in the subject-dependent setting and 14.9% to 31.8% in the subject-independent setting. Cross-attention further improves this performance by aligning the raw scene features with the context, helping the model leverage broader scene information.

**Combined Binary Mask Visual Embedding** $F_{EEG}$: The concatenated representation of $F_b$, $F_f$, and $F_r$ saw the most significant improvement after adding cross-attention. As seen in the results, this combination achieved a Top-1 accuracy of 34.1% to 54.0% in the subject-dependent setting and 22.7% to 31.2% in the subject-independent setting, which is substantially higher than the individual embeddings. Cross-attention allows for the effective integration of these features, leading to a more comprehensive and robust EEG representation.

In Figure 9, we present line charts that highlight the changes in accuracy before and after the integration of cross-attention for each feature. The four subplots represent the Top-1 and Top-5 accuracy changes for each feature: $F_b$, $F_f$, $F_r$, and $F_{EEG}$. These subplots illustrate the performance improvements across the subject-dependent and subject-independent settings. The line charts clearly show that, although individual embeddings benefit from cross-attention, the most significant boost is observed when combining all three embeddings. This visualization allows for a clearer comparison of how cross-attention enhances the overall performance of ViEEG across various feature combinations. More detailed results are available in the Appendix for further reference.

*Table 8.* Ablation study of EEG encoder: Top-1/Top-5 object recognition accuracy(%).

| Methods | Subject 1 | | Subject 2 | | Subject 3 | | Subject 4 | | Subject 5 | | Subject 6 | | Subject 7 | | Subject 8 | | Subject 9 | | Subject 10 | | Ave | |
|---|---|---|---|---|---|---|---|---|---|---|---|---|---|---|---|---|---|---|---|---|---|---|
| | top-1 | top-5 | top-1 | top-5 | top-1 | top-5 | top-1 | top-5 | top-1 | top-5 | top-1 | top-5 | top-1 | top-5 | top-1 | top-5 | top-1 | top-5 | top-1 | top-5 | top-1 | top-5 |
| *Subject dependent - train and test on one subject* | | | | | | | | | | | | | | | | | | | | | | |
| LSTM (Hochreiter, 1997) | 25.8 | 60.5 | 23.1 | 49.3 | 30.5 | 60.8 | 29.6 | 60.1 | 20.3 | 41.2 | 25.4 | 51.7 | 28.5 | 59.0 | 30.6 | 62.0 | 27.0 | 53.1 | 34.8 | 67.7 | 27.6 | 56.5 |
| EEGNet (Lawhern et al., 2018) | 21.6 | 55.4 | 25.3 | 59.2 | 25.3 | 61.2 | 30.5 | 67.1 | 16.4 | 46.5 | 26.7 | 61.3 | 25.2 | 58.6 | 37.1 | 71.1 | 26.0 | 61.2 | 32.8 | 67.6 | 26.7 | 60.9 |
| DGCNN [TAFFC'18] (Song et al., 2018) | 18.2 | 49.1 | 14.8 | 43.2 | 20.5 | 52.4 | 29.0 | 63.2 | 12.8 | 38.8 | 17.8 | 53.9 | 20.6 | 55.4 | 29.6 | 65.4 | 21.7 | 52.2 | 26.1 | 62.4 | 21.1 | 53.6 |
| NICE [ICLR'24] (Song et al., 2024) | 31.1 | 66.7 | 34.2 | 65.8 | 38.9 | 71.2 | 45.8 | 79.5 | 26.7 | 69.1 | 40.6 | 74.3 | 35.4 | 70.6 | 50.2 | 80.5 | 35.1 | 72.3 | 40.0 | 75.7 | 37.8 | 72.6 |
| ATM [NeurIPS'24] (Li et al., 2024) | 30.8 | 67.3 | 34.8 | 65.6 | 36.6 | 72.3 | 44.9 | 80.6 | 28.2 | 60.9 | 37.5 | 72.2 | 39.1 | 74.6 | 49.3 | 80.9 | 34.2 | 71.6 | 40.7 | 77.8 | 37.6 | 72.4 |
| CognitionCapturer [AAAI'25] (Zhang et al., 2024) | 30.2 | 65.6 | 33.1 | 64.1 | 37.4 | 70.8 | 43.8 | 78.9 | 25.7 | 57.4 | 34.6 | 63.5 | 35.9 | 71.1 | 48.7 | 78.8 | 35.6 | 70.3 | 38.2 | 76.5 | 36.3 | 69.7 |
| **ViEEG** [Ours] | **34.1** | **71.3** | **38.4** | **67.9** | **40.6** | **74.7** | **50.1** | **80.8** | **28.9** | **61.5** | **44.3** | **76.5** | **38.6** | **75.2** | **54.0** | **82.5** | **37.3** | **74.9** | **42.8** | **79.8** | **40.9** | **74.5** |
| *Subject independent - leave one subject out for test* | | | | | | | | | | | | | | | | | | | | | | |
| LSTM (Hochreiter, 1997) | 14.8 | 38.1 | 13.8 | 32.5 | 8.0 | 24.6 | 13.0 | 35.8 | 10.2 | 27.6 | 12.2 | 32.6 | 8.5 | 26.5 | 14.3 | 38.8 | 6.5 | 28.2 | 14.1 | 37.4 | 11.5 | 32.2 |
| EEGNet (Lawhern et al., 2018) | 13.5 | 36.2 | 15.9 | 40.7 | 7.6 | 27.8 | 16.0 | 37.4 | 10.3 | 30.0 | 12.2 | 38.2 | 8.9 | 29.5 | 16.8 | 39.5 | 10.5 | 30.6 | 21.2 | 51.0 | 13.3 | 36.1 |
| DGCNN (Song et al., 2018) | 12.9 | 36.5 | 17.5 | 39.3 | 6.3 | 25.5 | 12.8 | 31.0 | 9.2 | 32.0 | 13.7 | 36.9 | 7.3 | 28.8 | 14.3 | 32.8 | 14.2 | 36.4 | 15.6 | 41.3 | 12.4 | 34.1 |
| NICE [ICLR'24] (Song et al., 2024) | 18.4 | 50.3 | 21.9 | 47.3 | 14.5 | 44.6 | 22.1 | 53.9 | 15.0 | 39.8 | 20.2 | 48.4 | 16.3 | 43.1 | 18.4 | 44.6 | 19.7 | 45.2 | 27.8 | 45.5 | 19.4 | 46.3 |
| ATM [NeurIPS'24] (Li et al., 2024) | 20.7 | 48.0 | 22.2 | 49.6 | 14.3 | 39.6 | 20.5 | 48.2 | 18.2 | 43.0 | 18.0 | 44.1 | 20.5 | 42.3 | 21.6 | 46.3 | 19.1 | 41.6 | 27.2 | 55.7 | 19.8 | 45.8 |
| CognitionCapturer [AAAI'25] (Zhang et al., 2024) | 19.1 | 47.4 | 17.1 | 44.5 | 10.2 | 29.0 | 20.3 | 44.2 | 11.2 | 32.9 | 17.3 | 38.6 | 15.3 | 35.5 | 17.4 | 39.7 | 13.8 | 38.8 | 22.5 | 50.6 | 16.4 | 40.1 |
| **ViEEG** [Ours] | **22.7** | **53.5** | **24.7** | **52.5** | **19.0** | **48.4** | **25.5** | **54.1** | **19.8** | **47.3** | **20.7** | **49.3** | **20.9** | **49.4** | **20.8** | **46.8** | **23.8** | **52.7** | **31.2** | **60.3** | **22.9** | **51.4** |

*Table 9.* Ablation study of hierarchical processing order: Top-1/Top-5 object recognition accuracy(%).

| Methods | Subject 1 | | Subject 2 | | Subject 3 | | Subject 4 | | Subject 5 | | Subject 6 | | Subject 7 | | Subject 8 | | Subject 9 | | Subject 10 | | Ave | |
|---|---|---|---|---|---|---|---|---|---|---|---|---|---|---|---|---|---|---|---|---|---|---|
| | top-1 | top-5 | top-1 | top-5 | top-1 | top-5 | top-1 | top-5 | top-1 | top-5 | top-1 | top-5 | top-1 | top-5 | top-1 | top-5 | top-1 | top-5 | top-1 | top-5 | top-1 | top-5 |
| *Subject dependent - train and test on one subject* | | | | | | | | | | | | | | | | | | | | | | |
| contour→context→object | 34.2 | 70.6 | 37 | 68.2 | 36.8 | 73.3 | 47.6 | 81.8 | 28.6 | 61.0 | 41.8 | 75.4 | 39.7 | 73.8 | 53.2 | 81.3 | 36.3 | 73.1 | 42.7 | 77.9 | 39.7 | 73.6 |
| object→contour→context | 34.9 | 68.0 | 34.5 | 67.6 | 39.3 | 74.2 | 47.3 | 79.1 | 27.7 | 60.9 | 42.6 | 75.1 | 39.6 | 75.7 | 52.5 | 81.5 | 37.2 | 63.1 | 42.7 | 77.0 | 39.8 | 72.2 |
| object→context→contour | 34.4 | 69.4 | 35.8 | 67.8 | 40.2 | 74.1 | 46.9 | 79.0 | 28.3 | 62.7 | 40.2 | 72.6 | 40.2 | 75.3 | 54 | 80.5 | 37.5 | 76.1 | 43.6 | 77.8 | 40.1 | 73.5 |
| context→contour→object | 35.2 | 69.5 | 36.5 | 67.0 | 40.6 | 76.2 | 47.4 | 78.3 | 28.5 | 60.9 | 42.6 | 74.4 | 39.9 | 74.5 | 52.4 | 81.2 | 35.6 | 74.8 | 44.7 | 78.2 | 40.3 | 73.5 |
| context→object→contour | 33.8 | 67.7 | 37.2 | 66.9 | 38.4 | 74.0 | 45.5 | 79.0 | 31.1 | 61.7 | 40.9 | 74.9 | 38.3 | 74.9 | 53.8 | 81.0 | 36.2 | 73.5 | 45.2 | 80.1 | 40.0 | 73.3 |
| **contour→object→contex (ViEEG)** | **34.1** | **71.3** | **38.4** | **67.9** | **40.6** | **74.7** | **50.1** | **80.8** | **28.9** | **61.5** | **44.3** | **76.5** | **38.6** | **75.2** | **54.0** | **82.5** | **37.3** | **74.9** | **42.8** | **79.8** | **40.9** | **74.5** |
| *Subject independent - leave one subject out for test* | | | | | | | | | | | | | | | | | | | | | | |
| contour→context→object | 19.5 | 50.2 | 24.9 | 50.3 | 15.7 | 45.6 | 25.6 | 52.2 | 20.9 | 43.7 | 19.9 | 49 | 17.4 | 46.5 | 20.6 | 46.8 | 21.6 | 49.1 | 29.1 | 64.8 | 21.5 | 49.8 |
| object→contour→context | 21.4 | 53.3 | 23.7 | 52.3 | 18.9 | 49.6 | 25.6 | 54.2 | 16.4 | 45.6 | 20.5 | 47.7 | 19.5 | 47.8 | 21.5 | 43.6 | 22.3 | 54.6 | 30.1 | 63.4 | 21.9 | 51.2 |
| object→context→contour | 19.8 | 51.3 | 24.5 | 52.8 | 18.5 | 49.3 | 26.3 | 54.7 | 20.5 | 45.7 | 20.7 | 48.1 | 18.5 | 44.9 | 18.9 | 44.7 | 20.1 | 54.7 | 30.7 | 61.2 | 21.8 | 50.7 |
| context→contour→object | 21.6 | 51.8 | 24.6 | 51.9 | 17.4 | 48.0 | 25.7 | 55.9 | 20.0 | 45.2 | 21.1 | 49.6 | 20.2 | 47.2 | 20.6 | 44.1 | 22.3 | 52.7 | 30.0 | 59.9 | 22.3 | 50.6 |
| context→object→contour | 20.8 | 51.4 | 25.9 | 52.0 | 18.8 | 49.4 | 26.2 | 54.6 | 19.0 | 46.1 | 21.4 | 48.5 | 18.7 | 45.3 | 20.8 | 45.0 | 22.8 | 51.6 | 29.7 | 60.8 | 22.4 | 50.4 |
| **contour→object→contex (ViEEG)** | **22.7** | **53.5** | **24.7** | **52.5** | **19.0** | **48.4** | **25.5** | **54.1** | **19.8** | **47.3** | **20.7** | **49.3** | **20.9** | **49.4** | **20.8** | **46.8** | **23.8** | **52.7** | **31.2** | **60.3** | **22.9** | **51.4** |

## D.3. Ablation on EEG Encoder

We conducted a detailed ablation study of EEG Encoder. For ViEEG, the EEG feature encoder plays a crucial role as same as image segmentation, and it is the foundation for further modal alignment. Replacing our EEG encoder modules with other EEG encoders (e.g., ATM (Li et al., 2024) and NICE (Song et al., 2024)) for feature extraction led to an average performance significantly drop, as shown from Table 8 below. Our proposed ViEEG achieves the best performance in every experiment, which demonstrates that the SOTA performance of our method is not only dependent on image segmentation features, but also on the hierarchical EEG Encoder. Moreover, as shown by the "increase" value from tables, our proposed hierarchical visual decoding framework shows a significantly improvement in the accuracy rate of object recognition. Thus, ViEEG's SOTA performance arises from the synergy between biologically motivated segmentation and hierarchical EEG encoding.

## D.4. Ablation on Hierarchical Processing Order

While our framework adopts a bottom-up order (contour → object → scene) based on visual neuroscience, we also tested other alternative integration orders (e.g., object → contour → scene, random) in the Table 9. Results show that the biologically inspired order consistently outperforms others, suggesting that respecting the natural visual hierarchy better aligns EEG features with semantic structure.

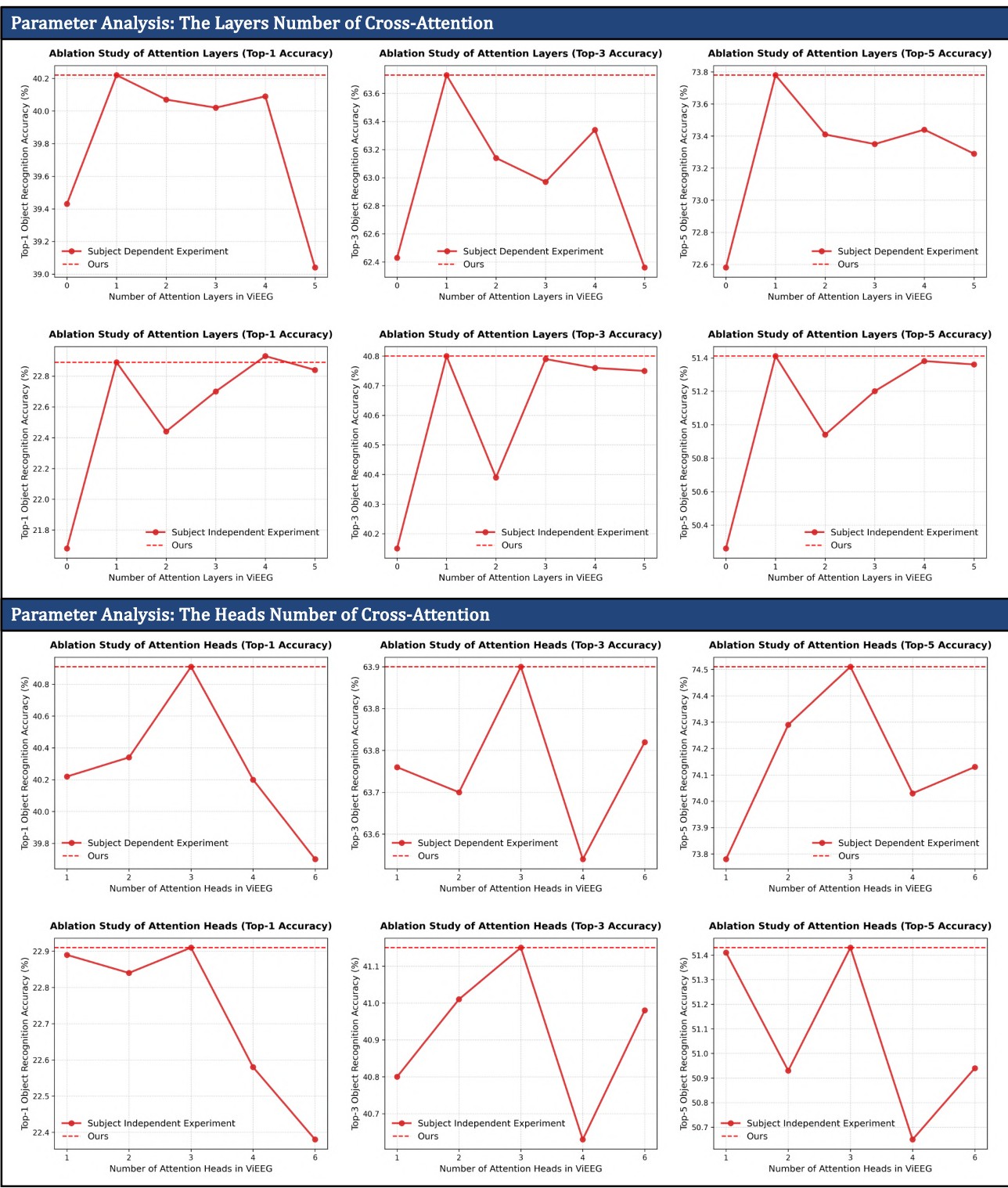

*Figure 10.* **Parameter analysis of attention modules in ViEEG.** Top row: Performance trends (Top-1, Top-3, Top-5 accuracies) for varying numbers of attention layers under subject-dependent and subject-independent settings. Bottom row: Performance trends for different numbers of attention heads under the same settings.

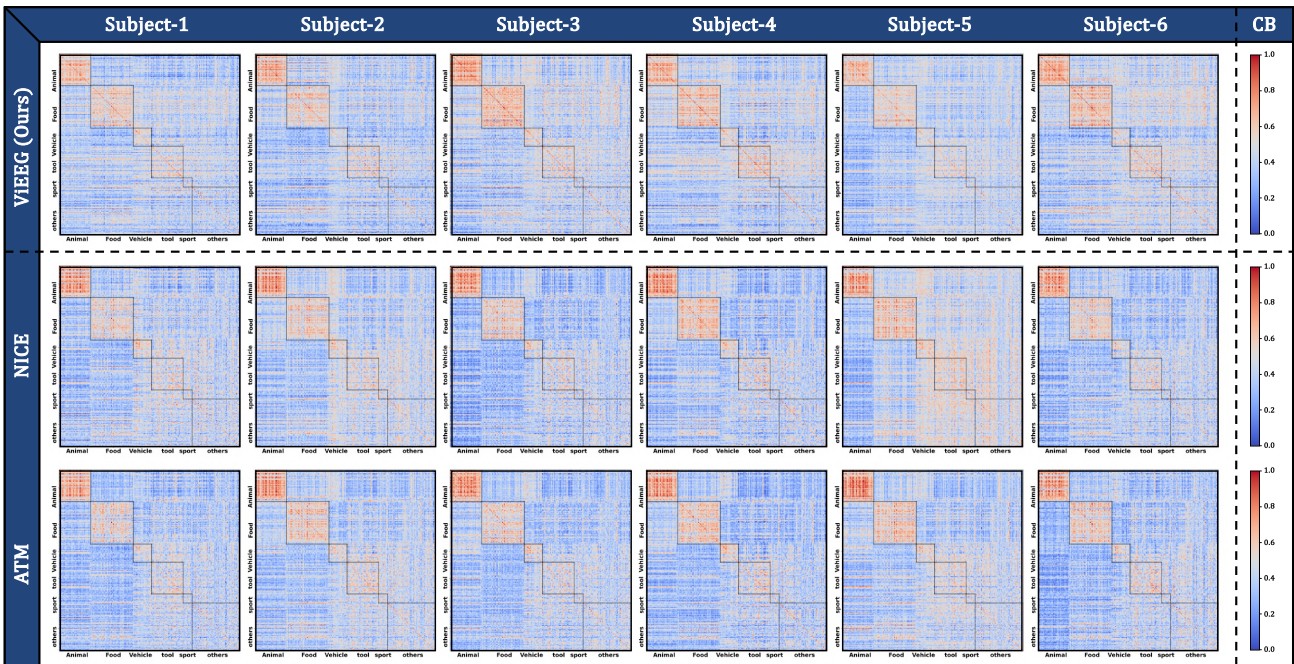

*Figure 11.* Representational similarity matrices across first six subjects: ViEEG vs. NICE vs. ATM (CB: Color bar).

### D.5. Parameter Analysis of Attention Modules

We conducted a detailed parameter analysis on the attention module in ViEEG, focusing on two key parameters: the number of attention layers and the number of attention heads. In our experiments, we recorded Top-1, Top-3, and Top-5 accuracies. For the layer analysis, we experimented with configurations ranging from 0 to 5 layers. In the within-subject object recognition task, a single attention layer yielded the best performance, significantly outperforming the no-attention baseline. Although configurations with 3 or 4 layers occasionally produced slightly higher Top-3 and Top-5 accuracies in the cross-subject task, the performance differences were negligible while incurring a substantial increase in training parameters. Hence, we adopt a single-layer attention architecture for its optimal balance between performance and computational cost.

For the head analysis, we evaluated five settings (1, 2, 3, 4, and 6 heads). While a configuration with 1 head also achieved acceptable performance, using 3 attention heads consistently provided superior results across all metrics. In the cross-subject setting, the influence of head number was less pronounced, suggesting that the head configuration is less critical for subject-independent generalization. Overall, our findings support the use of a simplified attention module with 1 layer and 3 heads in ViEEG, effectively enhancing feature integration without unnecessary complexity.

## E. Representational Analysis

To further support the analysis in the main text, we visualize the representational similarity matrices (RSMs) of ViEEG, NICE (Song et al., 2024), and ATM (Li et al., 2024) across the first six subjects from the THINGS-EEG dataset, as shown in Figure 11. Each matrix reflects how well different EEG conditions are organized in the learned representation space.

Compared to NICE and ATM, ViEEG consistently exhibits sharper diagonal structures across subjects—especially in categories with dense intra-class variation like Food and Animals—indicating better clustering and internal consistency. In contrast, categories such as Tool and Sports appear more scattered in all methods, likely due to inherent category ambiguity. This figure complements our main analysis by highlighting the stability of ViEEG across individuals.

## F. Model Computing Resource Comparison

Table 10 presents an integrated analysis of computational requirements and recognition performance. The comparison reveals three key insights through the following aspects:

*Table 10.* Comprehensive comparison of computational requirements and efficiency.

| Methods | FLOPs. | Param. | Train time (s) | Test time (s) |
|---|---|---|---|---|
| NICE [ICLR'24] (Song et al., 2024) | 42.7354 M | 2630.940 K | 0.0412 | 0.0058 |
| ATM [NeurIPS'24] (Li et al., 2024) | 98.209 M | 3072.662 K | 0.0524 | 0.0088 |
| CognitionCapturer [AAAI'25] (Zhang et al., 2024) | 178.973 M | 2954.073 K | 0.0591 | 0.0056 |
| **ViEEG** [Ours] | 127.973 M | 7924.488 K | 0.0682 | 0.0062 |

**Computational Efficiency and Accuracy Balance**: CognitionCapturer achieves the highest baseline accuracy (33.3% Top-1/60.5% Top-5) but requires significantly more computations than other methods, with 179.0M FLOPs – over three times that of NICE (42.7M FLOPs, 27.3% Top-1). Our ViEEG demonstrates a balanced design: while using 28% fewer computations than CognitionCapturer (128.0M vs 179.0M FLOPs), it achieves a remarkable 40.9% Top-1 accuracy (7.6% improvement). Notably, ViEEG maintains competitive inference speed (6.2ms), being only marginally slower than CognitionCapturer's 5.6ms despite substantial accuracy gains.

**Parameter Utilization Effectiveness**: The parameter counts reflect distinct design strategies: NICE (2.63M parameters) prioritizes lightweight design but limits performance (27.3% Top-1). CognitionCapturer (2.95M parameters) slightly increases parameters to improve temporal modeling (33.3% Top-1). ViEEG (7.92M parameters) strategically allocates parameters to hierarchical components, enabling multi-scale EEG feature integration. This design yields 74.5% Top-5 accuracy (+14% over CognitionCapturer), validating our spatial-frequency interaction mechanism.

**Practical Deployment Considerations**: ViEEG maintains practical efficiency across metrics: Training time per step (68.2ms) shows only moderate increase compared to CognitionCapturer (59.1ms). Inference latency (6.2ms) closely matches fastest baselines (NICE:5.8ms, CognitionCapturer:5.6ms). Accuracy superiority: Achieves 13.6% Top-1 improvement over NICE with negligible 0.4ms inference delay. The results validate that ViEEG successfully balances model capacity and efficiency through its hierarchical architecture, delivering state-of-the-art performance while maintaining practical deployment feasibility.

The results demonstrate ViEEG's ability to convert increased parameters into significant accuracy enhancements (40.9% vs 33.3% Top-1) while avoiding excessive computational costs. Compared to CognitionCapturer's 179.0M FLOPs, ViEEG reduces computations by 51.0M FLOPs while improving Top-1 accuracy by 7.6%, showing superior efficiency-accuracy trade-off. The test time comparison further confirms that ViEEG's accuracy gains (74.5% vs 60.5% Top-5) come without sacrificing real-time processing capability, making it suitable for practical brain signal applications.

# G. Image Retrieval Results of ViEEG

To provide further insight into the effectiveness of our multi-embedding retrieval strategy, we visualize the zero-shot image retrieval results from Subject 8 in the THINGS-EEG dataset. For each sample, we present the Top-10 retrieval results using four visual embeddings: binary object mask (BOM), foreground object (FO), raw scene (RS), and their concatenated form (Triple embedding), as shown in Figure 12, Figure 13, Figure 14, and Figure 15, respectively.

The retrieval results reveal that each embedding modality has its own strengths under different conditions. BOM performs best when the object has simple and distinctive contours (e.g., balls, cones), where boundary information alone is sufficient for recognition. FO excels when object is visually entangled with the background—such as small or embedded items—where isolating the foreground is critical for accurate decoding. RS generally offers robust performance across diverse scenarios, particularly when scene-level semantics are dominant and image embedding is distinctive.

Most notably, the Triple embedding consistently outperforms individual modalities. This is likely due to the complementary nature of the three features, when at least one embedding retrieves a confident match, the corresponding similarity dominates the joint representation, outweighing the weaker or noisy responses from other embeddings. This fusion mechanism effectively boosts retrieval robustness and increases the likelihood of locating the correct object within the Top-5 predictions.

This multi-perspective retrieval approach is also more aligned with the way the human visual system processes complex stimuli: integrating shape, object, and contextual cues for robust perception. Overall, these results reinforce the reliability of ViEEG's hierarchical decoding framework and highlight the advantages of combining diverse visual representations for EEG-based object recognition.

*Table 11.* Quantitative assessments of reconstruction quality.

| Method | SSIM↑ | Alex(2)↑ | Alex(5)↑ | Incep↑ | CLIP↑ | EffNet-B↓ | SwAV↓ |
|---|---|---|---|---|---|---|---|
| NICE [ICLR'24] (Song et al., 2024) | 0.353 | 0.774 | 0.870 | 0.736 | 0.789 | 0.825 | 0.588 |
| ATM [NeurIPS'24] (Li et al., 2024) | 0.351 | 0.779 | 0.873 | 0.745 | 0.792 | 0.823 | 0.591 |
| CognitionCapturer [AAAI'25] (Zhang et al., 2024) | 0.346 | 0.768 | 0.866 | 0.737 | 0.781 | 0.832 | 0.595 |
| **ViEEG** [Ours] | **0.356** | **0.787** | **0.881** | **0.748** | **0.798** | **0.811** | **0.584** |

## H. Image Reconstruction Results of ViEEG

### H.1. Stable Diffusion XL and IP-Adapter

To enable high-quality image reconstruction from EEG, we leverage Stable Diffusion XL (SDXL) (Podell et al., 2023), a state-of-the-art generative model known for its ability to produce photorealistic and semantically rich images. To bridge the modality gap between neural signals and visual content, we incorporate the IP-Adapter (Ye et al., 2023), which employs dual cross-attention modules to inject conditioning information into the denoising process. Specifically, we use CLIP-ViT-H/14[8] (Radford et al., 2021) to encode image representations, enabling them to effectively influence the denoising path within the SDXL, thus guiding the model toward more precise and semantically coherent outputs.

For improved inference speed, we utilize SDXL-Turbo[9], a streamlined version of SDXL optimized for rapid image generation. This variant maintains high visual fidelity while significantly reducing generation latency, making it well-suited for time-sensitive scenarios such as real-time brain signal decoding and neural interface applications.

### H.2. Image Reconstruction Results of ViEEG

We present additional image reconstruction results from ViEEG on the THINGS-EEG dataset, as shown in Figure 18, Figure 17, and Figure 16. The figures illustrate the reconstruction results in three semantic categories: vehicles, food, and animals. Overall, ViEEG successfully reconstructs images that maintain semantic consistency with the original stimuli. For example, in Figure 18 (vehicles), reconstructed images effectively capture key structural and shape features of cars, airplanes, and boats. Similarly, Figure 17 (food items) shows that ViEEG can generate food-related textures and shapes, demonstrating its ability to retain category-level information. In Figure 16 (animals), the model reconstructs recognizable features such as fur textures and body structures, reflecting a strong alignment with the original stimuli.

However, certain semantic inconsistencies are observed in some cases. For instance, in Figure 16, when the original image is a cat, some reconstructions depict human faces. This could be due to EEG-induced associative activations, where the subject subconsciously links cats to human interactions. Additionally, errors may arise from attention fluctuations during EEG recording, leading to unintended noise in brain activity. These cases highlight potential challenges in EEG visual decoding, emphasizing the need for further refinement in feature alignment and reconstruction fidelity.

### H.3. Quantitative Metrics of Reconstructed Image

We now provide quantitative evaluation of EEG-to-image reconstruction in Table 11, including standard metrics used in prior work (e.g., ATM (Li et al., 2024)). These include:

- **SSIM:** measures low-level structural similarity (e.g., texture, edges);

- **AlexNet (2/5) and Inception:** assess mid/high-level perceptual feature similarity;

- **CLIP:** evaluates alignment in joint vision-language space, relevant for semantic fidelity;

- **EffNet and SwAV:** capture feature compactness and unsupervised visual consistency.

Although the same diffusion model is used, ViEEG achieves consistently better scores due to improved EEG-CLIP alignment. This suggests that our hierarchical EEG encoder contributes to more semantically faithful reconstructions.

---

[8]https://huggingface.co/laion/CLIP-ViT-H-14-laion2B-s32B-b79K
[9]https://huggingface.co/stabilityai/sdxl-turbo

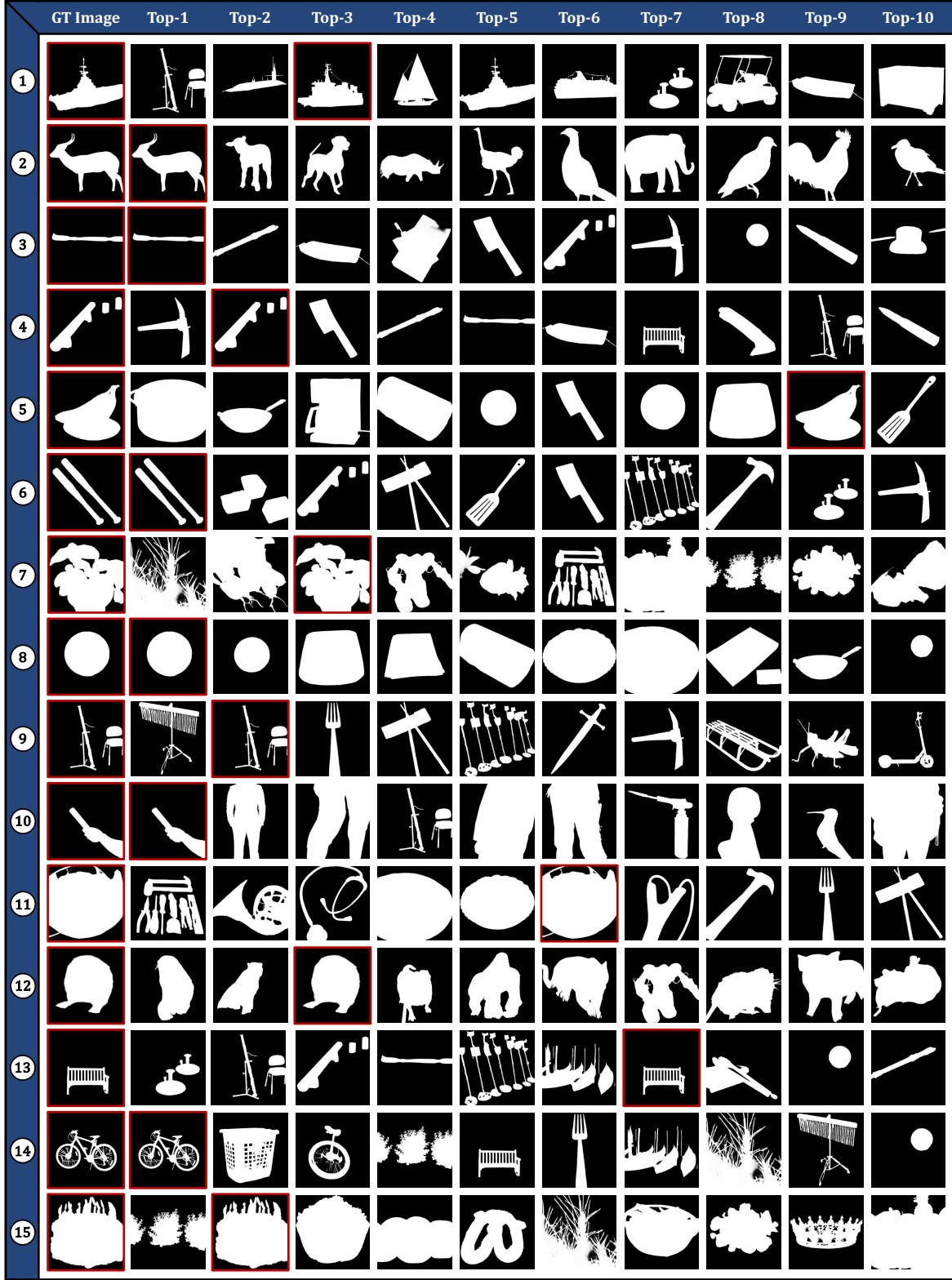

*Figure 12.* Zero-shot image retrieval results for 15 test samples using binary object mask (BOM) embeddings (Visualizations of the Top-10 retrievals for Subject-8 using binary contour-based EEG-to-image features).

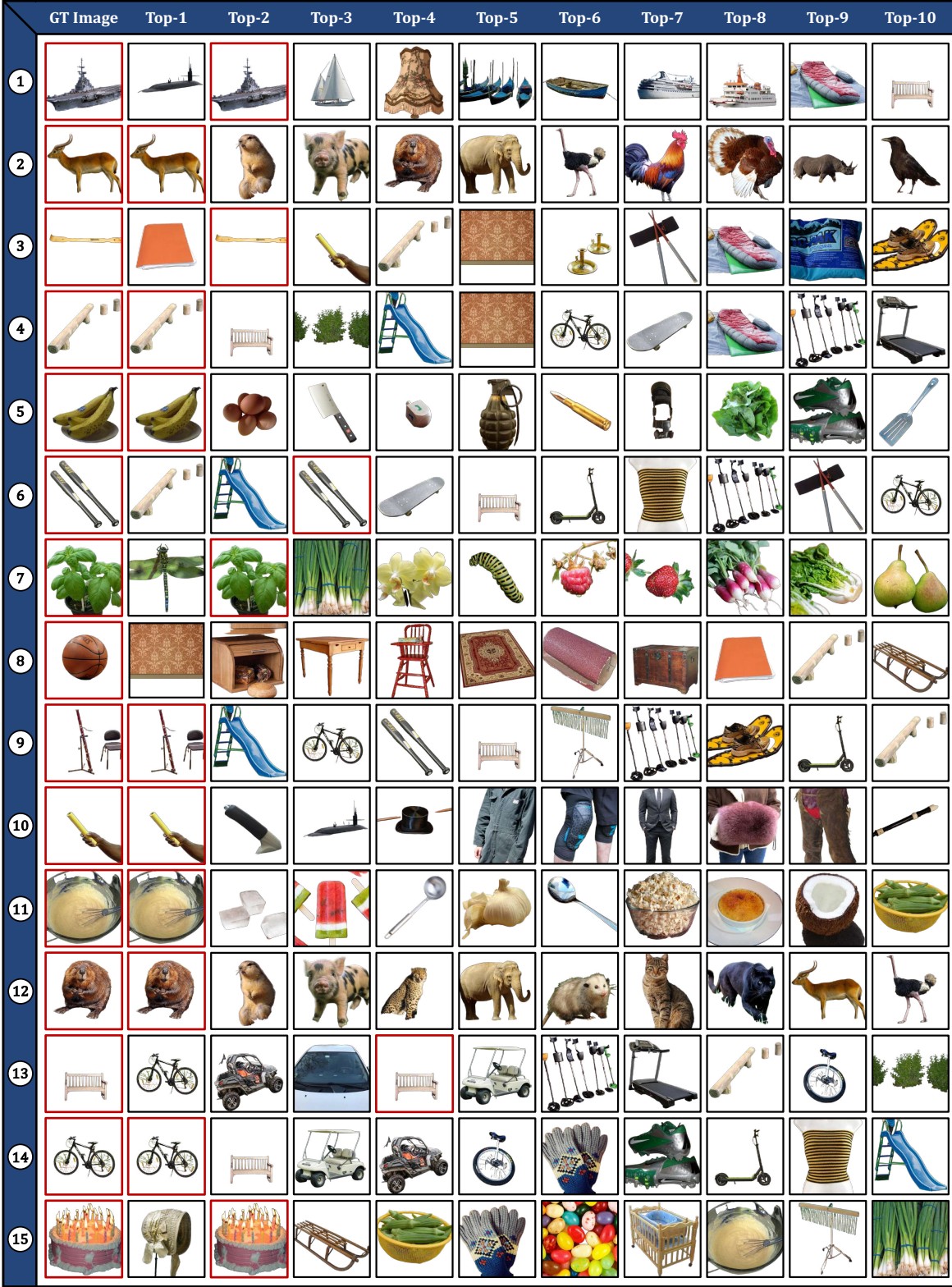

*Figure 13.* Zero-shot image retrieval results for 15 test samples using foreground object (FO) embeddings (Visualizations of the Top-10 retrievals for Subject-8 using EEG features aligned with foreground object representations).

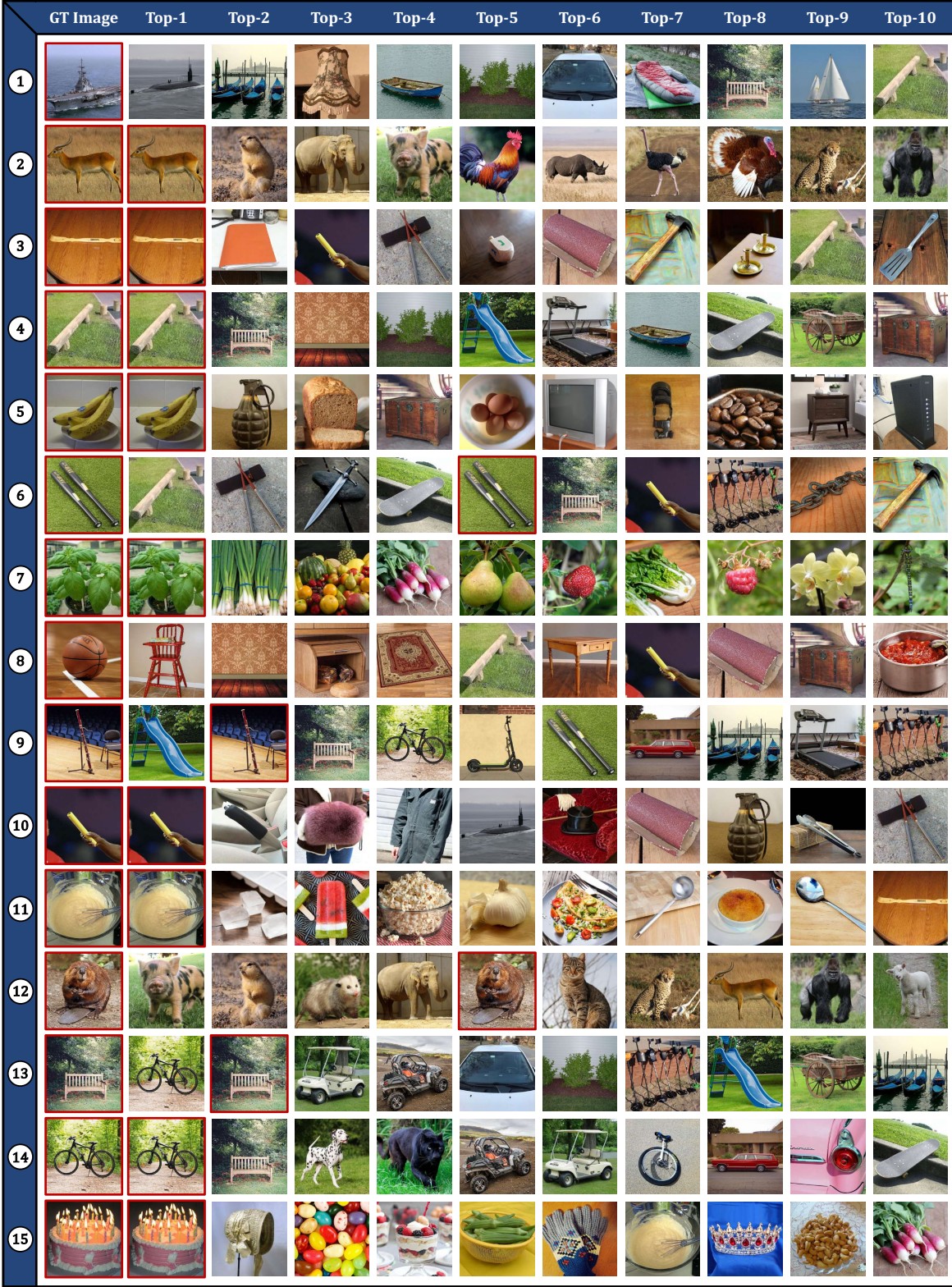

*Figure 14.* Zero-shot image retrieval results for 15 test samples using raw scene (RS) embeddings (Visualizations of the Top-10 retrievals for Subject-8 based on full-scene EEG-to-image alignment).

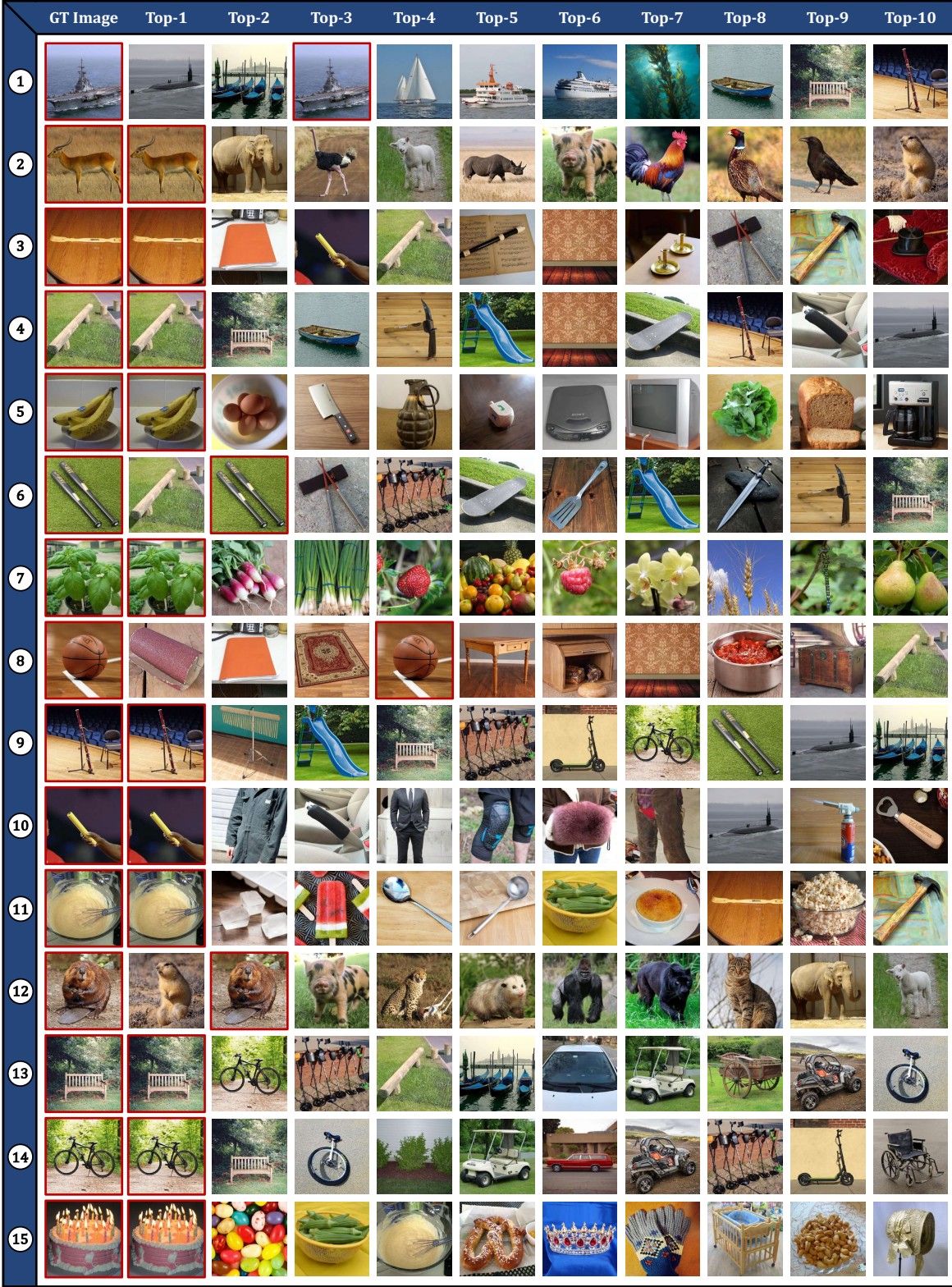

*Figure 15.* Zero-shot image retrieval results for 15 test samples using triple embedding fusion (BOM + FO + RS) (Visualizations of the Top-10 retrievals for Subject-8 using concatenated multi-scale EEG embeddings).

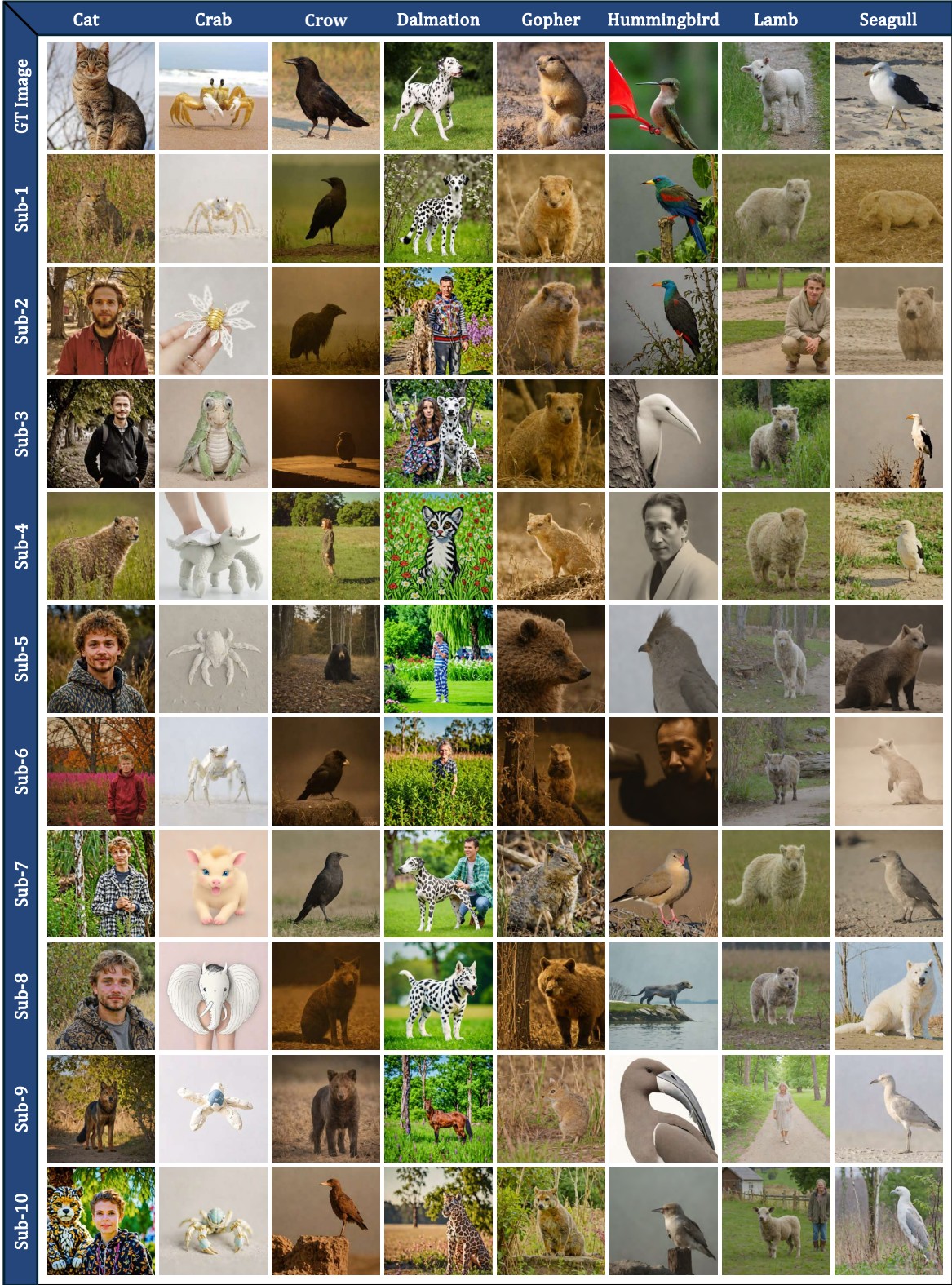

*Figure 16.* Reconstructed images of animals items using ViEEG.

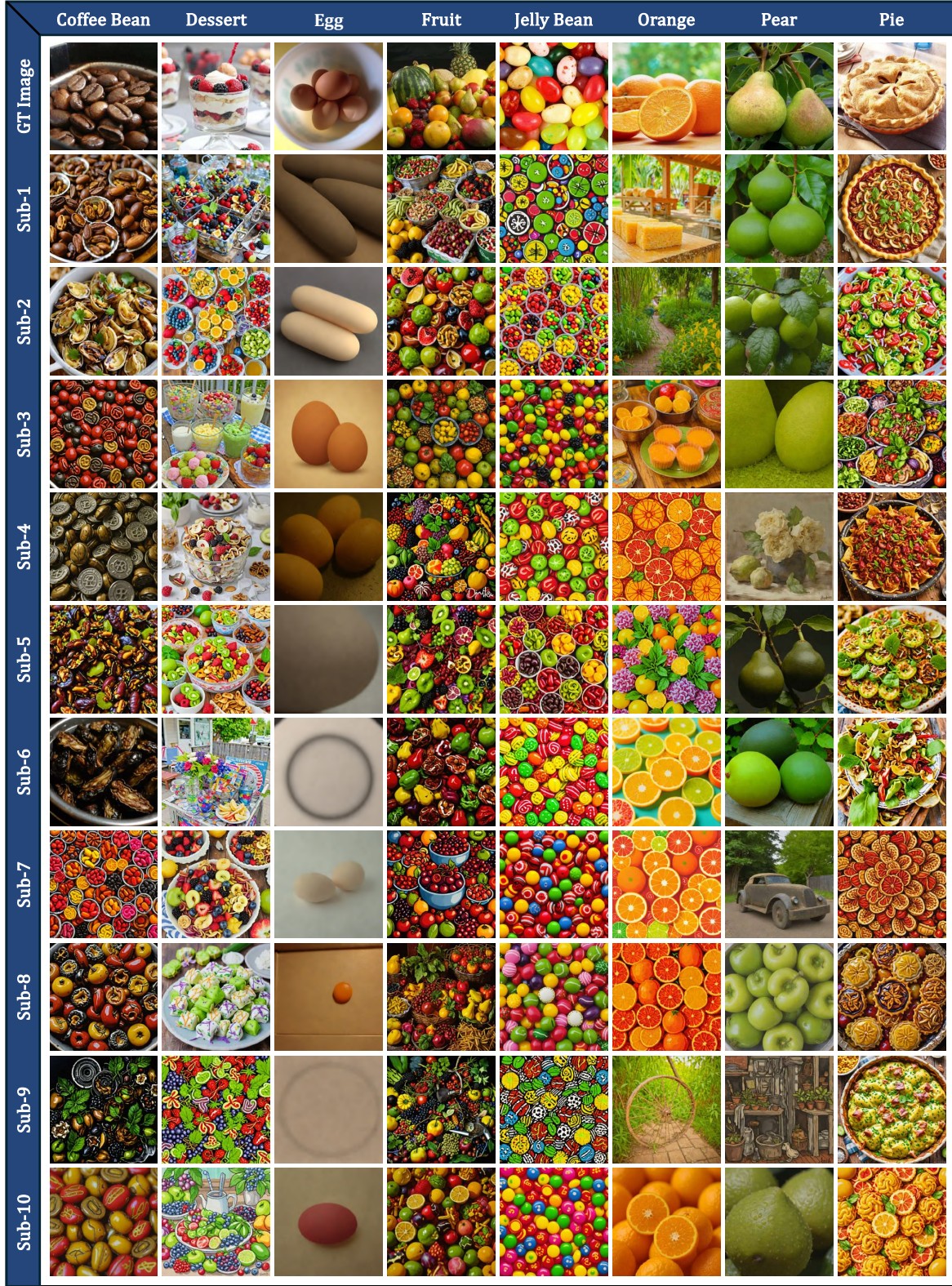

*Figure 17.* Reconstructed images of foods items using ViEEG.

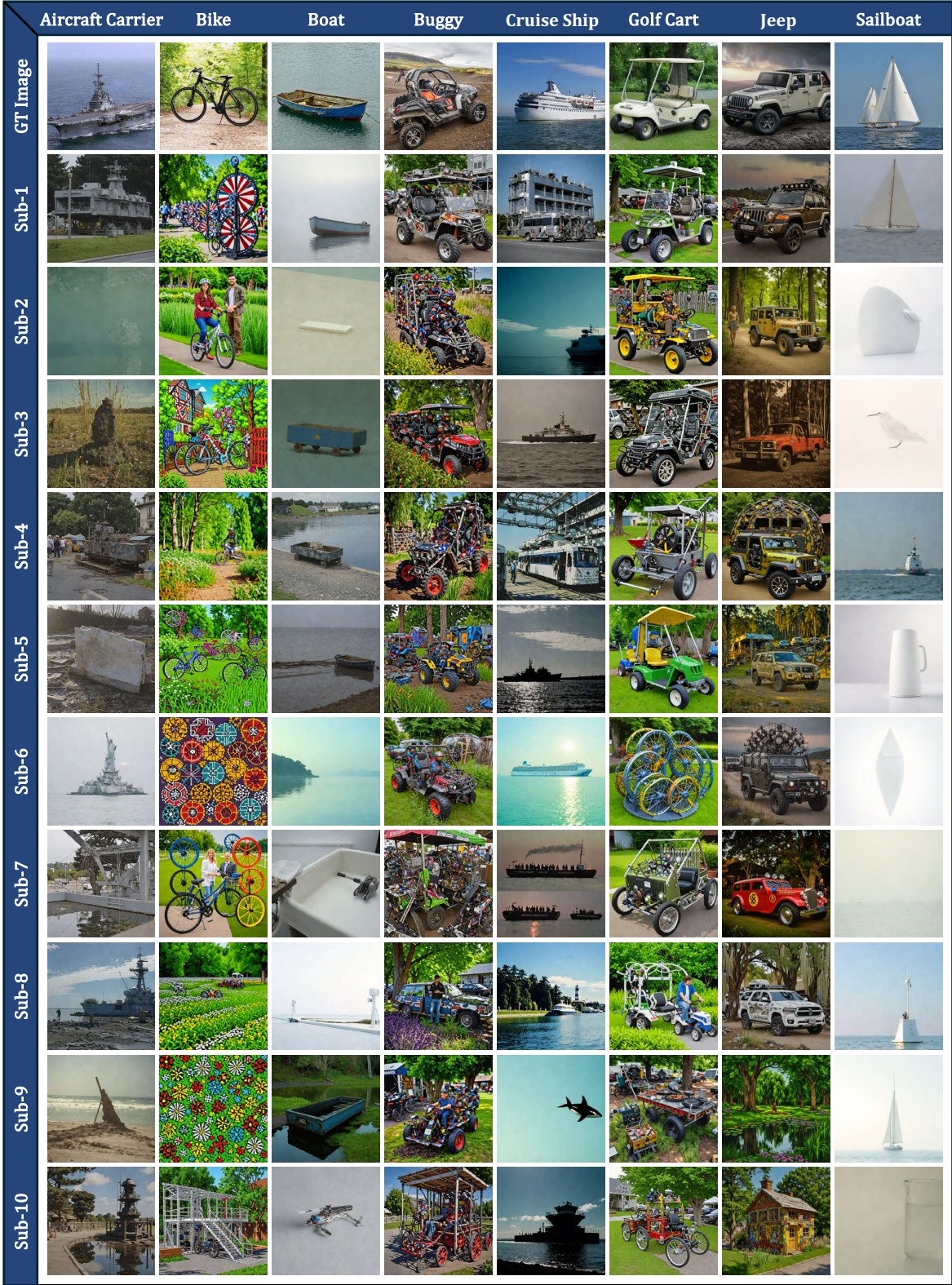

*Figure 18.* Reconstructed images of vehicles items using ViEEG.

