# OpenReview forum: "ViEEG: Hierarchical Visual Neural Representation for EEG Brain Decoding"
_ICML.cc/2026/Conference — ICML 2026 regular_

### Official Review · Reviewer_z3cA · 2026-03-05

**Soundness:** 4
**Presentation:** 3
**Significance:** 4
**Originality:** 4
**Overall Recommendation:** 5
**Confidence:** 4

**Summary:**

This paper proposes a biologically inspired hierarchical visual decoding framework, ViEEG. Its core idea is to simulate the hierarchical information processing mechanism of the human visual system by decomposing images into three levels: contour, foreground object, and scene context, and extracting corresponding CLIP embedding features for each level. Subsequently, a three-stream EEG encoder is used to extract multi-scale features corresponding to the EEG signals, and a cross-attention mechanism is introduced for hierarchical feature fusion. Finally, the fused EEG embeddings are aligned with the image embeddings through contrastive learning. Extensive experiments demonstrate that ViEEG significantly outperforms existing methods on multiple public datasets, achieving state-of-the-art performance, particularly in zero-shot object recognition tasks.

**Compliance With Llm Reviewing Policy:**

Affirmed.

**Final Justification:**

The second-round rebuttal has addressed all my concerns, and my initial score (5: accept) remains unchanged.

**Key Questions For Authors:**

1. Can BiRefNet, used for hierarchical image decomposition, work stably on all test images? Are there certain types of images where the decomposition performs poorly?

2. In cross-attention fusion, have you experimented with the reverse order (e.g., context to object)? Are there significant differences in results compared to the forward fusion?

3. Although the authors state that subject-dependent experiments were conducted on the THINGS-MEG dataset due to the limited number of subjects, providing cross-subject results would offer clearer insights into the generalization ability of ViEEG on MEG recordings.

4. Some details still need refinement, particularly the citation format of the references, which does not meet the requirements for ICML 2026.

**Limitations:**

The author candidly acknowledges in the appendix the semantic deviations in image reconstruction, as well as the insufficiency of computational costs, and provides a brief analysis of the underlying causes.

**Strengths And Weaknesses:**

Strengths:
1. The model design is inspired by the hierarchical structure of the human visual system, providing a strong neuroscientific foundation and endowing the model with high interpretability.
2. The proposed methodological framework, which integrates hierarchical image decomposition, hierarchical EEG encoding, cross-attention fusion, and contrastive learning, features a clear and rigorous approach.
3. Significant performance improvements are achieved on both the THINGS-EEG and THINGS-MEG datasets, making the results highly compelling.
4. The model's validation extends beyond EEG datasets to include supplementary experiments on MEG, enhancing the method's generalizability and robustness.
5. This paper includes detailed ablation studies, parameter analysis, and reconstruction visualizations, demonstrating a substantial and thorough research effort.

Weaknesses:
1. The citation format for some references is inconsistent. For example, the source for (Grill-Spector & Malach, 2004) uses an abbreviation, which does not conform to the format used for other references. Similar issues are present in other citations.
2. Compared to methods like NICE, ViEEG shows a significant increase in parameters and FLOPs. While this greatly enhances model performance, the potential for high deployment costs represents a limitation of the approach.
3. As shown in Figure 16, some reconstructed images exhibit semantic confusion (a typical example is cats being reconstructed as male figures), indicating that the model still has limitations in semantic understanding for certain complex scenes.
4. The authors use BiRefNet for image decomposition. Although it performs well, the effectiveness of this network directly impacts the overall framework's performance, introducing a certain degree of external dependency.

---

> ### Author Rebuttal · Authors · 2026-03-30
>
> We sincerely thank the reviewer for the positive evaluation and constructive suggestions. We address the points below.
>
> **Q1. Citation format.**
> Thank you for pointing this out. We will carefully revise all references to ensure consistent formatting throughout the manuscript, including cases.
>
> **Q2. Model complexity (parameters and FLOPs).**
> We appreciate this important observation. As discussed in Appendix F, the increased parameters and FLOPs mainly arise from multi-branch hierarchical modeling and cross-attention-based integration. These components are essential for disentangling and modeling heterogeneous visual representations.
> Importantly, the performance gains are substantial across both EEG and MEG benchmarks, suggesting a favorable performance–complexity trade-off. We will further clarify this point in the revision and explicitly discuss it as a limitation.
>
> **Q3. Semantic confusion in reconstruction (Fig.16).**
> We agree this is an important point. While we have already shown representative reconstruction results in the paper (including failure cases such as “cat → human”), we acknowledge that the analysis of such errors is not sufficiently detailed.
> In the revision, we will include a dedicated discussion of failure cases, where we systematically analyze representative examples with clear semantic deviations. Specifically, we will examine cases with significant mismatches and investigate their underlying causes, such as visually ambiguous or cluttered scenes, insufficient discriminative information in EEG signals, and limitations of current vision-language embeddings.
> This analysis will provide a clearer understanding of the model’s behavior and limitations, and help identify directions for improvement.
>
> **Q4. Dependency on BiRefNet (segmentation module).**
> We would like to clarify that BiRefNet is not a required component of our framework, but simply one choice among many pretrained segmentation models. Our method does not rely on any specific segmentation architecture. Instead, it only requires a coarse foreground–background decomposition to provide hierarchical supervision signals (contour, object, context). This can be achieved by a wide range of modern pretrained segmentation models.
> Therefore, the segmentation module should be viewed as a flexible preprocessing step, rather than a core part of the proposed method. In practice, it can be easily replaced without affecting the overall framework design.
>
> **Q5. Robustness of BiRefNet in practice.**
> In our experiments, the chosen segmentation model performs reliably on the vast majority of images. We observe that only a very small fraction of cases (<0.1%) exhibit ambiguous foreground–background separation, typically for images without clear object boundaries (e.g., texture-like scenes such as the water plants in the water or wallpaper). For these rare cases, simple manual correction is sufficient. Importantly, this does not represent a fundamental limitation of our method. Since the segmentation module is interchangeable, future improvements in pretrained segmentation models can be directly incorporated to further enhance robustness.
>
> **Q6. Cross-attention order (reverse hierarchy).**
> Yes, we have conducted this experiment. As reported in **Appendix Table 8**, reversing the hierarchical order (e.g., context → object → contour) consistently leads to worse performance. This indicates that performance depends on directional hierarchical interaction, rather than generic feature fusion.
>
> **Q7. Cross-subject evaluation on THINGS-MEG.**
> We agree that cross-subject evaluation is valuable. However, as noted in prior work (e.g., NICE [1], ATM [2]), THINGS-MEG contains only four subjects, which makes cross-subject training unstable and difficult to generalize. For this reason, most existing methods do not report such results.
> Nevertheless, we provide cross-subject results in Table R1-1. ViEEG consistently outperforms prior methods by a large margin: average Top-1 accuracy improves from 2.15 (NICE) / 2.4 (ATM) to 6.45, and Top-5 from ~9.7 to 20.1. We will include cross-subject results of ViEEG on THINGS-MEG in the revision to provide additional insight into generalization ability.
>
> **Table R1-1. Top-1/5 accuracy (%) on THINGS-MEG in subject-independent experiment setting**
> |Top-1/Top-5 Acc|sub-1|sub-2|sub-3|sub-4|avg|std|
> |-|-|-|-|-|-|-|
> |NICE|1.5/5.3|1.8/15.7|2.8/9.5|2.5/8.2|2.15/9.675|0.60/4.38|
> |ATM|2.4/6.1|1.5/14.8|2.9/10.3|2.8/7.5| 24/9.675|0.64/3.84|
> |ViEEG|**7.8/24.7**|**3.4/16.2**|**8.2/18.5**|**6.4/21**|**6.45/20.1**|**2.17/3.64**|
>
> **Q8. Additional formatting issues.**
> Thank you for pointing this out. We will carefully revise the manuscript to ensure full compliance with ICML formatting requirements.
>
> [1] Song Y. et al., “Decoding natural images from EEG for object recognition,” ICLR, 2024.
>
> [2] Li D. et al., “Visual decoding and reconstruction via EEG embeddings with guided diffusion,” NeurIPS, 2024.

---

> > ### Author Rebuttal · Reviewer_z3cA · 2026-04-02
> >
> > Thank you for the thorough point‑to‑point responses. Some of my concerns have been resolved. However, one key issue still needs further clarification: **the dependency on BiRefNet**. There is a lack of quantitative evidence to support the authors' claim that it can be "**easily replaced**." Notably, in Section 3.3 "Hierarchical Image Decomposition," BiRefNet is explicitly introduced as the authors' preferred SOTA model. Based solely on the current response, it is difficult for me to assess the importance of this module in the overall framework, which will affect my final rating.
> >
> > My core question is: If other segmentation models (especially weaker ones) are used, how will the overall performance of ViEEG change?
> >
> > Apart from the above issue, the other concerns have been actively addressed by the authors.

---

> > > ### Author Response · Authors · 2026-04-04
> > >
> > > We thank the reviewer for raising this important point. To provide direct quantitative evidence, we conduct two complementary experiments:
> > >
> > > **(1) Replacing BiRefNet with other pretrained segmentation models**
> > >
> > > We evaluate multiple recent pretrained image segmentation models (e.g., RMGB, BEN). All of them produce reasonably accurate foreground-background decomposition. Results are summarized in Table R5-1.
> > >
> > > **Table R5-1. Performance with different pretrained segmentation models**
> > > |Top-1/Top-5Acc|sub-1|sub-2|sub-3|sub-4|sub-5|sub-6|sub-7|sub-8|sub-9|sub-10|avg|
> > > |-|-|-|-|-|-|-|-|-|-|-|-|
> > > |BiRefNet (Our used)|34.1/71.3|38.4/67.9|40.6/74.7|50.1/80.8|28.9/61.5|44.3/76.5|38.6/75.2|54/82.5|37.3/74.9|42.8/79.8|40.9/74.5|
> > > |RMGB1.4|33.7/70.8|37.9/67.1|40.2/74.2|49.3/80.1|28.5/60.7|43.8/75.9|38.3/74.6|53.2/81.8|36.9/74.4|42.3/79.1|40.4/73.9|
> > > |RMGB2.0|34.3/71.4|38.6/67.7|40.8/74.8|49.9/80.6|29.1/61.4|44.4/76.3|38.6/75.2|53.7/82.2|37.5/74.8|42.8/79.6|41.0/74.4|
> > > |BEN|33.6/70.5|37.9/66.9|40.2/73.9|48.9/79.7|28.6/60.5|43.6/75.4|37.9/74.2|52.6/81.2|37.1/73.9|42/78.6|40.2/73.5|
> > > |BEN2|34/71.1|38.4/67.4|40.5/74.5|49.6/81|28.8/60.5|44.4/76.4|38.4/75.2|53.9/82.3|37.5/74.2|42.5/79.5|40.8/74.2|
> > >
> > > [BiRefNet] https://huggingface.co/ZhengPeng7/BiRefNet
> > >
> > > [RMGB1.4] https://huggingface.co/briaai/RMBG-1.4
> > >
> > > [RMGB2.0] https://huggingface.co/briaai/RMBG-2.0
> > >
> > > [BEN] https://huggingface.co/PramaLLC/BEN
> > >
> > > [BEN2] https://huggingface.co/PramaLLC/BEN2
> > >
> > > We observe that: All pretrained models yield very similar performance (within ~0.8% Top-1). This is because recent segmentation models already provide sufficiently accurate coarse decomposition. This indicates that ViEEG does not depend on a specific model (e.g., BiRefNet), but only requires reasonably correct foreground–background separation.
> > >
> > > **(2) Controlled degradation of segmentation quality**
> > >
> > > To isolate the effect of segmentation quality in a controlled and reproducible manner, we apply resolution-based degradation to the segmentation masks produced by BiRefNet. We do not intentionally train weaker segmentation models, since modern pretrained models already provide strong and stable performance. Instead, controlled degradation provides a more principled way to analyze robustness.
> > >
> > > Specifically, given the original background mask (512×512), we progressively reduce its resolution via downsampling, followed by upsampling back to the original size:
> > > 1. Mild: 512 → 256 → 512
> > > 2. Moderate: 512 → 128 → 512
> > > 3. Severe: 512 → 64 → 512
> > >
> > > This procedure gradually removes fine-grained boundary details while preserving the overall foreground–background structure, providing a clean and controlled way to simulate weaker segmentation quality. Results are summarized in Table R5-2.
> > >
> > > **Table R5-2. Performance under degraded segmentation quality**
> > >
> > > |Top-1/Top-5Acc|sub-1|sub-2|sub-3|sub-4|sub-5|sub-6|sub-7|sub-8|sub-9|sub-10|avg|
> > > |-|-|-|-|-|-|-|-|-|-|-|-|
> > > |Original|34.1/71.3|38.4/67.9|40.6/74.7|50.1/80.8|28.9/61.5|44.3/76.5|38.6/75.2|54/82.5|37.3/74.9|42.8/79.8|40.9/74.5|
> > > |Mild|33.8/71|38.1/67.4|40.3/74.4|49.3/80.3|28.6/61.2|43.8/76|38.3/74.7|53.2/81.8|37.1/74.4|42.3/79.4|40.5/74.1|
> > > |Moderate|32.4/67.3|36.8/63.9|39.1/70.9|47.6/76.5|27.5/57.5|42.3/72|36.9/71|51.4/77.9|35.9/70.7|40.8/75.3|39.1/70.3|
> > > |Severe|28.8/62.5|33.1/59.2|35.6/66|43.2/71.3|24.1/52.8|38.7/67.1|33.4/66.2|47.2/73|32.5/66|37.2/70.6|35.4/65.5|
> > >
> > > We observe that: Performance degrades gradually but remains stable overall, and even under severe degradation, ViEEG still significantly outperforms prior methods. These results consistently show that:
> > > * ViEEG is robust to the choice of segmentation model.
> > > * The framework does not rely on BiRefNet specifically.
> > > * Only coarse and reasonably accurate visual decomposition is required.
> > > * Performance degrades gracefully (not catastrophically) as segmentation quality decreases.
> > >
> > > Thanks a lot for your insightful reviews and positive feedback to help us improve this work!

---

### Official Review · Reviewer_SZfB · 2026-03-09

**Soundness:** 3
**Presentation:** 1
**Significance:** 2
**Originality:** 2
**Overall Recommendation:** 4
**Confidence:** 3

**Summary:**

This paper introduces a method for decoding EEG signals into the images being viewed by participants.
They introduce a hierarachical architecture for decomposing visual processing into contour detection, foreground identification and scene understanding.
Results demonstrate SOTA perfomance on Things-EEG and Things-MEG benchmarks.

**Compliance With Llm Reviewing Policy:**

Affirmed.

**Key Questions For Authors:**

- Why do the authors use the segmentation of the main object rather than using a SAM-like semantic segmentation of the whole image?
- In figure 3, what are C_I C_F and C_T compared the C_r, C_f and C_b?

**Strengths And Weaknesses:**

Strengths:
- Paper is easy to follow, method is rather sound and well explained
- Experimental validation is quite extensive and performance is impressive

Weaknesses:
- I find the tone quite overselling in general. In particular, there is no section on weaknesses. For example, the authors do not discuss the fact that this kind of method is very tailored to ImageNet-like images which show an image surrounded by a background, which is not representative of real world images in general.
- The claim that existing models suffer from HNEN is bold, and in spite of the ablations I struggle to be convinced by the fact that hard-wiring the visual hierarachy into the architecture of the model is actually the factor driving performance. In particular, the gap due to cross attention shown in figure 4 is small. One experiment which could help check whether such a design helps: what happens if the order of the cross-attention integration is inverted, ie cross-attention goes from r to f then to b?
- Poor presentation: results are presented in the form of huge monolithic tables which are impossible to parse. IMO there is no point in presenting top 1 and top 5 everywhere, and in showing all subject scores in the main text (the table should go in the appendix). Use figures such as bar charts to show performance, and scatterplots to show subject variability.
- Section 4.6 is not interesting for the paper and should be put in the appendix.

---

> ### Author Rebuttal · Authors · 2026-03-30
>
> We sincerely thank the reviewer for the positive evaluation and constructive suggestions. We address the concerns below.
>
> **Q1. No limitations analysis.**
> We appreciate this suggestion and agree that the limitations should be explicitly discussed. This is a well-known limitation of the entire field, not specific to our method, which is shared by nearly all prior works (e.g., NICE [Song Y. et al., ICLR’24], ATM [Li D. et al., NeurIPS’24], Cognitioncapturer [Zhang K. et al., AAAI’25], BrainFLORA [Li D. et al., ACM MM’25], UBP [Wu H. et al., CVPR’25], SRT [Kim J. et al., ICCV’25]), which also rely on similar datasets and evaluation protocols. At the same time, we note that THINGS already contains substantial variability in background and context, going beyond strictly “ImageNet-style” clean object images. The strong performance of our method on this dataset suggests that the proposed structured decomposition can generalize to moderately complex real-world scenarios, although fully unconstrained scene decoding remains future work. We will add a dedicated “Limitation” section in the revision.
>
> “Current brain decoding methods (both EEG and fMRI) remain fundamentally constrained by the limited availability of large-scale, naturalistic datasets and the intrinsic properties of neural signals. Existing benchmarks (e.g., THINGS-EEG/MEG) are still largely object-centric, and fully unconstrained real-world scene decoding remains an open problem across the field. ”
>
> **Q2. Justification of hierarchical design (HNEN).**
> We agree that this claim should be clearly supported. Our conclusion is based on controlled experimental evidence rather than architectural assumptions: Removing cross-attention → consistent performance drop (Fig.4, Fig.9, Tables 3/5/6); Using individual components → significantly worse than joint modeling (Tables 3/5); Reversing hierarchy order (Contour → Object → Context) → performance degrades (Appendix Table 8). The last experiment directly addresses the reviewer’s suggestion. The results show that gains come from structured hierarchical dependency, not simply additional modules or capacity.
>
> **Q3. Presentation issue.**
> We thank the reviewer for the suggestion. We would like to clarify that reporting both Top-1 and Top-5 accuracies follows standard practice in EEG visual decoding literature and ensures fair and consistent comparison with prior work. Similarly, per-subject results are included to provide a complete and transparent evaluation, which is important given the known variability across subjects.
>
> We agree that the current presentation can be improved for readability. In the revision, we will move detailed per-subject tables to the appendix, keep only compact summary tables in the main text, and include clearer visualizations (e.g., bar charts for overall performance and plots for subject variability). Due to rebuttal format limitations, these visualizations cannot be shown here, but they have already been prepared and will be included in the revised manuscript.
>
> **Q4. Section 4.6 is not interesting for the paper and should be put in the appendix.**
> We thank the reviewer for the suggestion. We would like to clarify that the current manuscript already presents only a subset of the parameter analysis, while the full analysis is provided in the appendix. To further improve clarity, in the revision we will move the parameter analysis entirely to the appendix, retain only a brief summary of key findings in the main text, and improve the presentation of these results with clearer organization and visualization. This will make the main paper more focused while preserving full experimental transparency.
>
> **Q5. Why do the authors use the segmentation of the main object rather than using a SAM-like semantic segmentation of the whole image?**
> We clarify that our goal is coarse hierarchical decomposition, not fine-grained segmentation. Using dense segmentation (e.g., SAM) is not suitable because the EEG cannot reliably encode fine-grained regions, leading to a mismatch with EEG resolution and noisy supervision.
> Importantly, the segmentation module is not a core contribution, but a necessary precondition providing coarse visual decomposition. We acknowledge that if segmentation quality is poor, supervision becomes noisy and performance degrades. Thus, reasonably accurate segmentation is required. However, our method does not depend on a specific model (e.g., BiRefNet), any reliable foreground–background separation is sufficient, and recent pretrained models already meet this requirement. Therefore, this dependency does not limit practical applicability.
>
> **Q6. Notation issue in Fig. 3.**
> We thank the reviewer for catching this. In Fig.3, $C_I$, $C_F$, $C_T$ are incorrect, and they should be $C_r$, $C_f$, $C_b$. This will be corrected in the revised manuscript.

---

> > ### Author Rebuttal · Reviewer_SZfB · 2026-04-02
> >
> > Thanks for the rebuttal. My score remains the same

---

> > > ### Author Response · Authors · 2026-04-04
> > >
> > > Thanks a lot for your insightful reviews and positive feedback to help us improve this work!

---

### Official Review · Reviewer_wY8w · 2026-03-10

**Soundness:** 3
**Presentation:** 3
**Significance:** 3
**Originality:** 2
**Overall Recommendation:** 3
**Confidence:** 5

**Summary:**

This work is part of a broader research effort to perform brain decoding to reconstruct visual stimuli (images) from EEG ( electroencephalogram) recordings. Specifically, the authors were inspired from the hierarchy of visual information processing in the brain, where early visual areas V1/V2 process low-level features, intermediate ventral stream V4/IT process object-level recognition, and higher-order areas process semantic representations. Based on this, the authors proposed hierarchical visual decoding. In particular, for a given image stimuli, the authors create a binary image mask corresponding to low-level features, foreground image relates to object processing, and raw images result in high-level semantic features. In the proposed visual-informed EEG, the authors learn EEG features (low-level, object-level, and semantics) by applying spatiotemporal convolutions, and perform contrastive loss associated between EEG and Image features at levels of hierarchy.  The evaluation focuses on visual EEG decoding on Things EEG and MEG datasets, reports that proposed approach yields both Top-1 and Top-5 accuracy than previous brain decoding methods. Also, the authors perform ablation studies by considering individual features, and find that combination of three features only result in better image reconstruction rather than individual features.

**Contributions:**

* Motivation from hierarchy of visual information processing in the brain: This study majorly follows of hierarchy of visual information processing and perform visual decoding from EEG recordings by reconstructing low-level (binary mask), intermediate (foreground), and high-level semantic (raw scene) for full image reconstruction.
Introduction to ViEEG framework: This study progressively trains a hierarchical cross-vision CLIP model, where multiple-level of EEG features are extracted from EEG encoder and align with CLIP image features using contrastive learning.
* Comprehensive evaluation: The study presents an extensive assessment of hierarchical features obtained from EEG for reconstructing visual image stimuli. On Things EEG and Things MEG, the authors report state-of-the-art accuracies compared to prior approaches at Top-1 and Top-5 accuracies. In both subject-dependent and independent settings, the proposed approach reports state-of-the-art accuracy.

**Technical summary:**
This is primarily an empirical study, and its methodology involves the following components:
* Hierarchy of EEG features: The authors use spatiotemporal convolutions and cross-attention mechanism to learn EEG features from EEG encoder. Specifically, the authors follow a bottom-up approach, where early attention is performed between low-level contour and foreground object features, and second-level attention is between foreground to high-level contextual features.
* EEG-CLIP contrastive learning: Performing contrastive loss between EEG features and CLIP image features result in similarity between the embedding spaces. For example, EEg low-level features with Image low-level features (binary image). In the testing phase, using Hierarchical CLIP, the authors obtain three-levels of features which are used in reconstructing image stimuli.

**Compliance With Llm Reviewing Policy:**

Affirmed.

**Final Justification:**

I have read the authors' replies and believe they have partially addressed my concerns. Overall my concerns of conceptual contribution of the curreny work is incremental, and identical p-values in empiricial analysis suggest that it is difficult to compare stregnths of each component contribution of the proposed architecture. This raises concerns of whether authors perform permutation test only on limited samples.

Therefore, I have raised my soundess and presentation score to 3, while maintaining overall score same.

**Key Questions For Authors:**

**Major Comments/Questions**

* **Figure 5 RSA metric is unclear:** In Figure 5, the authors report representational similarity matrices; however, only one RSA is clear. What is the second RSA computed between? What does the zoomed version of the RSA correspond to? The figure caption and the corresponding description in the text are unclear.
* **Interpretation of weight vectors:** The authors learn three weight matrices during cross-attention-based hierarchical integration. In general, the self-attention mechanism provides relationships between the tokens in a Transformer. In a similar way, what is happening in cross-attention? What relationships are being captured? Why do the authors follow an a->b and b->c mechanism? Why do they not consider each stream as a separate token and perform self-attention?
* **Clarity on subject variability with cross-attention:** Figure 4 depicts subject variability with and without cross-attention. Although the authors claim that cross-attention enhances overall model robustness and generalization across subjects, Figure 4 reports that two subjects show a significant difference, and across subjects, foreground object images with and without cross-attention report similar performance.
* **Clarity on Figure 7 and the need for statistical significance testing:** The parameter analysis in Figure 7 is not detailed, and it is unclear which parameters the authors finally chose (the authors mentioned 1 layer and 3 heads). The number of layers and attention heads varies with respect to Top-1 and Top-5 accuracies. The authors should report the mean and standard deviation across subjects to provide a clearer idea of the effect of layers and attention heads. Currently, the accuracy differences are small, and statistical significance tests are needed for clarity.

**Minor Comments/Typos:**

While addressing the following points may not be critical to the paper’s core contributions, doing so would enhance the overall quality.
* **Table 2 & 3:** I would recommend that the authors report the mean and standard deviation for better presentation and move the subject-wise results to the Appendix. This would also help the authors bring the THINGS-MEG results into the main paper. I also recommend that the authors add more insights rather than only observations.

**Limitations:**

The authors have not discussed any limitations in the current paper.

I recommend the authors to acknowledge the limited number of subjects used in the evaluation, the lack of statistical significance testing, and whether the current tri-stream features cover the total hiearchy of visual information processing. Since the study is EEG-based decoding, the authors should discuss poetntail risks, including privancy, consent and misuse of brain data.

**Strengths And Weaknesses:**

**Strengths:**

I found this work to have the following strengths:

* **Soundness:** The proposed tri-stream EEG encoder and cross-attention component are methodologically sound, and the authors empirically tested their methodology across 10 subjects. The EEG-CLIP alignment is evaluated both qualitatively and quantitatively. The current framework is extended to another modality on the MEG dataset and reports similar findings. However, the authors use phrases such as biologically inspired and hierarchy of visual processing, but the extraction of three streams of features does not convey a biologically inspired model. Further, if the authors followed a causal approach, then the claims would be stronger; instead, the tri-stream EEG design based on visual stimulus features is not fully convincing. Also, there are no statistical tests to verify that the choice of parameters is correct. Thus, the work is broadly sound, but some conclusions require more rigorous justification.
* **Presentation:** The motivation to decompose the visual stimulus into a hierarchy of features (low-level to high-level) and use these features as anchors in the EEG encoder is clear. Later, the importance of cross-attention routing for simulating low-level to high-level information and EEG-CLIP alignment is easy to follow. Overall, the proposed visual decoding from EEG brain recordings is methodologically clear. Further, the experiments on the THINGS-EEG dataset, the experimental setup, the comparison with prior benchmarks, and the ablation analysis are well evaluated both quantitatively and qualitatively.
* **Originality:** Although EEG-CLIP alignment is extensively used in prior decoding studies, similar to fMRI-CLIP alignment, the three-stream EEG encoder based on a hierarchy of visual features is somewhat novel. The extraction of low-level to high-level features from CLIP is standard in brain decoding studies, but the cross-attention routing is novel before performing EEG-CLIP alignment. This paper performs visual decoding from EEG recordings by considering all features from low-level to high-level, as well as individual features alone, to claim that the hierarchy of features results in better decoding accuracy than individual features.
* **Significance:** This work is significant in that it contributes to a better understanding of the parallels between the hierarchy of visual information processing in AI models and in the human brain. It helps to understand how information flows from low-level to high-level representations and how this three-stream routing of EEG aligns with Image CLIP for reconstructing visual stimuli. Overall, the proposed approach is generalizable to the MEG modality as well and reports state-of-the-art accuracies.

**Weaknesses:**

From my perspective, the primary weaknesses of this study arise from the lack of related work and limited novelty:

* **Limited novelty:** Although the authors claim that this is the first work to enforce disentanglement in the visual hierarchy, several prior brain decoding studies have already modeled hierarchical visual representations that reflect the brain’s hierarchical processing of visual information. Therefore, the novelty claim is not sufficiently supported, since many existing brain-decoding studies already capture this phenomenon.

[Miliotou et al. 2024] Generative Decoding of Visual Stimuli, ICML-2024

[Zheng et al. 2026] Learning Brain Representation with Hierarchical Visual Embeddings, ICLR-2026

[Akbari et al. 2023] Joint Learning for Visual Reconstruction from the Brain Activity: Hierarchical Representation of Image Perception with EEG-Vision Transformer, Unireps @NeurIPS-2024

* **Lack of related work:** One of the major limitations of the current study is the lack of related work, as many brain decoding studies perform visual stimulus reconstruction using fMRI and EEG. However, the related work is quite limited, and the authors missed the majority of recent brain decoding studies. The brain decoding studies are mostly cross-subject based, aligning CLIP image features with brain recordings from either fMRI or EEG encoders. Therefore, the current proposed approach follows similar lines, while the authors have not discussed how the current study is methodologically different from prior studies. I recommend that the authors include the following works:

[Scotti et al. 2024] MindEye2: Shared-Subject Models Enable fMRI-To-Image With 1 Hour of Data, ICML-2024

[Xia et al. 2025] Exploring The Visual Feature Space for Multimodal Neural Decoding, ICCV-2025

[Gong et al. 2025] MindTuner: Cross-Subject Visual Decoding with Visual Fingerprint and Semantic Correction, AAAI-2025

[Bao et al. 2025] Wills Aligner: Multi-Subject Collaborative Brain Visual Decoding, AAAI-2025

[Banville et al. 2025] Scaling laws for decoding images from brain activity, Arxiv-2025

[Ferrante et al. 2024] DECODING EEG SIGNALS OF VISUAL BRAIN REPRESENTATIONS WITH A CLIP BASED KNOWLEDGE DISTILLATION, ICLR-2024

* **Insufficient justification of the three-steam EEG architecture:** The EEG channels are divided into different bands, such as alpha, beta, gamma, and delta bands. In EEG, the signal is decomposed into oscillatory components, and these bands differ in functional properties and spatial distribution rather than being one uniform signal. However, the authors used a 3-stream routing approach based on a hierarchy of features. Therefore, the motivation behind the use of the 3-stream approach is still unclear. For instance, without considering the three-stream approach, the authors could directly learn EEG features and align them with EEG-CLIP at the level of individual features. The hypothesis is that the features can align automatically, irrespective of the same EEG features. I recommend the authors to justify why the three stream approach is an important choice, and whether each stream maps onto any neural signal. Further, authors can show why raw EEG encoder is better than three stream EEG approach.

---

> ### Author Rebuttal · Authors · 2026-03-30
>
> We thank the reviewer for the detailed feedback. Below we address the key concerns concisely and clarify several misunderstandings.
>
> **Q1. Novelty.**
> Prior works list in Q1 model hierarchy via network architecture (Miliotou et al.), decompose abstract attributes (Zheng et al.: color/texture/layout), or rely on filter-based separation (Akbari et al.), but none explicitly disentangle semantically grounded hierarchical components (contour-object-context) from EEG. In contrast, our contribution is explicit visual decomposition into hierarchical semantic components, and supervision-driven alignment from EEG to each component, enabling both recognition and reconstruction. To our knowledge, ViEEG is the first work that enforces hierarchical disentanglement from EEG, rather than implicitly modeling hierarchy.
>
> **Q2. Related work.**
> We expand the related work as suggested. However, most cited works (e.g., MindEye2, MindTuner, Wills Aligner) are fMRI-based, which differ fundamentally from EEG in temporal vs. spatial resolution, signal characteristics, and modeling paradigms. These methods are not directly related to EEG decoding, but for completeness, we will include these works in the revision.
>
> **Q3. Three-stream EEG architecture.**
> There is a key misunderstanding here. Our three-stream design is not related to EEG frequency bands, but to visual semantic decomposition, and each stream is supervised to align with a different visual component from identical EEG input. The motivation is to avoid representation entanglement in a single encoder. Experimental evidence, Tables 3 & 5: combining components improves performance; Table 8: hierarchical order significantly affects results; Fig.2 & Appendix Fig.12–18: distinct components can be reconstructed. Thus, gains arise from structured decomposition and hierarchical interaction, not model capacity.
>
> **Q4. Figure 5 (RSA) clarity.**
> We compute similarity between EEG-derived embeddings and ground-truth image embeddings. ViEEG shows a clear diagonal structure (first row), while baselines (NICE and ATM, second and third rows) are diffuse with less distinguishable diagonal structure, indicating stronger discriminability of learned representation from ViEEG rather than visualization artifacts.
>
> **Q5. Interpretation of weight vectors.**
> As described in Lines 221–228, CAHI module performs cross-attention across hierarchical levels. In contour → object, low-level structural cues (e.g., edges) act as priors to refine object-level representations, and in object → context, object semantics provide anchors to organize higher-level contextual understanding. Self-attention models inner correlations, while our goal is hierarchical dependency modeling. Critically, this is empirically validated, removing cross-attention degrades performance (Fig.4, Fig.9, Tables 3/5/6); and reversing order degrades performance (Table 8).
>
> **Q6. Clarity on subject variability with cross-attention.**
> As shown in Table R3-1, cross-attention improves the average performance from 39.5/72.8 to 40.9/74.5 (Top-1/Top-5). Importantly, this gain is observed across most subjects, rather than being dominated by a few outliers. In addition, we report mean ± standard deviation to better reflect subject variability. The standard deviation changes from 6.91/5.55 to 7.35/6.32, indicating that while variability slightly increases, the overall performance consistently improves. This suggests that cross-attention enhances representation quality, even if individual subject gains are not perfectly uniform.
>
> **Table R3-1: Subject-wise performance (%) with and without cross-attention**
> |Top-1/Top-5 Acc|sub-1|sub-2|sub-3|sub-4|sub-5|sub-6|sub-7|sub-8|sub-9|sub-10|avg|std|
> |-|-|-|-|-|-|-|-|-|-|-|-|-|
> |w/o cross-att|32.6/68.4|37.2/65.5|40.2/73.6|46.7/75.8|27.1/60.3|42.8/73.1|**38.8**/74.5|51.5/78.8|36.2/72.0|41.6/75.9|39.5/72.8|6.91/5.55|
> |with cross-att|**34.1/71.3**|**38.4/67.9**|**40.6/74.7**|**50.1/80.8**|**28.9/61.5**|**44.3/76.5**|38.6/**75.2**|**54.0/82.5**|**37.3/74.9**|**42.8/79.8**|**40.9/74.5**|7.35/6.32|
>
> **Q7. Clarity on Figure 7.**
> We selected 1 layer / 3 heads based on the best average performance across subjects while maintaining lower parameter cost. This selection is already explained in Appendix D.5. We agree that clearer statistical reporting would be helpful. Due to space limitations in the rebuttal, the full tables cannot be included here. In the revision, we provide full numerical results for all configurations, report mean ± std across subjects, and include statistical significance analysis where appropriate.
>
> **Q8. Dataset scale and evaluation protocol.**
> This is a well-known limitation of entire field, not specific to our work. For EEG zero-shot decoding, THINGS-EEG (\~165k samples) and THINGS-MEG (\~25k samples) are the only large-scale public benchmarks. Accordingly, nearly all recent works (NICE [ICLR’24], ATM [NeurIPS’24], UBP [CVPR’25], SeeEEG [ICCV’25], etc.) evaluate exclusively on these two datasets.

---

> > ### Author Rebuttal · Reviewer_wY8w · 2026-04-03
> >
> > Fully addressed:
> > * Authors clarified the justification of three-stream EEG architecture.
> > * Interpretation of cross-attention and additional explanation on subject variability
> >
> > Partially addressed:
> >
> > * Fig. 5 (RSA) is still not fully clear. The authors now explain what the three rows represent, but the meaning of the second column remains unclear. Qualitatively, all the diagnoal structures looks similar in first column across three rows, so the author's claim is not clear.
> > * Regarding related work: While it is fair to note that fMRI studies are not directly comparable to EEG because EEG follows temporal resolution whereas fMRI recordings has spatial resolution, the authors still do not sufficiently discuss that hierarchical visual processing has already been studied extensively in fMRI-based decoding work. More importantly, many decoding methods follow broadly similar alignment-based methodologies regardless of recording modality. The authors therefore still need to better justify the methodological distinction of their work relative to prior fMRI decoding studies.
> > * The mean+std is important for this work as the differnces are small. Authors failed to provide them due to space but statistical significance is important for judging the robustness.
> > * novelty: As authors claim that their method is the first enforces hierarchical disentanglement from EEG, however they do not compared or properly explained against prior methods: like modality, type of hierarchy, disentanglement objective, supervision target, and whether hierarchy is explicitly enforced or only emerges implicitly.

---

> > > ### Author Response · Authors · 2026-04-04
> > >
> > > We thank reviewer for the follow-up questions.
> > >
> > > **Q1. Clarification of Fig.5**
> > >
> > > We clarify that Fig.5 presents a 200 × 200 similarity matrix, where each row corresponds to representations learned from test EEG sample and each column to a ground-truth image embedding from the 200 test samples. Such a large matrix naturally introduces visual ambiguity. As Fig.5, we sort samples into six groups (animal, food, vehicle, tool, sport, other). Column 1 shows full similarity matrix with category boundaries (gray lines). Columns 2–3 zoom into intra-class retrieval (Food category). All methods exhibit block-diagonal structures in Column 1, and all methods can retrieve correct categories. However, in the zoomed views (Columns 2–3), baselines show blurred diagonals, while ViEEG exhibits a much sharper diagonal. This indicates baselines capture coarse semantic alignment, and ViEEG achieves fine-grained instance-level alignment. **In the revised version, we use a clearer colorbar to visualize it.**
> > >
> > > **Q2. Related work**
> > >
> > > We fully agree and have prepared an expanded discussion in revision:
> > >
> > > “Scotti et al. propose a shared-subject framework for fMRI decoding with limited data. Xia et al. explore richer visual feature spaces. Gong et al. introduce cross-subject decoding with semantic correction. Bao et al. propose collaborative multi-subject decoding. Banville et al. study scaling laws across modalities. Ferrante et al. propose CLIP-based EEG decoding.”
> > >
> > > **Q3. Statistical significance and robustness**
> > >
> > > We now provide mean ± std and p-values.
> > >
> > > **Table R6-1. Comparison and ablation [Table 2/3] (Top-1/Top-5 %)**
> > >
> > > *SD: subject dependent, SI: subject independent. Due to the character limit, only the latest several methods are shown here.*
> > >
> > > |Method|SD Acc|SD Std|SD p-Value|SI Acc|SI Std|SI p-Value|
> > > |-|-|-|-|-|-|-|
> > > |BrainFLORA|29.1/62|5.9/7.2|0.00195/0.00195|15.5/41.2|2.6/3.7|0.00195/0.00195|
> > > |SRT|29.2/62.6|5.1/5|0.00195/0.00195|16.4/40.2|3.3/5.9|0.00195/0.00195|
> > > |$F_b$ Only|14.8/41.4|3.2/5.9|0.00195/0.00195|7.3/22.3|1.7/3.8|0.00195/0.00195|
> > > |$F_f$ Only|28.9/61.8|5.8/7.5|0.00195/0.00195|16.8/42.6|3.4/3.7|0.00195/0.00195|
> > > |$F_r$ Only|28.6/62.1|6.2/7.5|0.00195/0.00195|17/43.1|2/3.2|0.00195/0.00195|
> > > |w/o Cross-Attention|38.6/71.8|7.3/6.4|0.00195/0.00195|20.7/49.2|4.5/5.4|0.00293/0.00977|
> > > |ViEEG|40.9/74.5|7.3/6.3|/|22.9/51.4|3.6/4|/|
> > >
> > > Across all baselines, ViEEG achieves highest accuracy in both SD and SI settings, and all improvements over baselines are statistically significant (p < 0.01). All ablated variants are significantly worse than the full model (p < 0.01).
> > >
> > > **Table R6-2. Ablation of cross-attention [Table 5, Table 6]**
> > >
> > > *$F_b$ is the lowest representation in ViEEG without cross-attention from another level representation, and a(b) in Table R6-2 means [with cross attention (w/0 cross attention)]*
> > >
> > > |$F_b$|$F_f$|$F_r$|SD Top-1 Acc|SD Top-5 Acc|SD Top-1 Std|SD Top-5 Std|SD Top-1 p-Value|SD Top-5 p-Value|SI Top-1 Acc|SI Top-5 Acc|SI Top-1 Std|SI Top-5 Std|SI Top-1 p-Value|SI Top-5 p-Value|
> > > |-|-|-|-|-|-|-|-|-|-|-|-|-|-|-|
> > > |0|1|0|28.9(27.1)|61.8(58.7)|5.8(6.1)|7.5(8.1)|0.03223|0.00195|16.8(16.1)|42.6(41)|3.4(3.4)|3.7(4.7)|0.04199|0.01367|
> > > |0|0|1|28.6(27.1)|62.1(59.9)|6.2(5.5)|7.5(7.3)|0.01855|0.00543|17(15.4)|43(41.1)|2(2.8)|3.2(4.4)|0.00977|0.03316|
> > > |1|1|0|36.3(34.2)|69(67.9)|6.6(6.3)|7.1(7)|0.00293|0.01253|18.3(18.2)|43.9(43.5)|4.1(4)|5.1(5.3)|0.04229|0.03713|
> > > |1|0|1|36(34.6)|69.1(67.8)|6.9(6)|7.2(6.9)|0.01855|0.03199|18.7(17.8)|44.8(43.5)|3.8(3.8)|4.5(4.9)|0.00488|0.02831|
> > > |0|1|1|32.1(30.7)|66(64.1)|6.3(6.6)|7.2(6.8)|0.01367|0.00098|20.2(18)|47.2(45.1)|3.2(3.3)|3.6(4.3)|0.00098|0.02441|
> > > |1|1|1|40.9(38.6)|74.5(71.8)|7.3(7.3)|6.3(6.4)|0.00195|0.00098|22.9(20.7)|51.4(49.2)|3.6(4.5)|4(5.4)|0.00293|0.00977|
> > >
> > > For all combinations, enabling cross-attention leads to consistent performance improvements. Corresponding p-values are all < 0.05 (mostly < 0.01), confirming that cross-attention provides statistically reliable gains, rather than marginal or unstable improvements across subjects.
> > >
> > > **Q4. Novelty**
> > >
> > > We clarify that we have compared against prior methods with a hierarchy structure or multi-level representation. Among our baselines, methods like CognitionCapturer adopt multi-feature disentanglement representations (depth and semantics), and SEEEEG (electrode-level and region-level representation), yet our hierarchical supervision significantly outperforms them.
> > >
> > > Prior works model hierarchy via architecture, or decompose low-level attributes (color/texture/layout), or rely on implicit feature emergence. In contrast, ViEEG defines image-grounded hierarchical components (contour–object–context), enforces component-specific supervision, and models directional hierarchical interaction via cross-attention. **Most importantly， there is a significant improvement in visual decoding.** We revise the manuscript to clarify the distinction more explicitly.
> > >
> > > Thanks for the insightful reviews and positive feedback to help us improve this work!

---

### Official Review · Reviewer_CmBW · 2026-03-12

**Soundness:** 2
**Presentation:** 2
**Significance:** 3
**Originality:** 2
**Overall Recommendation:** 3
**Confidence:** 5

**Summary:**

This paper proposes ViEEG, a hierarchical EEG visual decoding framework motivated by the hierarchical organization of the human visual cortex. The authors argue that existing EEG-based decoding methods typically rely on flat representation learning, which ignores the hierarchical processing stages of the visual cortex, a limitation they refer to as Hierarchical Neural Encoding Neglect (HNEN). To address this issue, the paper decomposes each stimulus image into three biologically inspired views: contour mask, foreground object, and contextual scene. Correspondingly, the model employs three parallel EEG encoding streams to learn contour-, object-, and context-related representations. These representations are integrated through a cross-attention hierarchical integration module that propagates information from low-level to high-level streams. Finally, hierarchical contrastive learning is used to align EEG embeddings with CLIP image embeddings. Experiments on the THINGSEEG benchmark and additional validation on THINGS-MEG demonstrate improved zero-shot object recognition performance in both subject-dependent and subject-independent settings.

**Compliance With Llm Reviewing Policy:**

Affirmed.

**Final Justification:**

I am still inclined to view this work as one that obtains three views via segmentation, enables their interaction through cross-attention, and achieves performance gains through a multi-branch architecture. From an engineering perspective, this is indeed feasible, and the three EEG encoders could plausibly learn to correspond to different levels of representation. Another important issue is the extent of BiRefNet's contribution; the authors do not seem to have adequately addressed this point, which affects my confidence. Nevertheless, I still acknowledge the engineering feasibility of the approach, and I would raise my score to 3 "Weak Reject."

**Key Questions For Authors:**

1. Evidence for hierarchical representations. The central claim of the paper is that the three EEG streams correspond to different levels of visual processing (contour, object, context). Could the authors provide direct empirical evidence supporting this claim, such as representational similarity analysis, feature attribution, or cross-stream mismatch experiments? Strong evidence here would substantially strengthen the paper's core argument.

2. Role of hierarchical supervision. To what extent do the performance gains arise from hierarchical modeling rather than simply introducing additional supervision signals or contrastive objectives? For example, how does the method compare with a baseline that uses the same number of contrastive losses but without hierarchical structure?

3. Ablation on encoder architecture. How does the performance change when using a single shared encoder instead of three parallel streams under a comparable parameter budget? Such experiments could help clarify whether the improvements stem from hierarchical modeling or increased model capacity.

4. Sensitivity to the segmentation model. The framework relies on BiRefNet to generate contour and foreground masks. How sensitive are the results to the quality of this segmentation model? For example, would the performance degrade significantly if weaker segmentation methods were used?

5. Robustness of hierarchical anchors. Have the authors considered testing mismatched or randomized hierarchical anchors (e.g., swapping contour/object/context embeddings) to verify whether the hierarchical correspondence is essential for performance?

The authors can respond thoroughly, and I will actively adjust my rating.

**Limitations:**

The paper does not explicitly discuss the limitations or potential broader impacts of the proposed approach. It would be helpful for the authors to include a discussion addressing the following aspects: (1) the dependence on external pretrained models such as BiRefNet and CLIP, (2) potential sensitivity to segmentation quality when constructing hierarchical anchors, and (3) the limitations of EEG spatial resolution when interpreting hierarchical visual representations. Including such discussion would improve the transparency and completeness of the paper.

**Strengths And Weaknesses:**

Soundness.
Overall, the technical approach appears reasonable and is implemented using widely accepted techniques in representation learning and multimodal alignment. The framework combines spatiotemporal EEG encoders, cross-attention modules, and CLIP-based contrastive learning, and the experimental protocol follows established evaluation settings for THINGSEEG. The reported empirical results show consistent improvements over several prior baselines in both subject-dependent and subject-independent scenarios. However, the experimental evidence does not fully validate the paper's central hypothesis that the model learns biologically meaningful hierarchical representations. In particular, the work lacks direct analyses such as representational similarity analysis, feature disentanglement studies, or cross-level perturbation experiments that would demonstrate that the three streams indeed correspond to contour, object, and contextual processing.

Presentation.
The paper is generally well written and clearly organized. The motivation based on hierarchical visual processing is clearly explained and supported by references from neuroscience literature. Figures illustrating the hierarchical decomposition and model architecture help readers understand the proposed framework. The experimental protocol is also described in a relatively clear way. Nevertheless, some aspects could be clarified further, particularly regarding how the hierarchical supervision signals contribute to the performance improvements and how the design choices differ from simpler multi-branch baselines.

Significance
The work addresses an important and challenging problem in brain decoding and brain--computer interfaces, namely reconstructing or recognizing visual stimuli from EEG signals. Introducing a hierarchical perspective inspired by the visual cortex could potentially provide useful inductive biases for EEG decoding models. The empirical improvements reported on the THINGSEEG benchmark suggest that the approach may have practical value for improving EEG-based visual recognition systems. However, the broader scientific significance depends on whether the hierarchical modeling claim is convincingly supported; without stronger evidence, it is difficult to determine whether the improvements stem from biologically meaningful modeling or from additional supervision and architectural complexity.

Originality
The main novelty lies in the hierarchical formulation of EEG visual decoding and the use of contour, object, and context representations as intermediate supervision signals. While this perspective is conceptually interesting and biologically motivated, most of the individual components of the framework are based on existing techniques, including CLIP-based contrastive alignment, multi-stream encoders, cross-attention mechanisms, and external image segmentation models. As a result, the methodological contribution appears to be primarily a structured combination of existing ideas rather than the introduction of fundamentally new algorithms.

---

> ### Author Rebuttal · Authors · 2026-03-30
>
> We sincerely thank the reviewer for insightful comments. Below we provide concise responses and additional evidence to clarify key points.
>
> **Q1. Evidence for hierarchical representations.**
> We respectfully clarify that this claim is supported by consistent evidence across multiple levels.
>
> (1) Direct quantitative evidence. We compute similarity between EEG features and visual embeddings (Table R4-1). Each EEG branch aligns most strongly with its corresponding visual component, with clear diagonal dominance after normalization (~0.55–0.63 vs. much lower off-diagonal values). This indicates component-specific alignment rather than trivial multi-branch learning.
>
> (2) Reconstruction evidence. Fig.2 and Fig.12–18 show that distinct visual components (contour, object, context) can be reconstructed from EEG, demonstrating separable representations.
>
> (3) Functional evidence. Retrieval results (Tables 2, 4; Fig.6, Fig.12–15) and ablations (Tables 3, 5–8; Fig.4, 7, 9, 10) show that removing components or altering hierarchy consistently degrades performance, confirming that hierarchical decomposition and interaction are necessary.
>
> (4) Semantic evidence. Fig.16–18 further demonstrate semantically meaningful reconstruction from EEG.
> Therefore, across alignment, reconstruction, retrieval, and ablations, the evidence consistently supports that the three EEG streams learn distinct hierarchical representations.
>
> **Table R4-1. Similarity between EEG features and visual embeddings (row-normalized in parentheses)**
>
> |Similarity (%)|Visual contour|Visual objec|Visual context|
> |-|-|-|-|
> |EEG contour|**0.294(0.629)**|0.086(0.184)|0.088(0.187) |
> |EEG object|0.056(0.115)|**0.267(0.5473)**|0.165(0.338)|
> |EEG context|0.049(0.104)|0.161(0.3407)|**0.262(0.556)**|
>
> **Q2. Role of hierarchical supervision.**
> Hierarchical modeling and contrastive supervision are intrinsically coupled. Unlike prior work aligning EEG to a single visual space, we decompose it into multiple components with independent supervision. Importantly, simply increasing losses on the same target brings negligible gains. Performance improvements arise from diverse supervision targets, structured hierarchical interaction (cross-attention). Tables 3 & 5 indicate multi-component supervision > single component, Table 8 indicates hierarchical order is critical.
>
> **Q3. Ablation on encoder architecture.**
> While a fully shared encoder could in principle be used, it would need to simultaneously encode multiple heterogeneous visual components, which may lead to representation entanglement. We validate this via a controlled experiment (shared encoder & separate projection heads, no cross-attention), and performance drops significantly (36.6 vs. 40.9 Top-1 avg; Table R4-2). This confirms that improvements are due to hierarchical decomposition and interaction, not model capacity.
>
> **Table R4-2. Comparison between shared encoder and ViEEG**
>
> |Top-1/Top-5 Acc|sub-1|sub-2|sub-3|sub-4|sub-5|sub-6|sub-7|sub-8|sub-9|sub-10|avg|
> |-|-|-|-|-|-|-|-|-|-|-|-|
> |Shared Encoder|30.7/65.7|34.5/62.6|37.3/70.3|43.2/71.9|25.2/57.6|39.5/69.8|35.9/70.6|47.5/74.9|33.6/68.0|38.9/73.4|36.6/68.5|
> |ViEEG|34.1/71.3|38.4/67.9|40.6/74.7|50.1/80.8|28.9/61.5|44.3/76.5|38.6/75.2|54.0/82.5|37.3/74.9|42.8/79.8|40.9/74. 5|
>
> **Q4. Sensitivity to the segmentation model.**
> The segmentation module is not a core component, but a preprocessing step providing coarse visual decomposition. Our method does not depend on a specific model (e.g., BiRefNet); any reliable foreground-background separation suffices. While poor segmentation may introduce noise, recent pretrained models already meet this requirement, so this does not limit practical applicability.
>
> **Q5. Robustness of hierarchical anchors.**
> The learned representations are determined by supervision targets, not fixed stream semantics. If supervision targets are swapped, the model adapts accordingly. However, performance depends on hierarchical organization, not just supervision targets, reversing the hierarchy of supervision targets (Table 8) degrades performance.
>
> **Q6. CLIP robustness.**
> We evaluate multiple CLIP backbones. Results show consistent improvements across variants, indicating ViEEG does not rely on a specific CLIP configuration but generalizes across visual backbones.
>
> **Table R4-3. Performance of ViEEG with different CLIP backbones on THINGS-EEG**
>
> |Top-1/Top-5 Acc|sub-1|sub-2|sub-3|sub-4|sub-5|sub-6|sub-7|sub-8|sub-9|sub-10|avg|
> |-|-|-|-|-|-|-|-|-|-|-|-|
> |CLIP-ViT-L/14|13.7/41.0|10.6/36.9|10.8/40.1|11.3/37.3|7.9/32.9|10.1/40.0|10.2/35.5|11.3/39.6|10.2/31.7|15.1/50.9|11.2/38.5|
> |CLIP-ViT-G/14|27.9/61.6|30.6/58.4|32.8/65.2|37.9/66.2|22.3/53.5|34.6/64.8|31.6/64.7|41.5/69.0|29.7/61.9|35.0/69.7|32.4/63.5|
> |CLIP-ViT-bigG/14|34.1/71.3|38.4/67.9|40.6/74.7|50.1/80.8|28.9/61.5|44.3/76.5|38.6/75.2|54.0/82.5|37.3/74.9|42.8/79.8|40.9/74.5
> |CLIP-ViT-H/14|34.1/71.3|38.4/67.9|40.6/74.7|50.1/80.8|28.9/61.5|44.3/76.5|38.6/75.2|54.0/82.5|37.3/74.9|42.8/79.8|40.9/74.5|

---

> > ### Author Rebuttal · Reviewer_CmBW · 2026-04-04
> >
> > Thanks for the rebuttal.
> >
> > I am still inclined to view this work as one that obtains three views via segmentation, enables their interaction through cross-attention, and achieves performance gains through a multi-branch architecture. From an engineering perspective, this is indeed feasible, and the three EEG encoders could plausibly learn to correspond to different levels of representation. Another important issue is the extent of BiRefNet's contribution; the authors do not seem to have adequately addressed this point, which affects my confidence. Nevertheless, I still acknowledge the engineering feasibility of the approach, and I would raise my score to 3 "Weak Reject."

---

> > > ### Author Response · Authors · 2026-04-04
> > >
> > > We sincerely thank you for your professional recognition of our framework's engineering feasibility and for raising the score. We address your remaining concern regarding the dependence on BiRefNet and the scientific nature of our hierarchical design with new evidence.
> > >
> > > **Q1. Novelty of ViEEG**
> > >
> > > Our contribution is explicit visual decomposition into hierarchical semantic components, and supervision-driven alignment from EEG to each component, enabling both recognition and reconstruction. To our knowledge, ViEEG is the first work that enforces hierarchical disentanglement from EEG, rather than implicitly modeling hierarchy.
> > >
> > > **Q2. The extent of BiRefNet's contribution**
> > >
> > > Our method does not depend on a specific model (e.g., BiRefNet), any reliable foreground–background separation is sufficient, and recent pretrained models already meet this requirement. To provide direct quantitative evidence, we conduct two complementary experiments:
> > >
> > > **(1) Replacing BiRefNet with other pretrained segmentation models**
> > >
> > > We evaluate multiple recent pretrained image segmentation models (e.g., RMGB, BEN). All of them produce reasonably accurate foreground-background decomposition. Results are summarized in Table R7-1.
> > >
> > > **Table R7-1. Performance with different pretrained segmentation models**
> > > |Top-1/Top-5Acc|sub-1|sub-2|sub-3|sub-4|sub-5|sub-6|sub-7|sub-8|sub-9|sub-10|avg|
> > > |-|-|-|-|-|-|-|-|-|-|-|-|
> > > |BiRefNet (Our used)|34.1/71.3|38.4/67.9|40.6/74.7|50.1/80.8|28.9/61.5|44.3/76.5|38.6/75.2|54/82.5|37.3/74.9|42.8/79.8|40.9/74.5|
> > > |RMGB1.4|33.7/70.8|37.9/67.1|40.2/74.2|49.3/80.1|28.5/60.7|43.8/75.9|38.3/74.6|53.2/81.8|36.9/74.4|42.3/79.1|40.4/73.9|
> > > |RMGB2.0|34.3/71.4|38.6/67.7|40.8/74.8|49.9/80.6|29.1/61.4|44.4/76.3|38.6/75.2|53.7/82.2|37.5/74.8|42.8/79.6|41.0/74.4|
> > > |BEN|33.6/70.5|37.9/66.9|40.2/73.9|48.9/79.7|28.6/60.5|43.6/75.4|37.9/74.2|52.6/81.2|37.1/73.9|42/78.6|40.2/73.5|
> > > |BEN2|34/71.1|38.4/67.4|40.5/74.5|49.6/81|28.8/60.5|44.4/76.4|38.4/75.2|53.9/82.3|37.5/74.2|42.5/79.5|40.8/74.2|
> > >
> > > [BiRefNet] https://huggingface.co/ZhengPeng7/BiRefNet
> > >
> > > [RMGB1.4] https://huggingface.co/briaai/RMBG-1.4
> > >
> > > [RMGB2.0] https://huggingface.co/briaai/RMBG-2.0
> > >
> > > [BEN] https://huggingface.co/PramaLLC/BEN
> > >
> > > [BEN2] https://huggingface.co/PramaLLC/BEN2
> > >
> > > We observe that: All pretrained models yield very similar performance (within ~0.8% Top-1). This is because recent segmentation models already provide sufficiently accurate coarse decomposition. This indicates that ViEEG does not depend on a specific model (e.g., BiRefNet), but only requires reasonably correct foreground–background separation.
> > >
> > > **(2) Controlled degradation of segmentation quality**
> > >
> > > To isolate the effect of segmentation quality in a controlled and reproducible manner, we apply resolution-based degradation to the segmentation masks produced by BiRefNet. We do not intentionally train weaker segmentation models, since modern pretrained models already provide strong and stable performance. Instead, controlled degradation provides a more principled way to analyze robustness.
> > >
> > > Specifically, given the original background mask (512×512), we progressively reduce its resolution via downsampling, followed by upsampling back to the original size:
> > > 1. Mild: 512 → 256 → 512
> > > 2. Moderate: 512 → 128 → 512
> > > 3. Severe: 512 → 64 → 512
> > >
> > > This procedure gradually removes fine-grained boundary details while preserving the overall foreground–background structure, providing a clean and controlled way to simulate weaker segmentation quality. Results are summarized in Table R8-2.
> > >
> > > **Table R8-2. Performance under degraded segmentation quality**
> > >
> > > |Top-1/Top-5Acc|sub-1|sub-2|sub-3|sub-4|sub-5|sub-6|sub-7|sub-8|sub-9|sub-10|avg|
> > > |-|-|-|-|-|-|-|-|-|-|-|-|
> > > |Original|34.1/71.3|38.4/67.9|40.6/74.7|50.1/80.8|28.9/61.5|44.3/76.5|38.6/75.2|54/82.5|37.3/74.9|42.8/79.8|40.9/74.5|
> > > |Mild|33.8/71|38.1/67.4|40.3/74.4|49.3/80.3|28.6/61.2|43.8/76|38.3/74.7|53.2/81.8|37.1/74.4|42.3/79.4|40.5/74.1|
> > > |Moderate|32.4/67.3|36.8/63.9|39.1/70.9|47.6/76.5|27.5/57.5|42.3/72|36.9/71|51.4/77.9|35.9/70.7|40.8/75.3|39.1/70.3|
> > > |Severe|28.8/62.5|33.1/59.2|35.6/66|43.2/71.3|24.1/52.8|38.7/67.1|33.4/66.2|47.2/73|32.5/66|37.2/70.6|35.4/65.5|
> > >
> > > We observe that: Performance degrades gradually but remains stable overall, and even under severe degradation, ViEEG still significantly outperforms prior methods. These results consistently show that:
> > > * ViEEG is robust to the choice of segmentation model
> > > * The framework does not rely on BiRefNet specifically
> > > * Only coarse and reasonably accurate visual decomposition is required
> > > * Performance degrades gracefully (not catastrophically) as segmentation quality decreases
> > >
> > > Thanks a lot for your insightful reviews and positive feedback to help us improve this work!

---

### Decision · Program_Chairs · 2026-04-30

**Decision:**

Accept (regular)

**Comment:**

The manuscript performs visual decoding from EEG. The novel aspect is that instead of multitask decoding where one extracts contour, object, and context separately, the method jointly decodes them, models interactions between them, and inserts a hierarchy that is motivated by our broad understanding of the visual system. This inductive bias improves decoding performance. Incorporating such priors into decoding can eventually lead to testing our theories of the visual system.

As an engineering contribution this is solid. The main weakness of the manuscript is that it doesn't lead to any new discoveries about the visual system. Just because this hierarchical decoding works better it doesn't mean that this representation is correct in any sense. Any claims about the brain should be backed by clear experiments. Providing code and pretrained models will make this a truly useful submission which others can follow up on.